# Early fibrotic niches establish tumour-permissive microenvironments

Erik C. Cardoso[1,2,9], Hyeyoung Lee[3,9], Frances J. England[1], Hyunjin Cho[3], Robin Lu[2], Sagar S. Varankar[1], Moo Suk Park[4], Natasha Rekhtman[5], Bon-Kyoung Koo[6], Benjamin D. Simons[1,7,8], Jinwook Choi[1,3 ✉] & Joo-Hyeon Lee[1,2 ✉]

Pathologic transformation represents a critical yet poorly defined window during which mutant epithelial stem cells actively construct the microenvironment that enables tumour initiation[1,2]. Here using integrated single-cell, spatial and functional analyses, we define the earliest multicellular events that licence this transition following oncogenic activation in the lung. *Kras^{G12D}*-mutant alveolar type II cells rapidly adopt regenerative-like states that act as signalling hubs, orchestrating coordinated stromal and immune reprogramming while enhancing epithelial plasticity. Through secretion of amphiregulin, mutant epithelial cells activate EGFR signalling in adjacent fibroblasts, inducing a fibrotic, injury-like programme. Reprogrammed fibroblasts, in turn, expand and reprogramme alveolar macrophages, amplifying inflammatory signalling and reinforcing epithelial plasticity. These reciprocal interactions establish a self-sustaining epithelial–stromal–immune circuit that generates a tumour-permissive niche before malignant outgrowth. Disruption of the amphiregulin–EGFR axis prevents early niche formation and abrogates tumour initiation. Conservation of this programme in *KRAS^{G12D}*-inducible human alveolar organoids and early-stage lung adenocarcinoma tissues identifies epithelial–microenvironment communication as a therapeutically actionable vulnerability and suggests that intercepting niche formation may prevent progression to treatment-resistant disease.

Oncogenic mutations can disrupt stem cell homeostasis, driving uncontrolled expansion of mutant cells and initiating malignant transformation[2]. However, tumour development extends beyond genetic alterations, involving dynamic interactions between mutant cells and their non-cancerous neighbours that shape evolving tumour ecosystems through spatiotemporal remodelling of the microenvironment[1,3,4]. Recent single-cell profiling studies have revealed early pathologic changes in both mutant cells and the surrounding stroma. However, it remains unclear how mutant cells orchestrate niche remodelling during tumour initiation—a process establishing tumour-permissive microenvironments and defining a crucial window for intervention[4,5]. This knowledge gap is particularly critical for lung adenocarcinoma (LUAD), in which early targeting of initial ecosystem changes could improve patient survival, but most cases are diagnosed at advanced, treatment-resistant stages with limited therapeutic options and poor outcomes.

In the lung, alveolar type II (AT2) cells serve as resident stem cells responsible for maintaining homeostasis in gas-exchange regions and enabling epithelial repair after injury[6–9]. Dysregulated expansion and differentiation of AT2 cells following oncogenic activation contribute to LUAD pathogenesis, with AT2 cells identified as a major cell of origin[5,7,8,10–12]. Pdgfrα-expressing alveolar fibroblasts support AT2 cell function and regeneration, providing structural scaffolding and essential paracrine signals[6,13–15]. Injury repair induces distinct fibroblast states that mediate extracellular matrix (ECM) remodelling and reciprocal epithelial regulation, whereas immune cells, especially interstitial macrophages, modulate AT2 cell fate through inflammatory signals[16–21]. Despite these advances, the mechanisms by which early tumour–niche interactions establish preneoplastic microenvironments remain unknown. Resolving how initial signals from mutant cell reprogramme the niche could identify effective intervention strategies before treatment resistance emerges.

We previously identified a regenerative epithelial state that emerges during lung regeneration and is co-opted in early tumorigenesis[7]. Here we define niche remodelling that co-evolves with *Kras^{G12D}*-mutant AT2 cells as they transition through this state. Using lineage tracing, single-cell profiling and human organoid models, we show that mutant AT2 cells activate an amphiregulin (Areg)–EGFR axis that reprogrammes fibroblasts and alveolar macrophages (AMs). Disrupting this circuit blocks mutant cell reprogramming and expansion. Inducible

[1]Cambridge Stem Cell Institute, Jeffrey Cheah Biomedical Centre, University of Cambridge, Cambridge, UK. [2]Developmental Biology Program, Sloan Kettering Institute, Memorial Sloan Kettering Cancer Center, New York, NY, USA. [3]Department of Life Sciences, Integrated Institute of Biomedical Research, Gwangju Institute of Science and Technology, Gwangju, Republic of Korea. [4]Division of Pulmonary and Critical Care Medicine, Department of Internal Medicine, Severance Hospital, Yonsei University College of Medicine, Seoul, Republic of Korea. [5]Department of Pathology and Laboratory Medicine, Memorial Sloan Kettering Cancer Center, New York, NY, USA. [6]Center for Genome Engineering, Institute for Basic Science, Daejeon, Republic of Korea. [7]Gurdon Institute, University of Cambridge, Cambridge, UK. [8]Department of Applied Mathematics and Theoretical Physics, Centre for Mathematical Science, University of Cambridge, Cambridge, UK. [9]These authors contributed equally: Erik C. Cardoso, Hyeyoung Lee. ✉e-mail: jinchoi@gist.ac.kr; leej49@mskcc.org

$KRAS^{G12D}$ human LUAD organoids identify AREG$^{high}$ epithelial states sufficient to initiate fibrotic niche formation. $EGFR^{L858R}$-mutant AT2 cells engage a conserved Areg–EGFR circuit, indicating a shared mechanism of niche construction across subtypes. Together, these findings define a spatiotemporal signalling axis driving tumour–niche co-evolution and uncover a targetable window to prevent treatment-resistant disease.

## Fibrotic niches arise through regenerative programmes

To define early microenvironmental changes during lung tumorigenesis, we used the $Kras^{G12D}$ multicolour reporter (Red2Kras) crossed with $Sftpc–Cre^{ERT2}$ mice, enabling stochastic labelling and tracking of mutant AT2 cells[7,22] (Fig. 1a). As RFP$^+$$Kras^{G12D}$ AT2 cells undergo clonal expansion through a regenerative-like state within 2 weeks of oncogenic activation, we performed single-cell transcriptomic profiling of mesenchymal and immune compartments at this nascent stage to delineate niche establishment preceding tumour formation[7].

We first analysed mesenchymal cells (CD31$^-$CD45$^-$EpCAM$^-$) from $Sftpc–Cre^{ERT2}$;Confetti (homeostasis) and $Sftpc–Cre^{ERT2}$;Red2Kras (oncogenesis) lungs 2 weeks after induction (Fig. 1a and Extended Data Fig. 1a). Single-cell profiling of 9,210 cells identified major mesenchymal populations, including alveolar fibroblasts (Col13a1, Tcf21 and Scube2), adventitial fibroblasts (Col14a1, Pi16 and Dcn), smooth muscle cells (Acta2, Myh11 and Thsd4), peri-bronchial fibroblasts (Csmd1, Hhip and Fgf18), pericytes (Pdgfrβ, Cspg4 and Postn), mesothelium (Wt1, Msln and Aqp1) and proliferating cells (Mki67 and Birc5) (refs. 23,24) (Extended Data Fig. 1b–d). Notably, a distinct fibroblast cluster emerged exclusively in Red2Kras lungs, marked by Fst, Tnc, Runx1 and Runx2, here termed 'reprogrammed fibroblasts' (Extended Data Fig. 1b–d). Red2Kras lungs also showed enrichment of cycling and mesothelial-like cells (Celf4, Lgals7, Npl and Wt1os) (Extended Data Fig. 1b–d).

Subclustering alveolar, adventitial and reprogrammed fibroblasts confirmed the selective presence of reprogrammed fibroblasts in tumours (Fig. 1b,c). Trajectory analyses suggested transition from alveolar fibroblasts (Extended Data Fig. 1e). Differential gene expression analysis revealed upregulation of fibrotic and injury-associated genes, including Acta2, Pdgfrβ, Runx1 and Runx2, with Gene Ontology enrichment for ECM remodelling, wound repair and tissue development[16–20,24] (Fig. 1d and Extended Data Fig. 1f,g). Immunofluorescence confirmed Pdgfrβ$^+$Acta2$^+$Runx1$^+$ fibroblasts with reduced Pdgfrα adjacent to RFP$^+$ mutant cells within tumours, indicating fibrotic transition[16], whereas such markers were rare in homeostatic lungs (Fig. 1e and Extended Data Fig. 1h). Integration with transcriptomic datasets from bleomycin-induced alveolar injury revealed a shared fibroblast population enriched in both injury and Red2Kras tumours, with parallel alveolar-to-reprogrammed transitions[24] (Extended Data Fig. 1i–o).

Collectively, these data demonstrate that at the preneoplastic stages, alveolar fibroblasts undergo regenerative-like fibrotic reprogramming, establishing a tumour-associated mesenchymal niche.

## Immune landscape shifts at tumour onset

To profile immune cell dynamics, stromal preparations (1:1 mixture of EpCAM$^-$CD45$^-$ and EpCAM$^-$CD45$^+$ fractions) were analysed, focusing on CD45$^+$ (Ptprc) immune cells, which comprised most captured cells (Extended Data Fig. 2a). Sixteen immune cell clusters were identified on the basis of canonical markers, including monocyte-derived macrophages, AMs, neutrophils and diverse lymphocyte subtypes (Extended Data Fig. 2b–d)[21,23,25].

Subclustering macrophages identified a distinct AM population in Red2Kras lungs transcriptionally divergent from homeostatic AMs (Fig. 1f,g). These 'reprogrammed AMs' retained canonical AM markers, such as SiglecF and MertK, but were distinguished by induction of genes, including Msr1, Cdh1 and Ch25h (Fig. 1h). They showed a hybrid inflammatory profile, co-expressing pro-inflammatory (IL-1a and IL-1b) and anti-inflammatory (Mrc1, Chil3 and Arg1) genes, with reduced MHC-II expression (H2-Ab1 and H2-Eb) and elevated chemokines (Cxcl2 and Cxcl16) implicated in neutrophil and γδ T cell recruitment[26,27] (Fig. 1h and Extended Data Fig. 2e). Flow cytometry confirmed expansion of CD64$^+$SiglecF$^+$ AMs with diminished MHC-II expression, alongside increased CD64$^+$SiglecF$^-$ interstitial and/or monocyte-derived macrophages (Fig. 1i–k). Immunofluorescence revealed marked accumulation of macrophages with high PD-L1 and Msr1 but low MHC-II expression specifically in intertumour regions (Fig. 1l,m and Extended Data Fig. 2f,g). Orthotopic engraftment of RFP$^+$ mutant organoids into $CCR2–Cre^{ERT2}$;ZsGreen mice showed that most tumour-associated macrophages were ZsGreen$^-$, indicating a resident AM origin rather than monocyte recruitment (Extended Data Fig. 3a,b).

Concomitantly, immunosuppressive T cell subsets, including regulatory T and PD-1$^+$ T cells, were expanded in Red2Kras lungs[28–30] (Extended Data Fig. 2d,h,i). Despite no noticeable expansion, CD8$^+$ T cells showed exhaustion features, upregulating Cd160, Btla and Havcr2 and shifting metabolism from oxidative phosphorylation towards glycolysis (Extended Data Fig. 2d,j–l)[31]. Neutrophils, including SiglecF$^{high}$ mature neutrophils, and γδ T cells were also enriched (Extended Data Fig. 2d,m–q)[32–34]. Spatial mapping revealed the accumulation of neutrophils and γδ T cells within tumours, whereas AMs were preferentially enriched in intertumour regions (Extended Data Fig. 2r,s).

Functional depletion of AMs by intratracheal clodronate liposomes markedly reduced tumour growth and impaired neutrophil and γδ T cell recruitment (Fig. 1n–p and Extended Data Fig. 3c–l). This was consistent with the upregulation of Cxcl2 and Cxcl16 in reprogrammed AMs, with their cognate receptors Cxcr2 and Cxcr6 predominantly expressed by neutrophils and γδ T cells, respectively, in Red2Kras lungs[32,35] (Fig. 1i and Extended Data Figs. 2e and 3f).

Altogether, these findings demonstrate early alveolar immune remodelling, in which expansion and reprogramming of resident AMs coordinate inflammatory and immunosuppressive circuits that establish a tumour-supportive microenvironment at the preneoplastic stage.

## Spatiotemporal tumour–niche circuits

To delineate how stromal populations shape the oncogenic niche, we mapped the temporal and spatial dynamics of fibroblasts and macrophages relative to RFP$^+$ mutant AT2 clones over 1–8 weeks following oncogenic activation. By 1 week, Pdgfrβ$^+$Runx1$^+$ reprogrammed fibroblasts were detected in direct contact with nascent RFP$^+$ tumours, and by 2 weeks nearly all expanding tumours were surrounded by fibrotic fibroblasts that persisted thereafter, indicating that fibrotic reprogramming initiates at tumour onset (Fig. 2a,c). By contrast, macrophage remodelling emerged later. Msr1$^+$ reprogrammed macrophages remained comparable with homeostasis at 1 week but became prominent from 2 weeks to 4 weeks, suggesting fibroblast reprogramming precedes major macrophage expansion and phenotypic changes (Fig. 2b,d,e). At this later stage, Pdgfrβ$^+$ fibroblasts were found both within RFP$^+$ tumours and along their borders, where they closely associated with expanded macrophages in intertumour regions (Fig. 2f). To investigate whether reprogrammed fibroblasts directly modulate AM phenotype, wild-type AMs were co-cultured with mesenchymal cells isolated from wild-type or 4-week Red2Kras lungs. Red2Kras mesenchyme promoted AM expansion and decreased MHC-II expression compared with controls, indicating direct fibroblast-mediated regulation of AM phenotype (Extended Data Fig. 4a–c).

To identify mediators of this interaction, we examined the top differentially expressed gene between reprogrammed and homeostatic alveolar fibroblasts and found tenascin-C (Tnc), an immunomodulatory ECM protein, markedly enriched in reprogrammed fibroblasts[36–38] (Extended Data Fig. 4d). Notably, its receptor, Toll-like receptor 4 (TLR4),

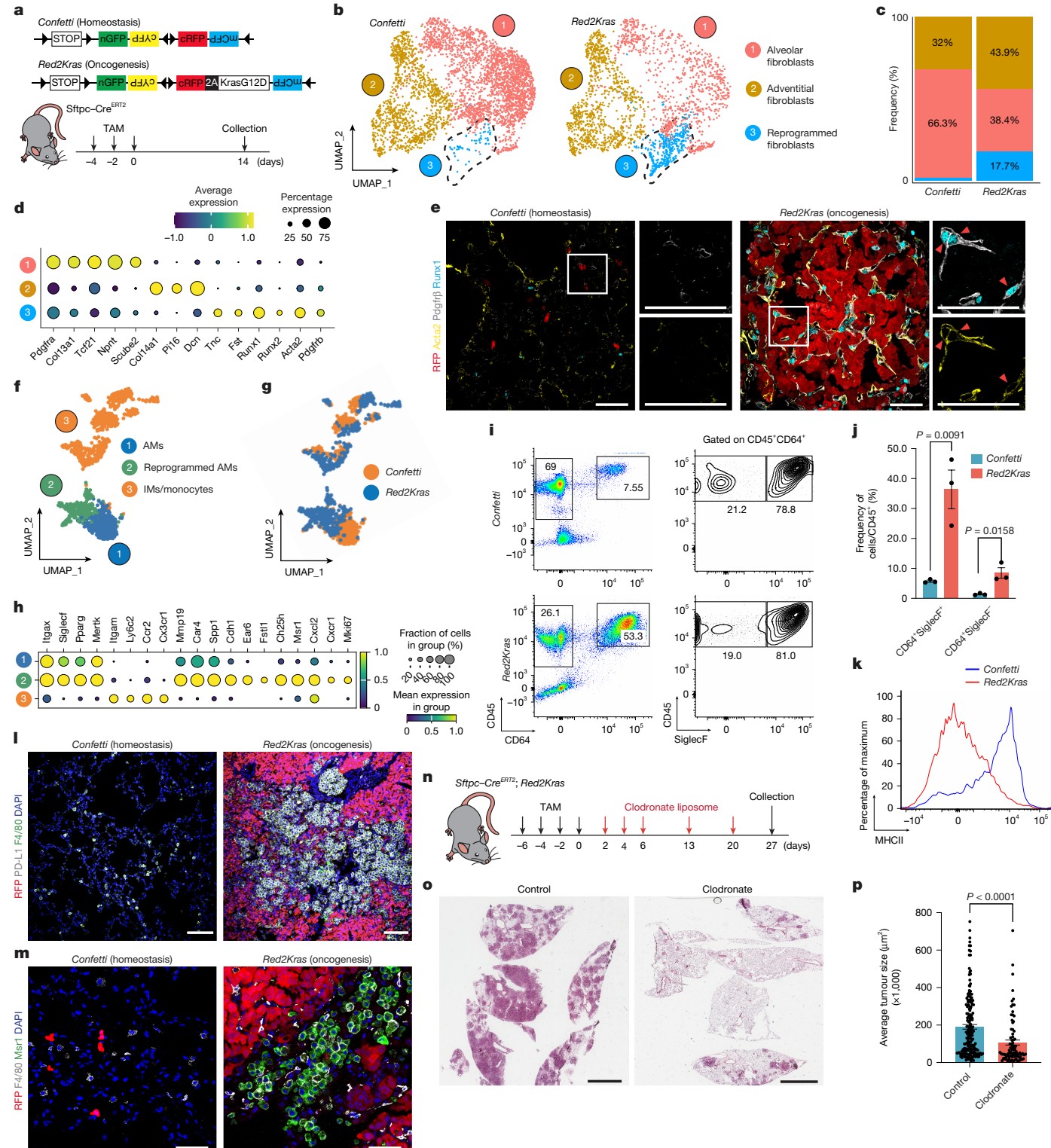

**Fig. 1** | See next page for caption.

was selectively and highly expressed on AMs, suggesting a Tnc–TLR4 axis in fibroblast-driven AM remodelling (Extended Data Fig. 4d). Immunofluorescence confirmed high Tnc expression in fibrotic Pdgfrβ⁺ fibroblasts adjacent to RFP⁺ mutant cells, especially at tumour borders, where Tnc⁺Pdgfrβ⁺ fibroblasts frequently engaged CD68⁺ macrophages at 4 weeks, coincident with macrophage remodelling (Fig. 2f). In vitro Tnc treatment of wild-type AMs induced proliferation and phenotypic remodelling, with increased Msr1 and Ki67 expression, effects

abrogated by the TLR4 inhibitor TAK-242 (Extended Data Fig. 4e–h). Likewise, co-culture of wild-type AMs with *Red2Kras* mesenchymal cells, but not wild-type mesenchyme, induced AM reprogramming that was abrogated by TLR4 inhibition (Extended Data Fig. 4i–k), identifying Tnc as fibrotic-fibroblast-derived cues driving AM expansion and remodelling through TLR4.

Given that inflammatory fibroblasts arise from alveolar fibroblasts before fibrotic changes during injury repair[16,17,19], we asked whether a

**Fig. 1 | Niche reprogramming supports early tumorigenesis. a**, Experimental design for labelling wild-type and mutant cells in *Confetti* and *Red2Kras* lungs. **b**, Uniform manifold approximation and projection (UMAP) showing the distribution of fibroblasts in *Confetti* and *Red2Kras* lungs. Cells are coloured by population. Dashed lines indicate reprogrammed fibroblasts enriched in *Red2Kras* lungs. **c**, Percentage of fibroblast subsets in **b. d**, Dot plot of key fibroblast marker genes annotated in **b. e**, Representative confocal images of reprogrammed fibroblast and lineage-labelled AT2 cells at 2 weeks post-oncogenic activation. Runx1, blue; Acta2, yellow; Pdgfrβ, grey and RFP, red. Images representative of $n = 3$ animals. **f**, UMAP showing all macrophage subsets. **g**, UMAP showing the distribution of *Confetti* and *Red2Kras* macrophages. **h**, Dot plot of key macrophage markers annotated in **f. i**, Flow cytometry of alveolar and monocyte-derived interstitial macrophages at 4 weeks post-oncogenic activation. **j**, Quantification of macrophage subsets from **i**. Data are presented as mean ± s.e.m. Each dot represents one mouse. *Confetti* ($n = 3$) and *Red2Kras* ($n = 3$) mice. *P* values were calculated using two-tailed unpaired *t*-test. **k**, Relative mean fluorescence intensity (MFI) of MHC-II expression in AMs (SiglecF⁺CD64⁺) gated in **i. l,m**, Representative confocal images of AMs and lineage-labelled AT2 cells at 4 weeks post-oncogenic activation. 4′,6-Diamidino-2-phenylindole (DAPI), blue; F4/80 (pan macrophage), green; PD-L1, grey and RFP, red (**l**). DAPI, blue; F4/80, grey; Msr1, green and RFP, red (**m**). Images representative of $n = 3$ animals. **n**, Experimental scheme for macrophage depletion by clodronate liposomes. **o**, Representative haematoxylin and eosin staining of control and clodronate-liposome-treated lungs. **p**, Quantification of average tumour size. Data are presented as mean ± s.e.m. Each dot represents an individual tumour mass. Control ($n = 4$) and clodronate-treated ($n = 3$) mice. *P* values were calculated using two-tailed unpaired *t*-test. Scale bars, 50 μm (**e,m**), 100 μm (**l,o**).

similar population emerges in early oncogenesis. Single-cell profiling revealed a subset of *Red2Kras* reprogrammed fibroblasts enriched for inflammatory markers, including *Lcn2*, *Saa3*, *Sfrp1* and *Cxcl12* (Extended Data Fig. 4l,m). Immunofluorescence identified Lcn2⁺Pdgfrα⁺ inflammatory fibroblasts emerging from 4 weeks post-induction, localizing predominantly with expanded macrophages at tumour peripheries and absent from inner RFP⁺ tumour areas (Fig. 2g,h). Unlike injury repair, these cells were rare at 1 week and lacked fibrotic markers, such as Tnc, indicating that inflammatory and fibrotic fibroblasts represent distinct mesenchymal populations arising during early tumorigenesis (Fig. 2h and Extended Data Fig. 4n,o). Finally, co-culture with AMs from *Red2Kras* lungs robustly induced Lcn2 expression in wild-type Pdgfrα⁺ fibroblasts, an effect recapitulated by interleukin-1β (IL-1β) treatment (Extended Data Fig. 4p,q).

Altogether, these findings demonstrate sequential tumour–niche remodelling initiated by early fibrotic fibroblast reprogramming at tumour onset. Fibrotic fibroblasts drive AM expansion and phenotypic rewiring through a Tnc–TLR4 axis, whereas expanded AMs amplify local inflammatory signalling to induce inflammatory fibroblasts and immune cell recruitment, reinforcing early multicellular niche circuits (Fig. 2i).

## Mutant epithelial hubs drive niche remodelling

We next asked how oncogenic AT2 cells initiate dynamic tumour–niche formation. Building on our recent findings that *Kras*^G12D activation drives AT2 cell reprogramming into a damage-associated transient progenitor (DATP)-like regenerative state, we integrated single-cell transcriptomic datasets from *Red2Kras* fibroblasts and lineage-labelled RFP⁺ mutant cells to identify signals orchestrating fibroblast reprogramming[7] (Extended Data Fig. 5a,b). CellChat revealed enhanced interactions between DATP-like cells and alveolar fibroblasts compared with AT2 cells, with the EGF–EGFR axis emerging as a top candidate (Extended Data Fig. 5c–g). Among enriched ligands, Areg displayed the highest interaction probability and was specifically upregulated in DATP-like states (Extended Data Fig. 5e–h). Immunofluorescence confirmed robust Areg induction in DATP-like cells from day 4 post-induction, persisting during tumour expansion and absent in homeostasis, establishing Areg upregulation as a defining feature of *Kras*-driven AT2 reprogramming (Fig. 3a,b and Extended Data Fig. 6a,b).

To directly test whether oncogenic AT2 cells remodel fibroblasts through EGFR activation, we first treated lung mesenchymal cells with Areg. Areg induced morphological changes and fibrotic programmes, marked by elevated Pdgfrβ and Acta2 expression (Extended Data Fig. 6c–e). By contrast, Areg stimulation did not alter the phenotype in AMs or RFP⁺ mutant cells, indicating selective fibroblast responsiveness (Extended Data Fig. 6f–n). To further interrogate mutant AT2–fibroblast crosstalk, we established organoid co-cultures of lineage-labelled *Pdgfrα*⁺ fibroblasts from *Pdgfrα–Cre*^ERT2;*ZsGreen* lungs with either tdTomato⁺ wild-type or RFP⁺ mutant AT2 cells, from

*tdTomato* or *Red2Kras* lungs, and treated with the EGFR inhibitor gefitinib (Extended Data Fig. 6o). Fibroblasts co-cultured with mutant AT2 cells acquired a fibrotic phenotype, marked by Pdgfrβ upregulation and organoid wrapping that recapitulated the in vivo architecture. Both effects were decreased by EGFR inhibition (Extended Data Fig. 6p). CellChat highlighted ECM-related pathways as potential mediators of fibroblast–tumour interactions (Extended Data Fig. 5i–j). These data demonstrate that DATP-like RFP⁺ mutant states reprogramme the surrounding fibroblasts towards a fibrotic phenotype through EGFR activation to initiate niche formation.

To dissect downstream immune effects, we established organoid tri-cultures combining freshly isolated wild-type mesenchyme, AMs and either wild-type or mutant AT2 cells from *tdTomato* or *Red2Kras* lungs (Fig. 3c). AMs co-cultured with fibroblasts and mutant AT2 cells showed reduced MHC-II expression and increased Msr1 expression relative to AMs with wild-type AT2 cells, and both changes were prevented by gefitinib treatment (Fig. 3d–g). To determine whether mutant cells act directly on AMs or primarily through fibroblasts, we compared AMs co-cultured with RFP⁺ mutant cells alone versus mutant cells plus wild-type mesenchyme (Extended Data Fig. 7a,b). Msr1 induction occurred only when fibroblasts were present, demonstrating fibroblast dependence (Extended Data Fig. 7c,d). AM expansion, however, was triggered by tumour cells even without fibroblasts (Extended Data Fig. 7e). These data indicate that fibrotic fibroblasts reprogramme AM phenotype downstream of EGFR-mediated signals from mutant cells, whereas mutant cells can independently support AM expansion. Exogenous Areg or Areg/Ereg stimulation in tri-cultures of wild-type mesenchyme, AMs and AT2 cells was sufficient to induce AM expansion and reprogramming, reinforcing the role of DATP-derived EGF signals in launching niche remodelling cascades (Extended Data Fig. 7f–n).

We next asked whether blocking mutant AT2–fibroblast interactions would alter mutant epithelial states. Gefitinib treatment of RFP⁺ mutant organoids co-cultured with fibroblasts reduced Sox9⁺ DATP-like populations and increased Lpcat1⁺ AT2 cells, demonstrating that EGFR-driven fibroblast inputs sustain epithelial reprogramming (Extended Data Fig. 7o–s). Notably, organoid size seemed largely unchanged, suggesting that EGFR inhibition uncouples epithelial identity from proliferation and highlights the primarily instructive role of reprogrammed fibroblasts. By contrast, gefitinib treatment had no direct effect on RFP⁺ mutant organoids cultured without fibroblasts, confirming that fibroblast-derived signals are required to maintain DATP-like states (Extended Data Fig. 7t–v). Consistently, reprogrammed fibroblasts revealed upregulated AT2 regulatory factors, including *Wnt5a*, *Igf1* and *Spp1* (Extended Data Fig. 5i,j).

To investigate whether fibrotic niche maintenance depends on mutant DATP-like cells, we treated *Red2Kras* mice with the *Kras*^G12D-specific inhibitor MRTX1133 for 10 days from 4 weeks post-induction[39,40] (Fig. 3h). MRTX1133 caused a pronounced reduction in Sox9⁺ DATP-like and CD177⁺ reprogrammed populations while increasing AT1 cells and

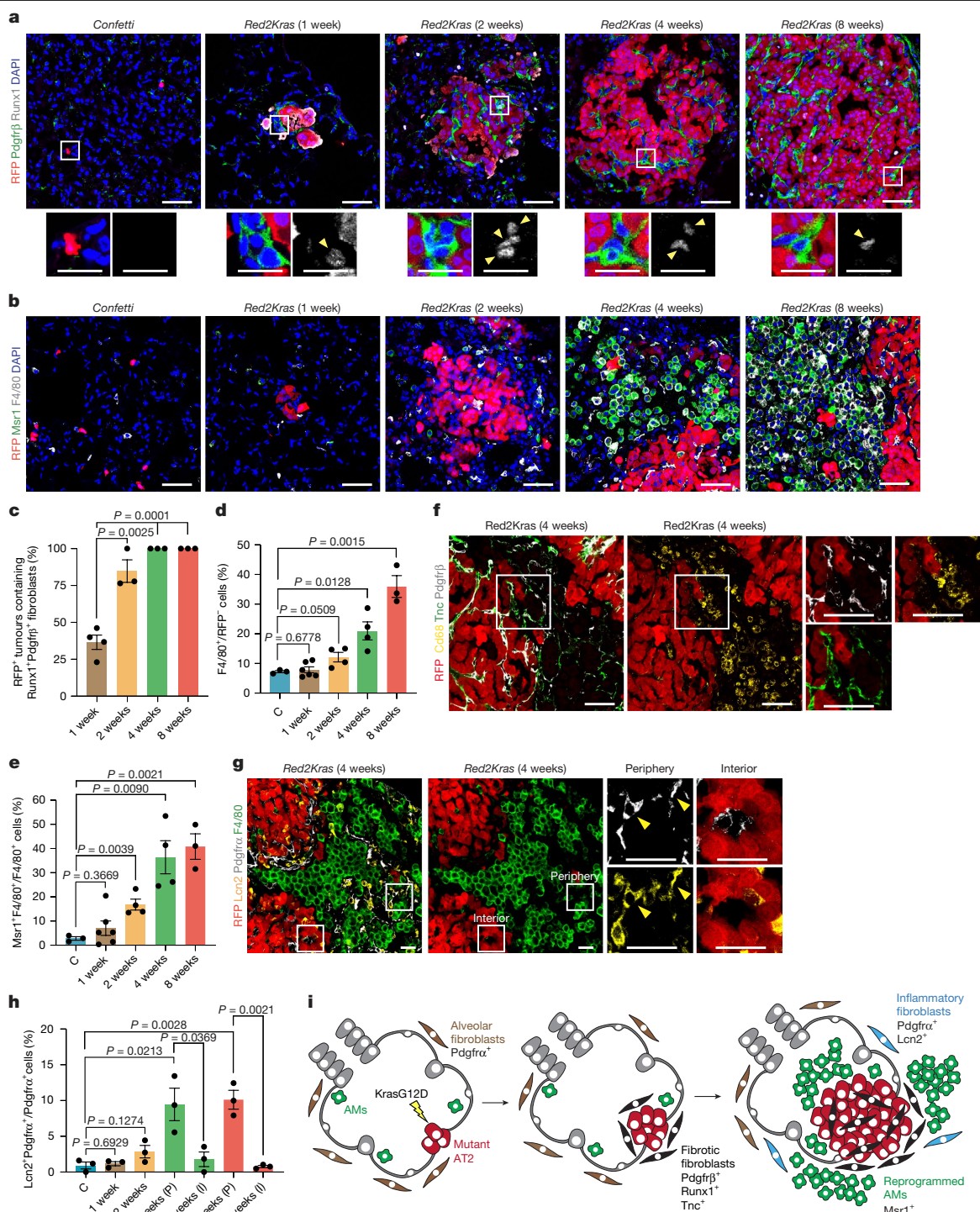

**Fig. 2 | Stromal crosstalk orchestrates spatiotemporal niche evolution during tumour initiation. a,b**, Representative confocal images of lineage-labelled AT2 cells with reprogrammed fibroblasts (**a**) and macrophages (**b**) in *Confetti* and *Red2Kras* lungs. DAPI, blue; Pdgfrβ, green; Runx1, grey and RFP, red (**a**). DAPI, blue; Msr1, green; F4/80, grey and RFP, red (**b**). **c**, Percentage of RFP⁺ tumours containing fibrotic fibroblasts. Data are presented as mean ± s.e.m. Each dot represents one mouse. 1 week, *n* = 4; 2 weeks, *n* = 3; 4 weeks, *n* = 3; 8 weeks, *n* = 3. *P* values were calculated using two-tailed unpaired *t*-test. **d,e**, Quantification of F4/80⁺ macrophages (**d**) and Msr1⁺ macrophages (**e**) within RFP⁻ stromal regions. Data are presented as mean ± s.e.m. Each dot represents one mouse. *Confetti*, *n* = 3; 1 week, *n* = 6; 2 weeks, *n* = 4; 4 weeks, *n* = 4; 8 weeks, *n* = 3. *P* values were calculated using two-tailed unpaired *t*-test. **f**, Representative confocal images showing close interactions between macrophages and reprogrammed fibrotic fibroblasts adjacent to lineage-labelled mutant AT2

cells at 4 weeks post-induction. Pdgfrβ, grey; Tnc, green; CD68, yellow and RFP, red. Images representative of *n* = 3 mice. **g**, Representative confocal images showing close interactions between inflammatory fibroblasts and macrophages adjacent to lineage-labelled mutant AT2 cells 4 weeks post-induction. Lcn2, yellow; F4/80, green; Pdgfrα, grey and RFP, red. **h**, Quantification of inflammatory fibroblasts. Peripheral and interior tumour regions were separately analysed at 4 weeks and 8 weeks. Data are presented as mean ± s.e.m. Each dot represents one mouse; *n* = 3 mice. *P* values were calculated using two-tailed unpaired *t*-test. **i**, Schematic illustrating mesenchymal–immune niche remodelling. At tumour onset, Pdgfrα⁺ fibroblasts adjacent to tumour cells reprogramme into fibrotic fibroblasts (Pdgfrβ⁺Runx1⁺Tnc⁺), which expand at tumour borders and remodel Msr1⁺ AMs, amplifying inflammatory cues and inducing Lcn2⁺ fibroblasts at the tumour periphery. I, interior; P, peripheral. Scale bars, 50 μm (**a**,**b**,**f**), 25 μm (**g**), 20 μm (**a**, magnified panels).

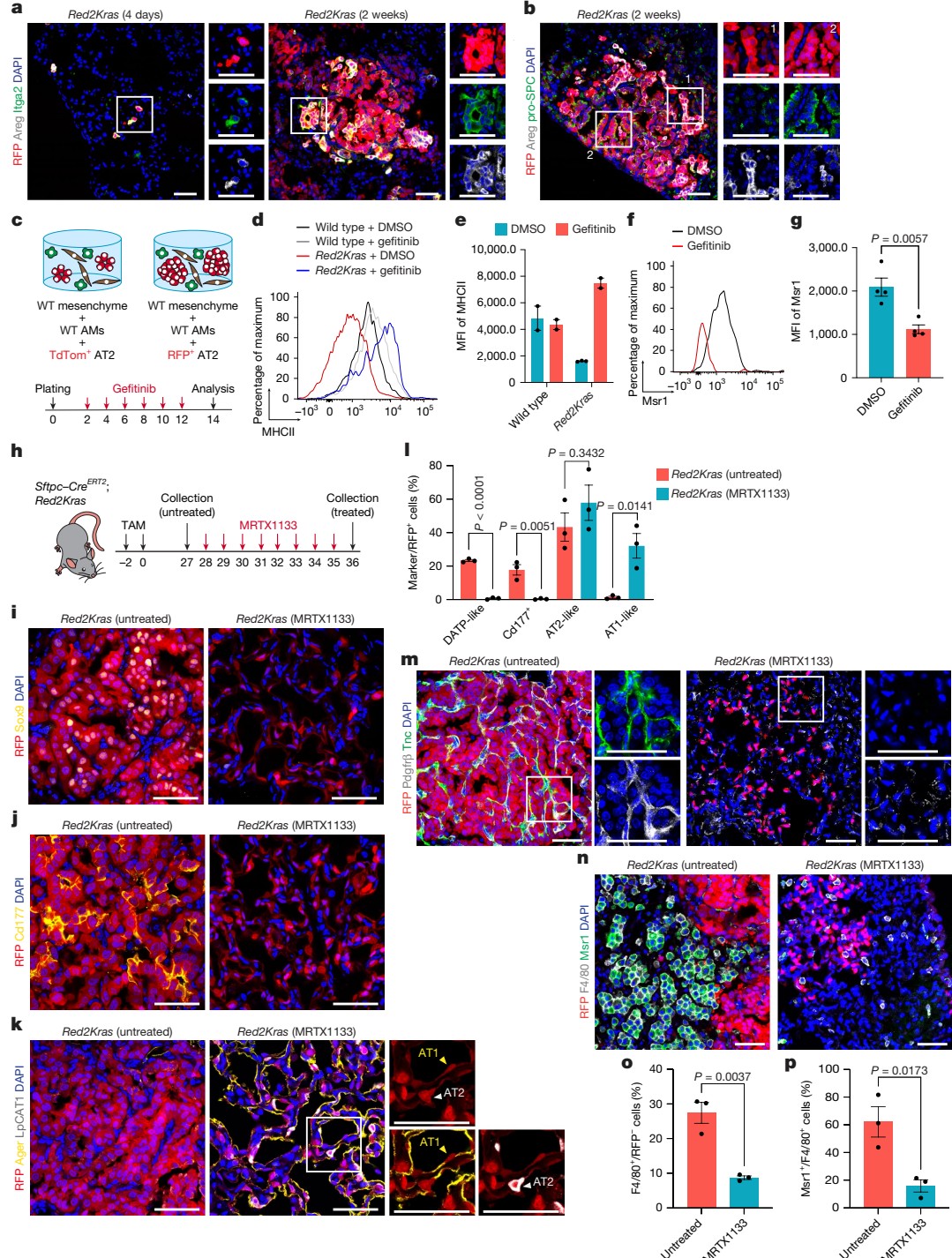

**Fig. 3 | Nascent tumour cells drive niche remodelling to sustain mutant cell states. a,b**, Representative confocal images showing Areg expression in DATP-like cells in *Red2Kras* lungs. DAPI, blue; Areg, grey; Itgα2, green and RFP, red (**a**). DAPI, blue; Areg, grey; pro-SPC, green and RFP, red (**b**). *n* = 2 mice. **c**, Schematic of 3D organoid co-cultures of lineage-labelled AT2 cells with wild-type AMs and mesenchyme, treated with dimethyl sulfoxide (DMSO) or gefitinib. **d,e**, Flow cytometry (**d**) and quantification (**e**) of MHC-II expression in AMs from **c**. Data are presented as mean ± s.e.m. Each dot represents an independent experiment. Wild-type DMSO, *n* = 2; wild-type gefitinib, *n* = 2; *Red2Kras* DMSO, *n* = 3; *Red2Kras* gefitinib, *n* = 2. **f,g**, Flow cytometry (**f**) and quantification (**g**) of Msr1 expression in AMs co-cultured with RFP⁺ mutant cells and wild-type mesenchyme, treated with DMSO or gefitinib. Data are presented as mean ± s.e.m. Each dot represents an independent experiment. DMSO, *n* = 4; gefitinib, *n* = 4. *P* values were calculated using two-tailed unpaired *t*-test.

**h**, Experimental design for Kras^G12D inhibitor (MRTX1133) administration. **i–k**, Representative confocal images showing lineage-labelled mutant cells. DAPI, blue; Sox9, yellow and RFP, red (**i**). DAPI, blue; Cd177, yellow and RFP, red (**j**). DAPI, blue; Ager, yellow; LpCAT1, grey and RFP, red (**k**). **l**, Quantification of cell states within RFP⁺ mutant cells from **i–k**. Data are presented as mean ± s.e.m. Each dot represents one mouse. Untreated, *n* = 3; MRTX1133, *n* = 3. *P* values were calculated using two-tailed unpaired *t*-test. **m,n**, Representative confocal images showing lineage-labelled cells with reprogrammed fibroblasts (**m**) and macrophages (**n**). DAPI, blue; Pdgfrβ, grey; Tnc, green and RFP, red (**m**). DAPI, blue; F4/80, grey; Msr1, green and RFP, red (**n**). Scale bar, 50 μm. **o,p**, Quantification of macrophages (**o**) and Msr1⁺ macrophages (**p**) from **n**. Data are presented as mean ± s.e.m. Each dot represents one mouse. Untreated, *n* = 3; MRTX1133, *n* = 3. *P* values were calculated using two-tailed unpaired *t*-test. Scale bars, 50 μm.

restoring an AT2:AT1 ratio of approximately 2:1, similar to homeostasis[41] (Fig. 3i–l). Targeting DATP-like states also reversed niche remodelling, with a marked loss of Pdgfrβ⁺Tnc⁺ fibrotic fibroblasts and decreased AM expansion and reprogramming (Fig. 3m–p). These findings demonstrate that tumour fibroblasts maintain a reversible, injury-like fibrotic phenotype dependent on continuous signals from reprogrammed mutant cells[16].

Collectively, our results demonstrate that DATP-like states in nascent *Kras*^*G12D*-mutant AT2 cells act as central signalling hubs that coordinate fibrotic and immune niche remodelling through Areg-driven EGFR activation of fibroblasts. Disrupting this signalling circuit reverses early niche reprogramming, underscoring a therapeutic window for intercepting tumour-permissive niche formation as its onset.

## Areg–EGFR drives tumour–niche assembly

To determine the requirement for fibrotic fibroblasts during early niche formation in vivo, we orthotopically engrafted RFP⁺ tumour organoids into *Pdgfra–Cre*^*ERT2*;*ZsGreen*;*DTR* mice, enabling lineage tracing and selective depletion of resident *Pdgfra*⁺ fibroblasts following intratracheal diphtheria toxin administration (Extended Data Fig. 8a). One week after engraftment, diphtheria toxin was administered every other day for 14 days, and tissues were analysed 19 days post-injection. In controls, lineage-labelled fibroblasts localized to RFP⁺ tumours and upregulated Pdgfrβ, confirming tumour-induced fibrotic reprogramming of resident *Pdgfra*⁺ fibroblasts (Extended Data Fig. 8b). Diphtheria-toxin-mediated depletion markedly reduced expansion of lineage-labelled Pdgfrβ⁺ fibroblasts, resulting in impeded tumour growth, diminished Sox9⁺ reprogrammed mutant cells and reduced macrophage expansion (Extended Data Fig. 8b–d). These data demonstrate that fibrotic fibroblasts that originated from resident alveolar fibroblasts are critical for early tumour development and immune niche establishment.

We next evaluated whether interrupting the Areg–EGFR axis disrupts tumour-induced niche remodelling in vivo. Pharmacologic EGFR inhibition with gefitinib in *Red2Kras* mice reduced tumour burden, with reduced Tnc⁺ fibrotic fibroblasts, decreased macrophage activation and impaired neutrophil recruitment (Extended Data Fig. 8e–n). Gefitinib also decreased Sox9⁺ DATP-like and CD177⁺ mutant states while increasing Lpcat1⁺ AT2 cells, indicating that EGFR-dependent fibroblast signalling is required to sustain both mutant epithelial reprogramming and immune remodelling (Extended Data Fig. 8o–r). To pinpoint the contribution of mutant-cell-derived Areg in vivo, we genetically deleted *Areg* in AT2 cells (*Areg*^*flox/flox*;*Sftpc–Cre*^*ERT2*;*Red2Kras*) (Fig. 4a). At 2 weeks post-induction, *Areg*-depleted lungs revealed reduced tumour formation compared with haplodeficient controls (Fig. 4b,c). Single-cell profiling on niche compartments (1:1 mixture of CD31⁻CD45⁻EpCAM⁻ mesenchymal and CD31⁻CD45⁺EpCAM⁻ immune cells) and lineage-labelled RFP⁺ mutant cells from *Areg*-deficient and haplodeficient lungs identified 8,206 mesenchymal cells comprising established stromal populations (Extended Data Fig. 9a–e). Subclustering fibroblasts revealed a reprogrammed subset expressing *Runx1*, *Acta2*, *Tnc* and *Pdgfrβ* (Fig. 4d and Extended Data Fig. 9f). Consistent with gefitinib treatment, *Areg*-deficient lungs showed reduced reprogrammed fibroblasts, confirmed by decreased Tnc⁺Pdgfrβ⁺ fibrotic fibroblasts, establishing mutant-derived Areg as a critical initiating signal for fibrotic niche remodelling (Fig. 4e–g).

Analysis of 7,731 immune cells identified 14 immune clusters (Extended Data Fig. 10a,c). In *Areg*^*flox/+* lungs, AMs and neutrophils predominated, whereas Areg deletion reduced immune remodelling (Extended Data Fig. 10b). AMs from *Areg*^*flox/+* lungs showed elevated pro-inflammatory signatures (*Cxcl2*, *Ccl6* and *Ccl9*) and reduced MHC-II complex genes compared with *Areg*^*flox/flox* lungs (Extended Data Fig. 10d,e). Immunofluorescence confirmed reduced Msr1⁺ macrophages in *Areg*-deleted lungs (Fig. 4h–j). Reclustering neutrophils

revealed Areg-dependent transcriptional reshaping (Extended Data Fig. 10f,g). Neutrophils from *Areg*^*flox/+* lungs upregulated *SiglecF*, *Tlr4*, *Clec5a* and *Cd177*, implicated in neutrophil activation and LUAD progression[42–44]. These data confirm that blocking EGFR-mediated fibrotic reprogramming prevents immune niche remodelling required for tumour formation in vivo.

Finally, profiling of 12,219 RFP⁺ mutant cells recovered previously defined clusters[7] (Fig. 4k,l). *Areg* deletion in *Kras*^*G12D*-mutant AT2 cells significantly reduced DATP-like and *CD177*⁺ reprogrammed populations while increasing *Sftpc*⁺ AT2 cells compared with controls (Fig. 4m).

Collectively, our findings elucidate a hierarchical signalling cascade, in which oncogenic AT2 reprogramming creates Areg-secreting DATP-like states that activate EGFR on adjacent fibroblasts, initiating fibrotic niche assembly. These remodelled fibroblasts feedback to sustain mutant epithelial plasticity and drive immune remodelling. Disrupting this circuit halts stromal and immune reprogramming, establishing Areg–EGFR signalling as a central regulator of early tumour-permissive niche formation and plasticity (Fig. 4n).

## Conserved niche circuits in human LUAD

Single-cell studies have uncovered early epithelial reprogramming in tissues of patients with LUAD[5]; however, direct evidence linking tumour–niche interactions to early human LUAD progression remains limited. To address this, we analysed published single-cell transcriptomic data from early-stage LUAD and matched normal lung tissues[45] (six pairs total; five matched pairs for analysis; Extended Data Fig. 11a). Given the mutual exclusivity of KRAS and EGFR mutations in LUAD[46,47], we focused on EGFR wild-type cases to enrich for KRAS-driven events (Extended Data Fig. 11a). Subclustering EpCAM⁺ epithelial cells identified AT2, AT1 and five tumour-specific populations (Extended Data Fig. 11b–e). Two clusters (1 and 2) showed high expression of DATP markers (*CLDN4*, *KRT8* and *KRT19*), consistent with regenerative transitional states in early tumorigenesis[5,21] (Extended Data Fig. 11f). Although cluster 2 was enriched in a single patient, cluster 1 was consistently detected across samples, including in stage I LUAD tissues, suggesting a conserved DATP-like state across early LUAD (Extended Data Fig. 11g,h). This conserved cluster exhibited high *AREG* and *EREG* expression, indicating upregulation of EGFR ligands during early tumour evolution (Extended Data Fig. 11i).

To investigate stromal remodelling, we reclustered *COL1A1*⁺ mesenchymal cells and subclustered fibroblasts expressing *PDGFRα*, *COL14A1* and *COL13A1* (Extended Data Fig. 11k). A LUAD-enriched fibroblast subset (fibroblasts_6) expresses fibrotic markers (*RUNX1*, *PDGFRβ*, *ACTA2* and *CTHRC1*), along with elevated ECM components, mirroring fibrotic reprogramming observed in *Red2Kras* lungs (Extended Data Fig. 11l–p). Immunofluorescence in early-stage *KRAS*^*G12D* LUAD specimens confirmed KRT8⁺SOX9⁺ DATP-like cells with high AREG expression (Fig. 5a and Extended Data Fig. 11j). Fibrotic fibroblasts expressing ACTA2, RUNX1 and CTHRC1 were located within tumours and frequently abutted KRT8⁺ tumour cells, indicating spatially coordinated epithelial–fibroblast interactions in human LUAD (Fig. 5b and Extended Data Fig. 11q).

To functionally model sequential transitions in tumour and niche interactions, we developed an ex vivo three-dimensional (3D) inducible human LUAD system by introducing doxycycline-inducible *KRAS*^*G12D* and an RFP reporter in primary human AT2 (hAT2) cells (Fig. 5c). EpCAM⁺HTII-280⁺ hAT2 cells from non-tumour lung parenchyma were expanded as organoids, transduced at day 14 and replated as purified RFP⁺ *KRAS*^*G12D*–hAT2 organoids (Fig. 5d,e)[48]. Single-cell profiling showed that control organoids consisted largely of *SFTPC*⁺ AT2 cells, whereas *KRAS* activation shifted cells into transitional states with reduced *SFTPC* expression, paralleling *Red2Kras* findings (Fig. 5f–h). *KRAS*^*G12D*–hAT2 organoids also retained *SFTPC*⁺*SCGB3A2*⁺ and *SFTPC*⁻*SCGB3A2*⁺*SCGB1a1*⁻ subsets, corresponding to AT0 and terminal and

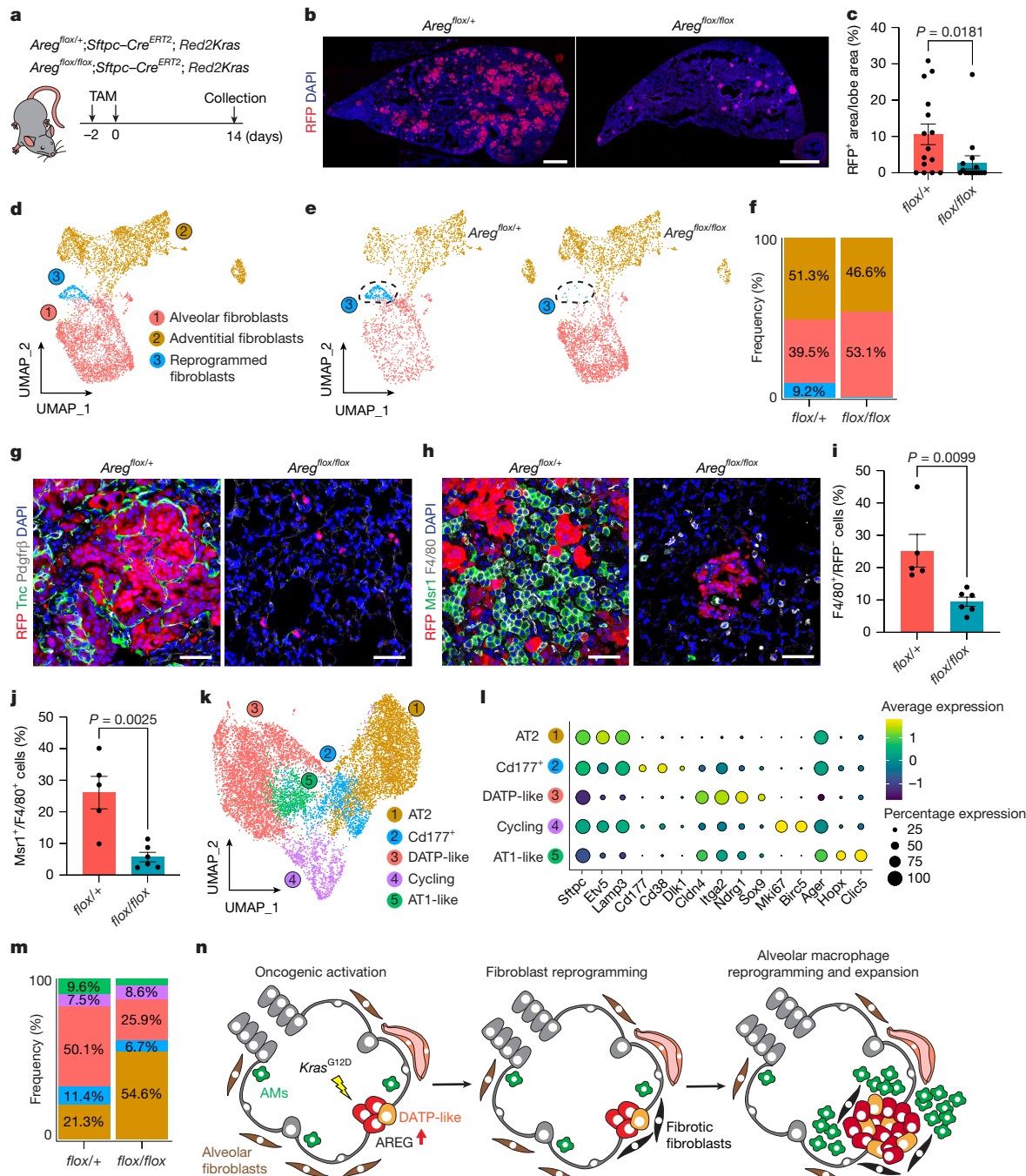

**Fig. 4 | Areg–EGFR axis establishes precancer niches essential for tumour development. a**, Experimental design for AT2-cell-specific genetic deletion of *Areg* in *Red2Kras* lungs. **b**, Representative whole-lobe tile scans of *Areg*^flox/+ and *Areg*^flox/flox lungs. DAPI, blue and RFP, red. **c**, Percentage of RFP^+ cell area relative to lobe area. Data are presented as mean ± s.e.m. Each dot represents one mouse. *Areg*^flox/+, n = 15; *Areg*^flox/flox, n = 15. Mice with no detectable RFP^+ expansion were assigned a value of 0. *P* values were calculated using two-tailed Mann–Whitney test. **d**, UMAP of fibroblast subclusters. **e**, UMAP showing the distribution of *Areg*^flox/+ or *Areg*^flox/flox fibroblasts. **f**, Percentage of cells distributed across each cluster annotated in **d**. **g**,**h**, Representative confocal images of reprogrammed fibroblasts (**g**) and macrophages (**h**) with lineage-labelled cells in *Areg*^flox/+ and *Areg*^flox/flox lungs. DAPI, blue; Pdgfrβ, grey; Tnc, green and RFP, red (**g**). DAPI, blue;

F4/80, grey; Msr1, green and RFP, red (**h**). Images representative of n = 3 mice. **i**, Quantification of F4/80^+ macrophages within the RFP^- stromal cells assessed in **h**. Data are presented as mean ± s.e.m. Each dot represents one mouse. *Areg*^flox/+, n = 5; *Areg*^flox/flox, n = 6. *P* values were calculated using two-tailed unpaired *t*-test. **j**, Percentage of Msr1^+ macrophages assessed in **h**. Data are presented as mean ± s.e.m. Each dot represents one mouse. *Areg*^flox/+, n = 5; *Areg*^flox/flox, n = 6. *P* values were calculated using two-tailed unpaired *t*-test. **k**, UMAP of lineage-labelled RFP^+ cells from *Areg*^flox/+ and *Areg*^flox/flox lungs. **l**, Dot plot of epithelial state marker genes in *Areg*^flox/+ and *Areg*^flox/flox lungs. **m**, Percentage of RFP^+ cells distributed across each cluster defined in **k**. **n**, Schematic illustrating the sequential events establishing precancer niches. Scale bars, 1,000 μm (**b**), 50 μm (**g**,**h**).

respiratory bronchiole secretory cells, previously shown to originate from AT2 cells (Fig. 5f–h)[49,50]. Notably, we identified an *AREG*^high populations co-expressing *SOX9*, *KRT8* and *ITGA2*, indicative of DATP-like states (Fig. 5f–h). Co-culture of *KRAS*^G12D–hAT2 organoids with primary

human lung mesenchymal cells (CD31^-CD45^-EpCAM^-) isolated from non-tumour lung parenchyma induced fibrotic phenotypes marked by PDGFRβ, which were fully abrogated by gefitinib treatment (Fig. 5i). These findings demonstrate that *KRAS*^G12D-driven AT2 reprogramming

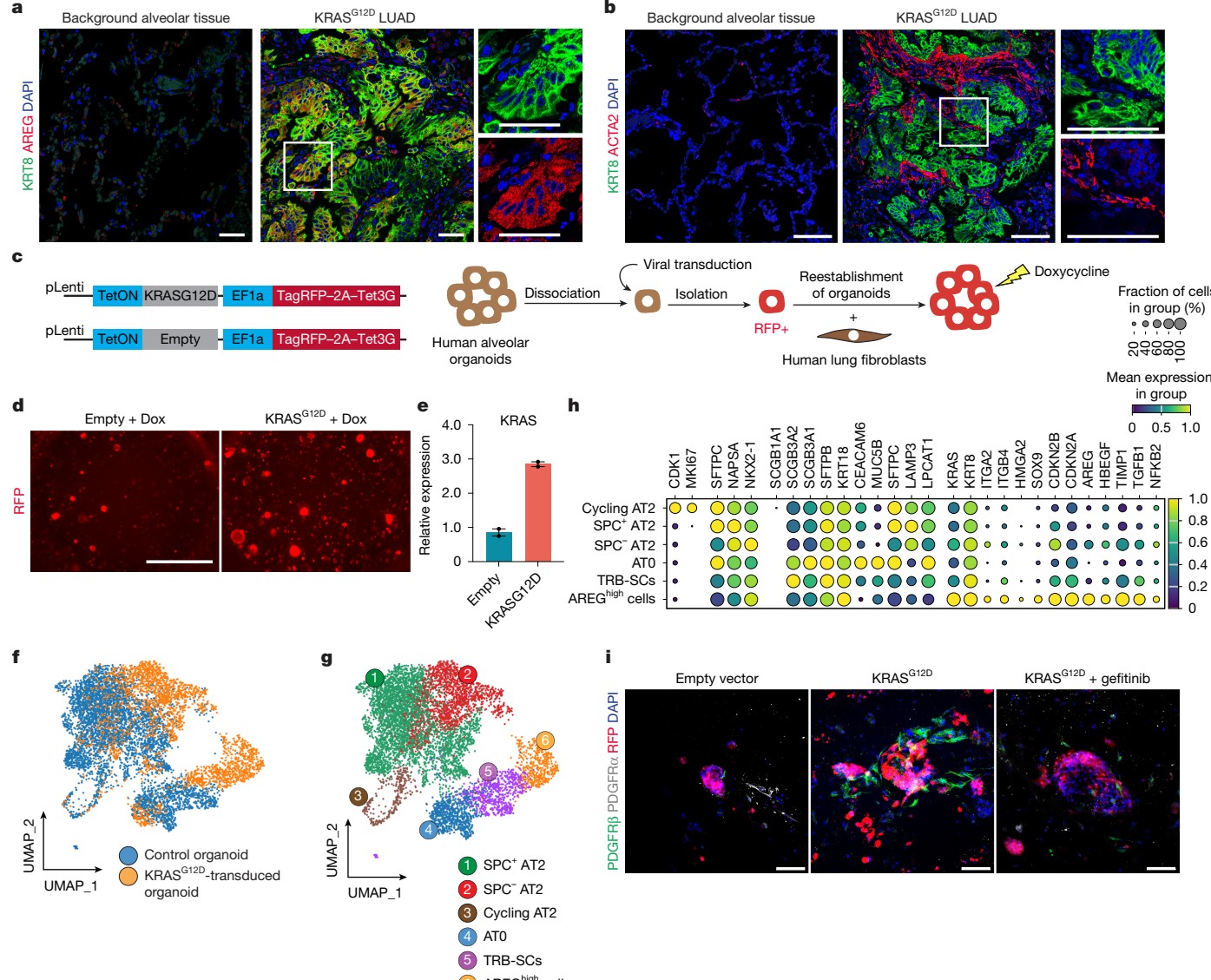

**Fig. 5 | Oncogenic activation of human AT2 cells drives fibrotic niches.**
**a,b**, Representative confocal images showing fibrotic fibroblasts and DATP-like AREG-expressing cells in matched background and early-stage *KRAS^{G12D}* tissues from patients with LUAD. DAPI, blue; KRT8, green; AREG, red (**a**). DAPI, blue; ACTA2, red and KRT8, green (**b**). **c**, Schematics of inducible viral constructs and experimental design for establishing inducible *KRAS^{G12D}* human alveolar organoid. **d**, Representative fluorescent images of alveolar organoids transduced with lentivirus vectors expressing empty or *KRAS^{G12D}* (**d**). **e**, Quantitative polymerase chain reaction analysis showing *KRAS* expression upon doxycycline treatment in *KRAS^{G12D}* and empty-vector-transduced cells. Data are presented

as mean ± s.e.m. Each dot represents an independent experiment. Empty, *n* = 2; *KRAS^{G12D}*, *n* = 2. **f**, UMAP of cells derived from control or *KRAS^{G12D}* organoids. Cells are coloured by dataset of origin. **g**, UMAP showing epithelial cell state diversification following *KRAS^{G12D}* induction. **h**, Dot pot of key marker genes defining epithelial populations in **g**. **i**, Representative confocal images showing PDGFRβ^{+} fibroblasts around *KRAS^{G12D}*-expressing organoids and suppression of fibrotic remodelling upon EGFR inhibition. RFP, tumour organoid; PDGFRβ, green; PDGFRα, grey and DAPI, blue. Images representative of *n* = 3 experiments. Scale bars, 50 μm (**a**), 2,000 μm (**d**), 100 μm (**b,i**).

into DATP-like states is conserved across mouse and human lungs and is sufficient to activate fibroblasts through the AREG–EGFR axis in early tumorigenesis.

To test whether this mechanism extends beyond KRAS, we developed a 3D inducible EGFR-mutant LUAD model by expressing *EGFR^{L858R}* with an EGFP reporter in primary mouse AT2 cells (Extended Data Fig. 12a). Lineage-labelled AT2 cells were cultured as organoids, transduced at day 14 and orthotopically engrafted into NSG mouse lungs (Extended Data Fig. 12a,b). After 3 weeks, tdTomato^{+}EGFP^{+} mutant cells formed expanding lesions enriched for Krt8 and Areg, hallmarks of DATP-like states observed in *Kras^{G12D}* models (Extended Data Fig. 12c,d). *EGFR^{L858R}*-mutant cells induced Pdgfrβ^{+} fibrotic fibroblasts and macrophage expansion in adjacent niches (Extended Data Fig. 12e).

These results demonstrate that oncogenic KRAS and EGFR mutations reprogramme AT2 cells into Areg^{+} DATP-like states that activate EGFR in surrounding fibroblasts, driving fibrotic and immune niche assembly. AREG–EGFR signalling thus represents a conserved central mechanism of early tumour–microenvironment co-evolution across LUAD subtypes.

## Discussion

Tissue homeostasis relies on tightly regulated stem cell–stromal interactions[51]. Tumour initiation disrupts these networks, triggering spatial and temporal microenvironmental remodelling. Although cancer hijacks regenerative programmes, the earliest steps converting normal

tissue into a tumour-permissive niche remain unresolved. Here we define these transitions at lung tumour onset. *Kras*[G12D]-mutant AT2 cells rapidly adopt a regenerative, Areg[high] DATP-like state that functions as a central signalling hub. Through Areg–EGFR activation, these cells induce fibrotic fibroblasts that remodel the ECM and reprogramme AMs through the Tnc–TLR4 axis, establishing a self-reinforcing fibrotic–immune niche that sustains epithelial plasticity and accelerates tumour expansion. *EGFR*[L858R]-mutant AT2 cells engage a convergent epithelial–fibrosis circuit, identifying regenerative epithelial states as a conserved early driver of niche construction across LUAD subtypes.

We demonstrate that oncogenic signals co-opt regenerative pathways to generate mutant epithelial states and spatially distinct fibroblast populations. At tumour onset, DATP-like cells induce fibrotic fibroblasts that activate injury-like ECM remodelling, establishing the matrix required for tumour initiation[52]. The Areg–EGFR axis emerges as an early determinant of this epithelial–mesenchymal crosstalk. Notably, this circuit is reversible. Kras inhibition reduces DATP-like and CD177[+] mutant states, restores the AT2–AT1 balance and reverses niche remodelling, whereas genetic or pharmacologic blockage of Areg–EGFR signalling prevents fibroblast reprogramming, extinguishes reprogrammed mutant states and limits immune remodelling. These findings reveal reciprocal dependency between DATP-like cells and their niches, identifying Areg–EGFR signalling as a therapeutically actionable vulnerability at the preneoplastic stage, consistent with previous clonal analyses showing that mutant AT2 expansion requires sustained niche support[7].

Unlike injury repair in which inflammatory fibroblasts precede fibrotic differentiation, Lcn2[+] inflammatory fibroblasts emerge later in tumour development, coinciding with macrophage remodelling[16,17,19]. Probably induced by IL-1β from reprogrammed AMs, they localize to tumour peripheries and lack fibrotic markers such as Tnc, establishing spatial hierarchy in which fibrotic cues dominate tumour cores, whereas inflammatory programmes persist at the periphery. This compartmentalization mirrors fibroblast hierarchies in repair but diverges in timing, as inflammatory fibroblasts are rare at tumour onset[17]. Early fibrotic fibroblasts directly modulate mutant epithelial states, whereas inflammatory fibroblasts, together with AMs, shape the immune milieu later by upregulating chemoattractants such as *Cxcl12*, a feature absent in reprogrammed AMs, indicating distinct recruitment mechanisms[17]. Notably, advanced LUAD harbours heterogeneous cancer-associated fibroblasts, including p16[+] ApoE-secreting fibroblasts[53,54], which are not detected in early lesions, indicating temporally distinct fibroblast programmes, with early Areg-dependent fibrotic states remaining plastic and reversible (Extended Data Fig. 1f). Our findings suggest that later cancer-associated fibroblast complexity arises from initial epithelial–stromal interactions.

Immune remodelling occurs downstream of fibroblast activation. Resident AMs undergo phenotypic rewiring, acquiring hybrid inflammatory profiles and reduced MHC-II expression resembling tumour-associated macrophages in solid tumours[55]. These AMs derive primarily from resident AMs and are remodelled by fibrotic fibroblasts through the Tnc–TLR4 axis, establishing a sequential stromal–immune signalling cascade. AM depletion impairs tumour growth and prevents neutrophil and γδ T cell recruitment, demonstrating that AM remodelling is essential for assembling immunosuppressive niches. Spatial segregation further defines immune roles. AMs accumulate peri-tumourally, whereas neutrophils and γδ T cells infiltrate tumour cores[33,34]. Although reprogrammed AMs facilitate tumour development through niche remodelling, neutrophils expressing high IL-1β probably provide dominant inflammatory cues sustaining tumour reprogramming alongside fibrotic fibroblasts, consistent with restricted tumour development in *IL1R1*-deficient *Kras*[G12D] AT2 cells[7]. Collectively, immune remodelling represents a critical downstream consequence of fibroblast activation in preneoplastic niche assembly.

Our findings revealed both parallels and distinctions between injury repair and oncogenesis. In both contexts, fibrotic fibroblasts support

AT2 cell expansion and reprogramming. However, during regeneration, they resolve as DATP states differentiate, whereas in oncogenesis sustained NF-κB activation maintains DATP-like mutant cells, creating a self-reinforcing loop[7,16]. This pathological circuit echoes persistent fibrosis in LUAD and idiopathic pulmonary fibrosis. Early-stage human LUAD confirms conservation of this epithelial–fibrotic interactions, and similar EGFR-dependent epithelial–fibroblast circuits have been implicated in idiopathic pulmonary fibrosis[56]. The Areg–EGFR signalling thus represents the apex of a hierarchical cascade orchestrating tumour–niche formation across fibrotic lung pathologies.

To overcome limitations in modelling early tumour–niche interactions in humans, we established an inducible LUAD organoid platform enabling temporal KRAS activation and sequential mutant reprogramming in primary hAT2 cells. This system captures the transition from normal to pre-malignant states, recapitulating emergence of AREG[high] DATP-like cells and de-differentiation into AT0 or terminal and respiratory bronchiole secretory cell populations, indicating redeployment of regeneration-associated transitional states during early LUAD[49,50]. *KRAS*[G12D]-expressing hAT2 cells induce fibrotic reprogramming of human lung mesenchyme in an EGFR-dependent manner, recapitulating early-stage *KRAS*[G12D] LUAD architecture. This platform enables dissection of epithelial–stromal signalling and testing of preventive interventions. Notably, clinical responses to EGFR tyrosine kinase inhibitors in EGFR wild-type non-small cell lung cancer, particularly in AREG[high] tumours, suggest that such EGFR-dependent states may already be exploited therapeutically[57].

In summary, we map the dynamic interplay between mutant epithelial cells and their microenvironment during early lung tumorigenesis. Oncogenic AT2 reprogramming activates an Areg-driven EGFR axis that initiates sequential fibrotic and immune niche assembly. The reversibility of these preneoplastic circuits defines a therapeutic window before progression to treatment-resistant disease. Our inducible human LUAD platform provides a framework to interrogate patient-specific tumour–niche interactions and target conserved EGFR-dependent mechanisms of tumour ecosystem formation.

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

# Methods

## Mice

*Sftpc–Cre^ERT2* (028054), *R26R–Confetti* (013731), *Pdgfrα–Cre^ERT2* (032770), *R26R–iDTR* (007900), *NOD/Scid Il2rg null Tg* (NSG: 005557) and *Ai6/RCL–ZsGreen* (007906) animals were obtained from The Jackson Laboratory. *Areg^loxP/loxP* animals were kindly provided by M. Clatworthy from the University of Cambridge. *Red2Kras* mice were generated in-house as previously described[22]. *CCR2–Cre^ERT2* mice were kindly provided by B. Becher (University of Zurich)[58]. All transgenic mouse strains were maintained on a C57BL or C57BL/6Brd-Tyr 597 c-Brd mixed background. Mouse studies in the UK were approved under UK Home Office Project Licences PC7F8AE82 and PP3176550, and experiments in the US and Korea were approved by the Memorial Sloan Kettering Cancer Center (MSKCC) Institutional Animal Care and Use Committee (protocol no. 24-04-003) and Gwangju Institute of Science and Technology (GIST) Institutional Animal Care and Use Committee (protocol no. GIST-2022-043). All procedures complied with institutional and national guidelines. The mice were housed under specific pathogen-free conditions at the Gurdon Institute (University of Cambridge), MSKCC and GIST on a 12-h light/dark cycle with food and water provided ad libitum.

Both male and female mice aged 6–15 weeks were used. Experiments were randomized where feasible. Blinding was not performed, as treatment effects on tumour volume were readily distinguishable between groups. Humane end points were defined as a single tumour exceeding 2 cm in diameter, a tumour burden exceeding 10% of body mass or multiple tumours with a cumulative volume greater than 3,000 mm³. As this study focused on early tumour development, these limits were not approached or exceeded in any experiment.

## Mouse procedures

**Tamoxifen administration.** Tamoxifen (Sigma; T5648) was dissolved in corn oil (Sigma; C8267) at 20 mg ml⁻¹. Aliquots were heated to 50 °C and vortexed before administration. Animals were weighed and received tamoxifen by oral gavage. They received either a single dose (0.1 mg g⁻¹ body weight) for clonal analysis or two to four doses (0.2 mg g⁻¹ body weight) administered every other day. Tissue collection time points are specified in the relevant figures and detailed in Methods.

**MRTX1133 administration.** *Red2Kras* animals received two doses of tamoxifen through oral gavage (0.2 mg g⁻¹ body weight) every other day to induce *Kras^G12D* expression. At 4 weeks post-induction, the mice received freshly prepared MRTX1133 (MedChemExpress; HY-134813) through intraperitoneal injection at 15 mg kg⁻¹ twice daily for 10 days. The stock solution was prepared in DMSO and diluted in 40% polyethylene glycol 300 (PEG300; MedChemExpress; HY-Y0873), 5% Tween-80 (MedChemExpress; HY-Y1891) and 45% phosphate-buffered saline (PBS) for injection, as recommended by the manufacturer.

**Gefitinib administration.** *Red2Kras* animals received two doses of tamoxifen through oral gavage (0.2 mg g⁻¹ body weight) every other day to induce *Kras^G12D* expression. Four days after the final tamoxifen dose, the mice received freshly prepared gefitinib (80 mg kg⁻¹ in 50 μl of DMSO) or DMSO (vehicle control) through intraperitoneal injection every 4 days for 20 days.

**Clodronate administration.** *Red2Kras* animals received four doses of tamoxifen through oral gavage (0.2 mg g⁻¹ body weight) every other day to induce *Kras^G12D* expression. For selective depletion of macrophages in the lungs, five doses of PBS-loaded or clodronate-loaded liposomes (5 mg ml⁻¹; LIPOSOMA) were administered through intratracheal injection (25 μl) at the time points depicted in the experiment scheme, beginning 2 days after the final tamoxifen dose. Lungs were collected and analysed 7 days after the final clodronate liposome administration.

**Diphtheria toxin administration.** To deplete fibroblasts, 8–10-week-old *Pdgfrα–Cre^ERT2;ZsGreen;iDTR* mice were used. The animals received four doses of tamoxifen through oral gavage (0.2 mg g⁻¹ body weight) every other day to mark *Pdgfrα^+* cells. Following organoid engraftment (see section "Orthotopic engraftment of *Kras^G12D* organoids"), diphtheria toxin (Sigma) was dissolved in PBS and administered intratracheally at 50 ng per mouse every other day for seven doses, starting 21 days after the final tamoxifen dose. Lungs were collected 7 days after the final diphtheria toxin injection for analysis.

**Orthotopic engraftment of *Kras^G12D* organoids.** RFP⁺ *Kras^G12D*-mutant organoids co-cultured with mesenchymal cells were orthotopically engrafted into *CCR2–Cre^ERT2;ZsGreen* or *Pdgfrα–Cre^ERT2;ZsGreen;iDTR* mice. A total of 50,000 RFP⁺ mutant epithelial cells from one to two passages of organoids were isolated by fluorescence-activated cell sorting (FACS) and mixed with 20,000 freshly isolated lung mesenchymal cells (CD31⁻CD45⁻EpCAM⁻) from wild-type lungs to enhance epithelial cell recovery during engraftment. The epithelial–stromal cell mixture was resuspended in 20 μl of PBS and transplanted intratracheally into recipient mice 1 day after a single dose of bleomycin treatment (1.0 U kg⁻¹). Lungs were collected and analysed 21 days post-engraftment to assess differentiation of the engrafted cells.

**Generation of *EGFR^L858R*-transduced organoids for orthotopic engraftment.** AT2 organoids expressing tdTomato (passages 1 and 2) derived from *Sftpc–Cre^ERT2;tdTomato* lungs were dissociated into single cells and transduced with lentivirus encoding pHAGE–EGFR^L858R–EGFP (Addgene plasmid no. 116276) by spin infection (2,000 rpm; 32 °C; 60 min) in the presence of polybrene (8 μg ml⁻¹; Sigma). Transduced cells were subsequently co-cultured with mesenchymal cells. FACS was used to isolate tdTomato⁺GFP⁺ cells on day 10 post-infection, which were then expanded in co-culture with mesenchymal cells for further passages. Organoids at passages 1 and 2 were used for orthotopic engraftments. A total of 50,000 FACS-isolated epithelial cells were combined with 20,000 freshly isolated lung mesenchymal cells (CD31⁻CD45⁻EpCAM⁻) from wild-type lungs to support epithelial cell survival. The cell mixture was resuspended in 20 μl of PBS and transplanted intratracheally into NSG mice 1 day after a single dose of bleomycin treatment (1.0 U kg⁻¹). Lungs were collected and analysed 21 days post-engraftment to assess differentiation of the transplanted cells.

## Human adult lung tissue

The Royal Papworth Hospital NHS Foundation Trust (Research Tissue Bank Generic Research Ethics Committee approval, Tissue Bank Project no. T02233) provided de-identified LUAD and matched normal background lung tissues obtained from lobectomies. Fresh tissues were either dissociated to establish human alveolar organoids, followed by *KRAS^G12D* introduction for single-cell profiling, or fixed overnight in 4% paraformaldehyde (PFA; Thermo Fisher Scientific; 10131580) at 4 °C and processed into paraffin-embedded sections (7 μm) for immunofluorescence analysis. Paraffin-embedded sections of human LUAD tissues harbouring confirmed *KRAS^G12D* mutations were obtained from MSKCC following surgical lobectomy (Institutional Review Board no. 12-245). For human alveolar organoid co-cultures, de-identified non-tumour lung parenchymal tissues were obtained from lobectomies at Severance Hospital (IRB nos. 4-2019-0447, 4-2012-0685 and 4-2013-0770). Written informed consent was obtained from all donors before tissue collection under approved IRB protocols. No human participants were recruited specifically for this study.

## Tissue collection, fixation and sectioning

All animals were euthanized by cervical dislocation, and lungs were perfused with 10 ml of PBS (Sigma; D8537) to remove blood. Lungs were slowly inflated through intratracheal injection of 2–3 ml of 4% PFA (Thermo Fisher Scientific; 10131580) in PBS, dissected and fixed

in 4% PFA for 2–4 h at 4 °C. Tissues were washed three times in PBS at room temperature for 15–20 min each and then stored in PBS overnight at 4 °C. The lungs were dehydrated sequentially in 15% then 20% sucrose (Sigma; S5016) in PBS at room temperature for 1 h each, followed by immersion in 30% sucrose in PBS overnight at 4 °C. Individual lobes were separated, trimmed into smaller pieces and embedded in cryomolds filled with optimal cutting temperature compound (VWR; 361603E). Moulds were frozen on dry ice and stored at −80 °C. Frozen tissues were sectioned at 15–20 µm thickness using a cryostat, mounted onto glass slides and stored at −80 °C until staining.

## Lung tissue dissociation

For lung cell isolation, mice were euthanized by cervical dislocation, and lungs were perfused with 10 ml of PBS to remove blood. The lungs were inflated through intratracheal instillation with 2–3 ml of dispase solution (Thermo Fisher Scientific; 11553550) through intratracheal injection. For mesenchymal cell isolation, collagenase I (Gibco; 17100017) was added to the dispase solution at 350 U ml$^{-1}$ before inflation. The lungs were carefully dissected from the thoracic cavity and then placed on ice. Individual lobes were separated, transferred to 50-ml tubes and minced into small fragments. Tissue fragments were washed down with 3 ml of PBS. For epithelial cell isolation, 60 µl of 100 mg ml$^{-1}$ of collagenase–dispase solution was added per tube. Samples were incubated in a shaking incubator at 37 °C, 190 rpm, for 45 min. DNase I (7.5 µl of 1% solution; Sigma; D4527) was added during the final 10 min of incubation. Cell suspensions were sequentially filtered through 100-µm and 40-µm strainers and washed with 2 ml of PF10 (10% fetal bovine serum (FBS) in PBS). Samples were centrifuged at 800 rpm for 5 min at 4 °C. Supernatants were removed, and pellets were resuspended in 1 ml of red blood cell lysis buffer (prepared in-house: 150 mM NH$_4$Cl and 10 mM KHCO$_3$ in distilled H$_2$O) for 60 s at room temperature. Lysis was neutralized with 6 ml of Dulbecco's modified Eagle's medium (DMEM)/F12 (Invitrogen; 11330057). To enrich for viable cells, 500 µl of filtered FBS was carefully layered at the bottom of each tube, followed by centrifugation at 800 rpm for 5 min at 4 °C. Final cell pellets were resuspended in PF10 and transferred to 1.5-ml tubes for antibody staining.

## Flow cytometry analysis

Fluorophore-conjugated antibodies were added to each sample according to the cell population being sorted. Antibodies (Supplementary Table 1) were used at a 1:200 dilution in PF10 and incubated for 20–40 min at 4 °C. DAPI was added during the final 10 min of incubation to label dead cells. A small aliquot of each sample was reserved for unstained and single-stained controls. Following incubation, cells were centrifuged, and pellets were resuspended in PF10. Cell suspensions were filtered through a 35-µm cell strainer (VWR; 352235) into polypropylene FACS tubes (Corning; 352063). Samples were sorted using a BD Influx cell sorter equipped with a 100-µm nozzle, and individual cell populations were collected into chilled 1.5-ml tubes containing 500 µl of FBS.

## Primary 3D mouse lung organoid cultures

All established organoids were validated by genotyping and routinely tested for *Mycoplasma* contamination.

**Feeder-free organoid cultures.** At 7–10 days after three doses of tamoxifen (0.2 mg g$^{-1}$ body weight) injection every other day, RFP$^+$ cells were obtained from *Red2Kras* lungs. Organoids from RFP$^+$ labelled cells were established as previously described[7]. Briefly, sorted CD31$^-$CD45$^-$EpCAM$^+$RFP$^+$ lineage-labelled cells were centrifuged at 300$g$ for 10 min at 4 °C and resuspended in Wnt basal medium (Cambridge Stem Cell Institute's Tissue Culture Core Facility) containing advanced DMEM (Thermo Fisher Scientific; 12491023) supplemented with 10 mM HEPES buffer (Invitrogen; 15630080),

1% penicillin–streptomycin and 1% L-glutamine (Cambridge Stem Cell Institute). Cells were counted, centrifuged and resuspended in growth-factor-reduced Matrigel (Matrigel Growth Factor Reduced (GFR); Corning; 356231). A total of 5,000–10,000 cells in 20 µl of Matrigel GFR were plated per well in eight-well LabTek Chamber Slides (Thermo Fisher Scientific; 154534 K). Matrigel was allowed to solidify at 37 °C for 30 min before adding 300 µl complete Wnt medium per well. Complete Wnt medium comprised Wnt basal medium supplemented with 1× B-27 (Thermo Fisher Scientific; 17504044), 100 ng ml$^{-1}$ of recombinant FGF7 (PeproTech; 100-19-100), FGF10 (PeproTech; 100-26-100), Noggin (PeproTech; 250-38), 50 ng ml$^{-1}$ of recombinant EGF (Life Technologies; PMG8043), 1 mM N-acetylcysteine, 10 mM nicotinamide and 2 µM CHIR99021 (Tocris Bioscience; 4423). Organoids were cultured at 37 °C with 5% CO$_2$, and the medium was changed every other day. The rho kinase (ROCK) inhibitor Y-27632 (10 µM; Cambridge Bioscience; SM02-1) was added during the first 48 h of culture. For EGFR inhibition experiments, gefitinib (Selleckchem; S1025) was added from day 2 post-plating at a final concentration of 5 µM and maintained throughout the experiment. DMSO-treated cultures served as controls.

**Organoid co-cultures with lung fibroblasts.** At 7–10 days after three doses of tamoxifen (0.2 mg g$^{-1}$ body weight) injection every other day, RFP$^+$ mutant (CD31$^-$CD45$^-$EpCAM$^+$RFP$^+$) and tdTomato$^+$ AT2 (CD31$^-$CD45$^-$EpCAM$^+$tdTomato$^+$) cells were obtained from *Red2Kras* and *Sftpc–Cre$^{ERT2}$;tdTomato* lungs, respectively. To isolate lineage-labelled fibroblasts (CD31$^-$CD45$^-$EpCAM$^-$ZsGreen$^+$), *Pdgfrα–Cre$^{ERT2}$; ZsGreen* mice received five daily doses of tamoxifen (0.2 mg g$^{-1}$ body weight), and lungs were collected 7 days post-induction. Lung organoid co-cultures were established as previously described[21]. Briefly, freshly sorted epithelial cells and fibroblasts were centrifuged at 300$g$ for 10 min at 4 °C and resuspended in 3D basic medium comprising DMEM/F12 (Gibco; 11330-032) supplemented with 10% FBS and insulin–transferrin–selenium (Corning; 25-800-CR). Cells were counted and combined at a ratio of 7,000–9,000 epithelial cells to 45,000–50,000 fibroblasts per well. Following centrifugation, cell pellets were resuspended in Matrigel GFR. A 30-µl Matrigel–cell mixture was plated per well in eight-well LabTek Chamber Slides for whole-mount staining. Domes were allowed to solidify at 37 °C for 30 min before adding 300 µl of 3D basal medium per well. Cultures were maintained at 37 °C with 5% CO$_2$. The medium was changed every other day. The ROCK inhibitor Y-27632 (10 µM) was included for the first 48 h of culture. For EGFR inhibition experiments, gefitinib (5 µM) was added from day 2 post-plating and maintained until the end of the culture period. DMSO-treated cultures served as controls.

**Organoid tri-cultures with lung mesenchymal cells and alveolar macrophages.** To investigate the effect of gefitinib in organoid co-cultures, RFP$^+$ mutant and tdTomato$^+$ AT2 cells were isolated as described above. AMs (CD45$^+$CD64$^+$SiglecF$^+$) and lung mesenchymal cells (CD31$^-$CD45$^-$EpCAM$^-$) were isolated from wild-type lungs. Freshly sorted cells were centrifuged at 300$g$ for 10 min at 4 °C and resuspended in 3D basic medium. Cells were counted and combined at a ratio of 1:5:10 (5,000 AT2 cells, 30,000 AMs and 50,000 mesenchymal cells per well). The cell mixture was centrifuged, resuspended in 100-µl Matrigel GFR containing 50% 3D basic medium and plated into 24-well Transwell inserts with 0.4-µm pore size. A total of 500 µl of 3D basic medium was added to the lower chamber, and cultures were maintained at 37 °C with 5% CO$_2$. For inhibition of EGFR signalling, gefitinib was added at a final concentration of 5 µM. DMSO-treated cultures served as controls.

To investigate the effect of Areg and Ereg in initiating niche remodelling, tdTomato$^+$ AT2 cells were co-cultured with wild-type mesenchymal cells and AMs as described above. A total of 500 µl 3D basic medium supplemented with granulocyte–macrophage colony-stimulating factor (20 ng ml$^{-1}$; PeproTech) was added to the lower chamber, and cultures

were maintained at 37 °C with 5% $CO_2$. Recombinant Areg (20 ng ml$^{-1}$; PeproTech; 315-36) and/or Ereg (20 ng ml$^{-1}$; PeproTech; 100-04-5) was added to the medium and maintained for 5 days.

**Co-cultures of lung mesenchymal cells and alveolar macrophages.** To evaluate whether tumour-derived mesenchymal cells modulate AMs, co-cultures were established. AMs (CD45$^+$CD64$^+$SiglecF$^+$) were isolated from wild-type lungs, and mesenchymal cells (CD31$^-$CD45$^-$EpCAM$^-$) were isolated from either wild-type or *Red2Kras* lungs at 4 weeks after three doses of tamoxifen (0.2 mg g$^{-1}$ body weight) injection on alternate days. Freshly sorted AMs and mesenchymal cells were centrifuged at 300*g* for 10 min at 4 °C and resuspended in 3D basic medium. Cells were counted and combined to create mixtures of 30,000 AMs with either 50,000 wild-type or 25,000 *Red2Kras* mesenchymal cells per well. Cells were centrifuged, resuspended in 100-µl Matrigel GFR containing 50% 3D basic medium and plated into 24-well Transwell inserts with 0.4-µm pore size. A total of 500-µl 3D basic medium was added to the lower chamber and replaced every other day. Cultures were maintained for 14 days, after which AMs were analysed.

To assess the effect of TLR4 inhibition on fibroblast–AM interactions, wild-type AMs and mesenchymal cells from either wild-type or *Red2Kras* lungs were isolated as described above. Cells were counted, combined at a ratio of 30,000 AMs to 40,000 mesenchymal cells and then plated in eight-well LabTek Chamber Slides. Co-cultures were maintained for 4 days. For TLR4 inhibition, selected wells were treated with the TLR4 inhibitor TAK-242 (3 µM; Sigma; 614316) beginning at plating (day 0) and continuing throughout the experiment. To assess the effect of inflammatory cues in fibroblasts, wild-type mesenchymal cells were treated with IL-1β (20 ng ml$^{-1}$; PeproTech 211-11B-10UG) or co-cultured with AMs isolated from *Red2Kras* lungs for 48 h.

**Mesenchymal cultures.** To test whether EGFR activation induces fibrotic phenotypes, mesenchymal cells (CD31$^-$CD45$^-$EpCAM$^-$) were isolated from wild-type lungs, centrifuged at 300*g* for 10 min at 4 °C and resuspended in 30-µl Matrigel GFR. A total of 50,000 mesenchymal cells per well were seeded in eight-well LabTek Chamber Slides for whole-mount staining. The Matrigel GFR domes were left to set for 30 min at 37 °C, before 300-µl 3D basic medium was added to each well. Recombinant Areg (100 ng ml$^{-1}$; PeproTech; 315-36) was added to the medium for 5 days.

**Alveolar macrophage cultures.** AMs sorted from wild-type lungs were cultured under three conditions: (1) AMs alone; (2) AMs co-cultured with tumour cells; and (3) AMs co-cultured with tumour cells and fibroblasts. For each condition, cell numbers were as follows: 50,000 AMs for condition 1, 50,000 AMs with 5,000 RFP$^+$ AT2 tumour cells for condition 2 and 50,000 AMs with 5,000 RFP$^+$ tumour AT2 cells and 25,000 fibroblasts for condition 3. Cells were embedded in 20-µl Matrigel GFR domes and cultured in 3D basic medium for 7 days. For Areg treatment, recombinant Areg (20 ng ml$^{-1}$) was added to the culture medium.

To assess responses to Tnc, AMs were isolated from wild-type lungs, plated in eight-well LabTek Chamber Slides and left to adhere overnight. Cells were then treated with Tnc (2 µg ml$^{-1}$; MedChemExpress; HY-P700834) alone or in combination with the TLR4 inhibitor TAK-242 (3 µM; Sigma; 614316) for 48 h.

### Inducible human LUAD organoid development
**Primary human lung alveolar organoid cultures.** Human alveolar organoids were established following a previous study[48]. AT2 cells (CD45$^-$EpCAM$^+$HTII-280$^+$) were isolated from non-tumour lung parenchyma tissues using FACS, resuspended in 20-µl Matrigel GFR and plated in 48-well plates. Domes were incubated at 37 °C for 15 min to allow solidification before adding 250-µl alveolar medium. Alveolar medium consisted of DMEM/F12 supplemented with 1× B27 (Thermo Fisher Scientific; 17504044), 50 ng ml$^{-1}$ of murine EGF

(PeproTech; 100-15), 100 ng ml$^{-1}$ of human FGF7/KGF (PeproTech; 100-19), 100 ng ml$^{-1}$ of human FGF10 (PeproTech; 100-26), 100 ng ml$^{-1}$ of human NOGGIN (PeproTech; 120-10 C), 3 µM CHIR99021 (Tocris; 4423), 500 nM A83-01 (Tocris; 2939), 10 µM SB431542 (Tocris; 616461), 1× penicillin–streptomycin, 500 µg ml$^{-1}$ of Primocin (InvivoGen; ant-pm-1) and 1.25 mM *N*-acetylcysteine (Merck; A9165). The ROCK inhibitor Y-27632 (10 µM) was added to the medium for the first 2 days of culture, and the medium was replaced every 2–3 days.

**Inducible vector construction, viral production and organoid infection.** The plasmid pHAGE–KRAS$^{G12D}$ was a gift from G. Mills and K. Scott (Addgene plasmid no. 116423; PIRD: Addgene_116423). For the inducible system, the *KRAS$^{G12D}$* sequence was cloned into the EF1a–TagRFP–2A–tet3G vector using In-Fusion cloning (vector kindly provided by the Emma Rawlins laboratory, University of Cambridge). Lentivirus was produced by transfecting HEK293T cells (American Type Culture Collection; CRL-11268) using a calcium phosphate protocol, and viral supernatants were collected 48 h post-transfection. Human AT2-cell-derived alveolar organoids (passages 0–2) were recovered from Matrigel GFR using dispase (1 mg ml$^{-1}$; 40 min; 37 °C) and dissociated to single cells with TrypLE (5 min; 37 °C). Cells were subjected to spin infection (2,000 rpm; 32 °C; 60 min) with viral supernatant in the presence of polybrene (8 µg ml$^{-1}$; Sigma), followed by feeder-free culture in Matrigel GFR supplemented with alveolar medium as described above. RFP$^+$ cells were then isolated by FACS on day 14 or 21 post-infection, and approximately 50,000 cells were embedded in Matrigel GFR for co-culture experiments or for single-cell profiling under feeder-free conditions, respectively. For induction of the *KRAS$^{G12D}$* gene, doxycycline (2 µg ml$^{-1}$; Merck) was added every 2 days, starting on day 7.

**Primary human lung alveolar organoid co-cultures with primary human mesenchymal cells.** Fourteen days post-infection, RFP$^+$ infected cells were sorted by FACS and co-cultured with freshly isolated primary human lung mesenchymal cells (EpCAM$^-$CD31$^-$CD45$^-$) at a 1:5 ratio (approximately 1,000 RFP$^+$ cells with 5,000 mesenchymal cells per well) in 20-µl Matrigel GFR. Cultures were established in eight-well chamber slides (µ-Slide 8 wells; ibidi) and maintained in co-culture medium consisting of a 1:1 mixture of alveolar medium and Pneuma-Cult (STEMCELL Technologies). KRAS$^{G12D}$ expression was induced with doxycycline (2 µg ml$^{-1}$; Merck), added every 2 days starting on day 7. Immunofluorescence analysis was performed 7 days post-induction. For EGFR inhibition experiments, gefitinib (Selleckchem; S1025; 5 µM) was added from day 7 following doxycycline induction and maintained for extra 7 days. DMSO-treated cultures served as controls.

### Immunofluorescence staining
**Mouse lung tissue sections.** Individual cryosections were circled using a Hydrophobic PAP Pen (Sigma; Z377821) and placed in a humidified chamber. Sections were permeabilized with 0.3% Triton X-100 (Sigma; X100) in PBS for 15 min, followed by blocking with 0.3% Triton X-100 in PBS containing 5% normal donkey serum (Jackson ImmunoResearch Labs; 017-000-121) for 1 h at room temperature. Primary antibodies (Supplementary Table 1) were diluted in blocking buffer and incubated overnight at 4 °C. Sections were washed three times in 0.2% Tween-20 (Sigma; P9416) in PBS for 5 min each at room temperature, followed by incubation with secondary antibodies (Supplementary Table 1) diluted in PBS for 1 h at room temperature. The nuclear staining DAPI (Sigma; D9542) was added to the secondary antibody mix at 0.5 µg ml$^{-1}$. Following staining, sections were washed three times in PBS, mounted in RapiClear 1.52 (SUNJin Lab; RC152002), enclosed with glass coverslips (VWR; 631-1574) and sealed with nail polish.

**Human lung tissue sections.** Human paraffin-embedded tissue sections were deparaffinized, and antigen retrieval was performed

by incubation at 95 °C for 15 min in sodium citrate buffer (pH 6) (Sigma; S4641) containing 0.05% Tween-20. For immunofluorescence staining, the protocol described above for cryosections was followed.

**Organoid whole-mount staining.** Organoid cultures grown in LabTek Chamber Slides were fixed with 200 μl of 4% PFA for 20 min at room temperature and washed three times with PBS. Cells were permeabilized with 0.5% Triton X-100 in PBS for 15 min at room temperature, followed by blocking in 0.3% Triton X-100 in PBS containing 5% normal donkey serum for 1 h. Primary antibodies (Supplementary Table 1) diluted in blocking buffer were added to each well and incubated overnight at 4 °C. Samples were washed three times with 0.2% Tween-20 in PBS and incubated with secondary antibodies (Supplementary Table 1) diluted in 0.2% Tween-20 in PBS for 2 h at room temperature. DAPI (0.5 μg ml$^{-1}$) was included in the secondary antibody solution for nuclear staining. Wells were washed three times with PBS. Chambers were then removed according to the manufacturer's instructions. Samples were mounted in RapiClear 1.52, enclosed with glass coverslips and sealed with nail polish. The slides were allowed to dry at room temperature before imaging.

## Confocal imaging, processing and quantification

Immunofluorescence images of stained sections and organoids were acquired using a Leica STELLARIS 8 white light laser inverted confocal microscope or an Olympus FV3000RS. Standard configurations were used for all experiments. All representative images were acquired using ×20 or ×40 oil objectives, except for whole-lobe tile scans, which were acquired using a ×10 objective. Confocal images were processed and analysed using Fiji (ImageJ). Signal thresholds were manually adjusted during image processing, and identical settings were applied to all representative images within the same experiment. An exception was made for endogenous fluorescent reporters (RFP and tdTomato), for which thresholds were adjusted as necessary to enable clear visualization of labelled cells in representative images.

Cell quantification in tissue sections was performed manually using the CellCounter plugin or by automated detection of DAPI$^+$ nuclei using the 'Analyze particles' function. Signal thresholds were manually adjusted to distinguish marker-positive and marker-negative cells, with identical settings applied to all images within the same experiment. For most analyses, representative images and quantifications were obtained from a minimum of seven fields of view per mouse sample or experimental condition. For analyses of tumours containing fibrotic fibroblasts, 10–20 individual mutant clones were analysed per mouse. Tumour burden was quantified by defining RFP$^+$ signal thresholds and measuring total RFP$^+$ area using the 'Analyze particles' function. The RFP$^+$ area was normalized to the total area of the whole-lobe cross section, with two to four independent tissue sections analysed per mouse. For quantification of cell numbers in organoids, detection thresholds were manually adjusted, and DAPI$^+$ and SOX9$^+$ nuclei were quantified using the 'Analyze particles' function with a minimum particle size of 10 μm$^2$.

## Quantitative reverse transcription–polymerase chain reaction

Total RNA was isolated using a QIAGEN RNeasy Micro or Mini-plus Kit according to the manufacturer's instructions. Equivalent quantities of total RNA were reverse transcribed with SuperScript IV complementary DNA (cDNA) Synthesis Kit (Life Technologies). Diluted cDNA was analysed by real-time polymerase chain reaction (StepOnePlus; Applied Biosystems). SYBR Green assays were used for human or mouse gene expression with SYBR Green Master Mix (2×; Thermo Fisher Scientific). The primer sequences are as follows:

Mouse Gapdh: F-AGGTCGGTGTGAACGGATTTG, R-TGTAGACCATGT AGTTGAGGTCA

Mouse Arg1: F-CTCCAAGCCAAAGTCCTTAGAG, R- AGGAGCTGTC ATTAGGGACATC

Mouse Ym-1: F-TGGAATTGGTGCCCCTACAA, R- CCACGGCACCTCC TAAATTG

Mouse Tnf: F-CCCTCACACTCAGATCATCTTCT, R- GCTACGACG TGGGCTACAG

Human GAPDH: F-GGAGCGAGATCCCTCCAAAAT, R- GGCTGTTGTC ATACTTCTCATGG

Human KRAS: F-AGTGCCTTGACGATACAGCT, R-CCTCCCCAGTC CTCATGTAC.

## Single-cell transcriptomics

**Library preparation and sequencing. Lung mesenchymal cells from *Confetti* and *Red2Kras* lungs.** Two weeks after three doses of tamoxifen induction (0.2 mg g$^{-1}$ body weight; administered every other day), lung tissues were collected, and mesenchymal cells (CD45$^-$CD31$^-$EpCAM$^-$) were isolated from *Sftpc–Cre^ERT2;Confetti* and *Sftpc–Cre^ERT2;Red2Kras* mice. Cells from three mice of the same genotype were pooled into a single-cell suspension to generate two separate libraries (1× *Confetti* and 1× *Red2Kras*). Cell suspensions were spun down, counted and resuspended in 0.04% bovine serum albumin (BSA; Sigma; A3294) in PBS to achieve a cell concentration of approximately 345 cells μl$^{-1}$. Single-cell 3′ RNA sequencing libraries were generated according to the manufacturer's instructions (Chromium Single Cell 3′ Reagent v.3 Chemistry Kit; 10X Genomics), and cDNA quality was assessed. Libraries were sequenced to a minimum depth of approximately 20,000 reads per cell using Illumina NovaSeq X 1.5B.

**Lung stromal and immune cells from *Confetti* and *Red2Kras* lungs.** Two weeks after three doses of tamoxifen induction (0.2 mg g$^{-1}$ body weight; administered every other day), lung tissues were collected, and stromal and immune cells (1:1 mixture of immune cells (CD45$^+$EpCAM$^-$) and stromal cells (CD45$^-$EpCAM$^-$)) were isolated from *Sftpc–Cre^ERT2;Confetti* and *Sftpc–Cre^ERT2;Red2Kras* mice. Cells from three mice of the same genotype were pooled into a single-cell suspension to generate two separate libraries (1× *Confetti* and 1× *Red2Kras*). Libraries were generated as described above and sequenced to a minimum depth of approximately 20,000 reads per cell using the Illumina NovaSeq 6000.

**RFP$^+$ mutant, mesenchymal and immune cells from *Areg^flox/+* and *Areg^flox/flox* lungs.** Two weeks after two doses of tamoxifen induction (0.2 mg g$^{-1}$ body weight; administered every other day), lung tissues were collected from *Sftpc–Cre^ERT2;Red2Kras;Areg^flox/+* and *Sftpc–Cre^ERT2;Red2Kras;Areg^flox/flox* animals. Lungs from three mice of the same genotype were dissociated and pooled into a single-cell suspension. Lineage-labelled epithelial cells (EpCAM$^+$RFP$^+$) and a mixed population of mesenchymal and immune cells (1:1 ratio of mesenchymal cells (CD45$^-$CD31$^-$EpCAM$^-$) and immune cells (CD45$^+$CD31$^-$EpCAM$^-$)) were then sorted. For each genotype, two separate libraries were generated, resulting in a total of four libraries (2× *Areg^flox/+* and 2× *Areg^flox/flox*). Libraries were sequenced to a minimum depth of approximately 20,000 reads per cell using Illumina NovaSeq 6000.

**Human lung alveolar organoids.** Feeder-free organoids from control or *KRAS^G12D*-induced RFP$^+$ cells were used for single-cell RNA sequencing (scRNA-seq) analysis by isolating cells on day 7 following doxycycline-mediated induction. For cell isolation, organoids were incubated with dispase (1 mg ml$^{-1}$; 30–60 min), dissociated with TripLE (Gibco) for 5 min and washed with PBS. Libraries were prepared as described above and sequenced to a minimum depth of approximately 20,000 reads per cell using Illumina NovaSeq 6000.

**Read alignment.** Raw FASTQ files containing droplet-based sequencing data were preprocessed in CellRanger (v.6.0.2). Reads were aligned to the Ensembl *Mus musculus* GRCm38 reference genome or *Homo sapiens* GRCh38 (GENCODE v.38), empty droplets were filtered out and the number of unique molecular identifiers (UMIs) mapped to each protein-coding gene was quantified to generate the final count matrices.

**Quality control.** Analysis of count matrices was performed in R using the Seurat package[59] or Scanpy[60] pipeline (v.1.9.1). Quality control

metrics for each library were first assessed and used to define thresholds for filtering out low-quality cells and possible doublets. Standard cutoffs of less than 10% mitochondrial genes, more than 1,000 detected genes, more than 2,000 UMIs and fewer than 50,000 UMIs were used for most cases. For the immune cell dataset from the *Red2Kras* versus *Confetti* experiment and human alveolar organoid, cells were filtered by custom cutoff (more than 500 and less than 7,000 detected genes and more than 2,000 UMI count).

**Dimensionality reduction, clustering and analysis.** Seurat pipeline was used for further data processing. Briefly, filtered data were log-normalized and scaled, and the top 2,000 highly variable genes were used for principal component analysis (PCA). The first 30 PCAs were used for downstream analyses. Nearest neighbours were calculated. Cells were clustered using Louvain algorithm and visualized using UMAP. Seurat objects were integrated using Harmony. After integration, count matrices were renormalized, and PCA-based dimensionality reduction, clustering and UMAP visualization were performed. Markers for each clusters were identified using the FindAllMarkers() function, and individual populations were annotated on the basis of previously described gene markers for immune, mesenchymal and epithelial lung cell types[7,23,24]. Unwanted cell types were manually removed where appropriate. Cell populations of interest were subset, reclustered and reprocessed. Gene expression between clusters was visualized using the DotPlot(), VlnPlot(), FeaturePlot() and Heatmap() functions. Gene Ontology terms for 'biological processes' were obtained using g:Profiler on the top differentially expressed genes, applying a significance threshold of $P < 0.05$. Selected biologically relevant terms were presented in the figures. Cell trajectory analysis for selected populations was performed using the Monocle 3 package[61]. Identification of communication networks and ligand–receptor pairs between epithelial cells and fibroblasts was performed using CellChat by following standard analysis protocols[62].

**Integration of oncogenesis and regeneration mesenchyme datasets.** To compare mesenchymal transcriptional profiles during injury response and early oncogenesis, our scRNA-seq dataset was integrated with a previously published dataset of mesenchymal cells during lung regeneration[24]. Both datasets were processed and filtered as described above. A total of 10,000 anchor features were identified using the FindIntegrationAnchors() function, and the datasets were integrated using IntegrateData() function. Dimensional reduction, log normalization, scaling, clustering and downstream analyses were performed as described above.

**Analysis of early-stage human LUAD scRNA-seq datasets.** To investigate the presence of mutant epithelial states and fibrotic fibroblasts in early-stage human LUAD, we reanalysed a previously generated scRNA-seq dataset for primary LUAD tumours (stages I–III) and distant normal tissue[45]. Only samples with confirmed EGFR wild-type status were used for the analysis. For quality control, cutoffs of less than 20% mitochondrial genes, more than 200 detected genes and fewer than 3,000 detected genes were used. Dimensional reduction, log normalization, scaling, clustering and downstream analyses were performed as described above using the Seurat (v.5) package on R studio (v.4.4.2). From the resulting dataset, epithelial cells were reclustered on the basis of EPCAM expression, with ciliated cells (FOXJ1) excluded. Mesenchymal cells were reclustered on the basis of COL1A1 expression, and fibroblasts were further reclustered on the basis of PDGFRα, COL13A1 and COL14A1 expression. Gene expression between cell clusters was visualized as described above.

**Statistical analysis and reproducibility**
All in vivo experiments were performed in at least two independent experiments, with individual animals considered as biological replicates. All in vitro assays were performed in at least three independent experiments, and summary statistics were calculated from experiment-level mean values, unless otherwise stated. For most quantifications, 10–20 tumours or a minimum of seven fields of view were analysed per mouse or per experimental condition, and measurements were treated as nested within each mouse or condition and averaged to obtain mouse-level or experiment-level values for statistical analysis. For tumour burden, tumour area was normalized to the total lobe area across sections, with two to four independent tissue sections analysed per mouse, and section-level measurements were averaged to yield a single mouse-level value for statistical analysis. For individual tumour size, two slides per animal, spaced 100 μm apart, were evaluated. Total lobes and lesion areas were defined manually and measured using QuPath open-source software (v.0.6.0).

Data are presented as mean ± s.e.m. Statistical analyses were performed using Prism software (GraphPad; v.7.0) or R. Statistical significance was assessed using two-tailed unpaired Student's *t*-test or two-tailed Mann–Whitney test, as indicated in the figure legends. The number of animals or in vitro assays is stated in the figure legends ($n = x$ mice per group; $n = x$ independent experiments per condition). Representative images are shown, and the corresponding quantifications are derived from the indicated numbers of animals and experiments. Nested statistical analyses that accounted for within-mouse or within-experiment variability yielded results consistent with analyses of mouse-level or experiment-level means. Therefore, these summary values were used for statistical comparisons, and the specific statistical tests and exact *P* values are reported in the figures and figure legends.

**Reporting summary**
Further information on research design is available in the Nature Portfolio Reporting Summary linked to this article.

## Data availability
Single-cell RNA sequencing datasets have been deposited in the Gene Expression Omnibus under the following accession numbers: human alveolar organoids (GSE310335); mesenchymal (GSE316241) and immune (GSE316243) cells from *Confetti* and *Red2Kras* lungs, respectively; and tumours and stromal cells from *Areg*^flox/+^ and *Areg*^flox/flox^ lungs (GSE316244). Source data are provided with this paper.

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

**Acknowledgements** We thank members of the Lee laboratory for discussions; C. Rudin for coordinating the acquisition of human lung tissue; and the Royal Papworth Hospital NHS Foundation Trust, NIHR Cambridge BRC Cell Phenotyping Hub, Cambridge Research UK Cancer Institute Genomics Core Facility, UBS Biofacility, Cambridge Stem Cell Institute Core Facilities, Sloan Kettering Institute's Molecular Cytology Core Facility, Sloan Kettering Institute's Flow Cytometry Core Facility and GIST Advanced Institute of Instrumental Analysis for technical assistance. This study was supported by a Wellcome Trust Senior Research Fellowship (221857/Z/20/Z) and Suh Kyung-bae Foundation Award (SUHF-20010033). J.C. was supported by the National Research Foundation of Korea grant funded by the Korean government (MSIT) (NRF-2022R1A2C1091644, RS-2024-00438368 and RS-2024-00411768) and by the Ministry of Health and Welfare (RS-2025-25459531 and RS-2025-24536036). H.L. was supported by the National Research Foundation (RS-2025-25428221). F.J.E. was supported by a Wellcome PhD Studentship (220088/Z/20/Z). B.D.S. is supported by the Wellcome Trust (219478/Z/19/Z) and the Royal Society EP Abraham Research Professorship (RP\R\231004).

**Author contributions** E.C.C., J.C., F.J.E. and J.-H.L. conceived and designed the study and interpreted the data. E.C.C., J.C. and J.-H.L. wrote the paper. E.C.C. led the project and performed most of the experiments and analyses. J.C. performed immune cell analyses and generated single-cell data from human lung organoids. H.L. performed clodronate, engraftment and

organoid assays under the supervision of J.C. F.J.E. identified fibroblast reprogramming and generated single-cell data of mutant epithelial and immune cells. H.C. supported organoid assays. R.L. supported the immunofluorescence of human tissues. S.S.V. optimized mesenchymal cell isolation. M.S.P. and N.R. provided human samples. B.-K.K. shared the *Red2Kras*$^{G12D}$ mouse. B.D.S. co-supervised E.C.C. and F.J.E. and provided the discussion. J.-H.L. supervised the study, directed the project and secured funding.

**Competing interests** The authors declare no competing interests.

**Additional information**

**Correspondence and requests for materials** should be addressed to Jinwook Choi or Joo-Hyeon Lee.

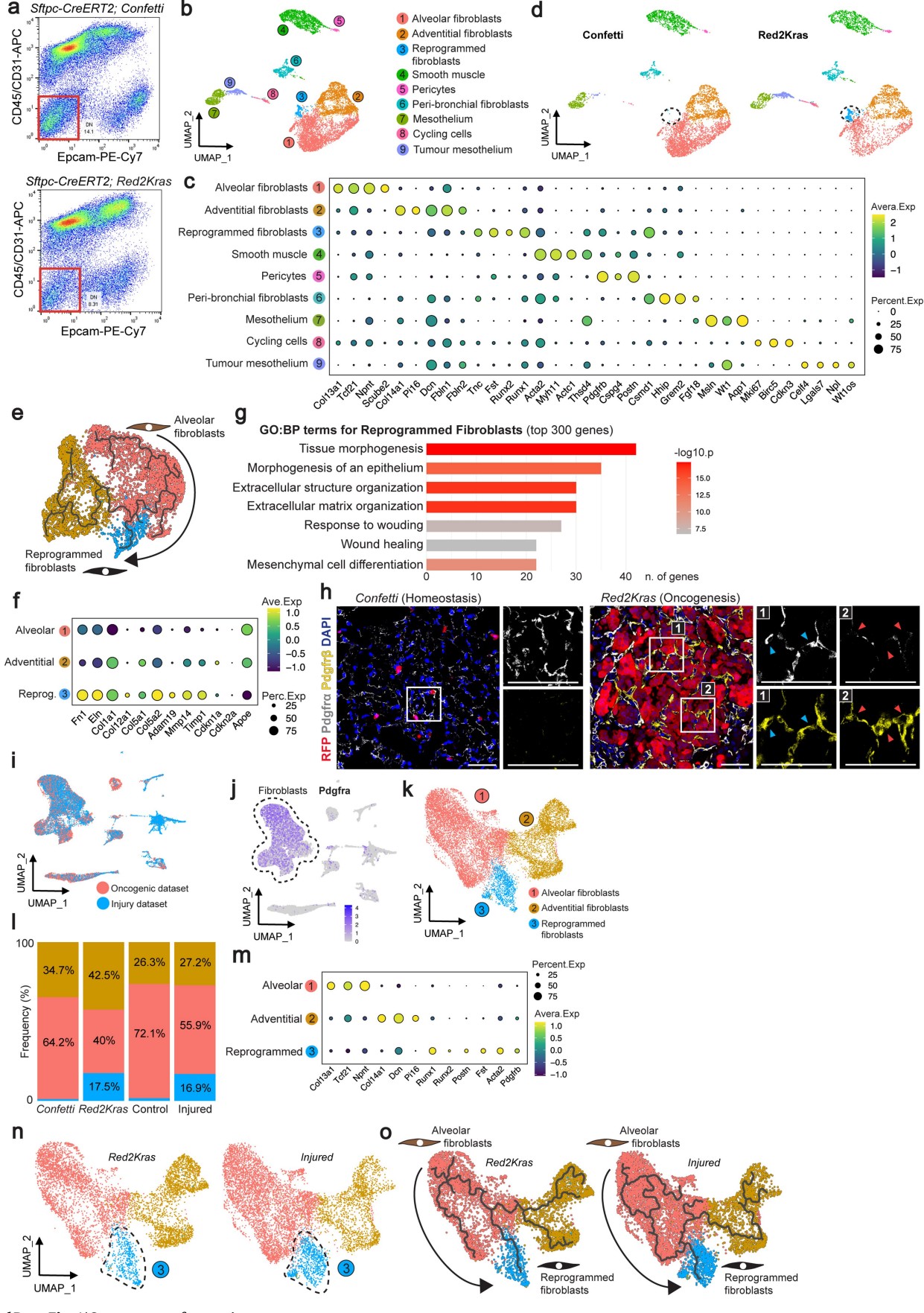

**Extended Data Fig. 1** | See next page for caption.

**Extended Data Fig. 1 | Alveolar fibroblasts acquire a regenerative-like signature during early oncogenesis. a**, Fluorescence-Activated Cell Sorting (FACS) gating strategy for isolating mesenchymal cells (CD45⁻CD31⁻EpCAM⁻) from *Confetti* (top) and *Red2Kras* (bottom) lungs for single-cell RNA sequencing (scRNA-seq). Red boxes highlight the sorted population used for library generation in each sample. **b**, UMAP plot showing all integrated single-cell transcriptomics of mesenchymal cells obtained from *Confetti* and *Red2Kras* datasets, annotated into the nine identified populations. Cells are coloured and numbered by population. **c**, Dotplot showing the expression of key marker genes distinguishing each cluster annotated in (**b**). Each dot is coloured based on the average expression and sized based on the percentage of cells expressing that gene. **d**, UMAP plots showing the distribution of mesenchymal cells in *Confetti* and *Red2Kras* lungs separately. Dashed lines highlight a cluster of reprogrammed fibroblasts enriched in *Red2Kras* lungs. Cells are coloured by population. **e**, All predicted trajectories generated from the dataset of fibroblasts from *Red2Kras* lungs. Cells are coloured by populations. Analysis was performed on the dataset prior to annotation. Lines illustrate all different cell trajectories predicted by the analysis. **f**, Dotplot showing the expression of ECM-related and senescence marker genes in each fibroblast population annotated in Fig. 1b. Each dot is coloured based on the average expression and sized based on the percentage of cells expressing that gene. **g**, Gene Ontology (GO) terms significantly enriched in reprogrammed fibroblast gene signatures compared to alveolar fibroblasts, plotted against their corresponding -$\log_{10}$ adjusted *p*-values. Only the top 300 differentially expressed genes (DEGs) were used for the analysis. **h**, Representative confocal images showing reprogrammed fibroblast markers and lineage-labelled AT2 cells in *Confetti* and *Red2Kras* lungs. DAPI (blue), Pdgfrα (grey), Pdgfrβ (yellow) and RFP (red, tumour). Right panels are magnifications of left panels. Blue arrows, Pdgfrα⁺Pdgfrβ⁺ fibroblasts; Orange arrows, Pdgfrα^low Pdgfrβ⁺ fibroblasts. Scale bar, 50 μm. Images representative of *n = 3* mice. **i**, UMAP plot showing the distribution of mesenchymal cells in bleomycin injury and our *Red2Kras* datasets after integration of mesenchymal cell transcriptomics[24]. Cells are coloured according to their dataset of origin. **j**, UMAP feature plot showing fibroblasts expressing *Pdgfra* in (**i**), as indicated by dashed lines in the plot. **k**, UMAP plot showing all fibroblasts obtained from the integration of the *Red2Kras* and bleomycin-injury datasets[24], clustered into three distinct populations. Cells are coloured and numbered by population. **l**, Percentage of cells distributed across each cluster annotated in (**k**). Percentages smaller than 5% are omitted from the bar plots. **m**, Dotplot showing the expression of key marker genes for each fibroblast state annotated in (**k**). Each dot is coloured based on the average expression and sized based on the percentage of cells expressing that gene, as shown. **n**, UMAP plots showing the distribution of *Red2Kras* and bleomycin-injury datasets separately. Dashed lines highlight a cluster of reprogrammed fibroblasts enriched in *Red2Kras* lungs. Cells are coloured by population. **o**, All predicted trajectories generated for fibroblasts from *Red2Kras* (left) and bleomycin-injury (right) datasets. Cells are coloured by populations. Analysis was performed on the dataset prior to annotation. Lines illustrate all different cell trajectories predicted by the analysis.

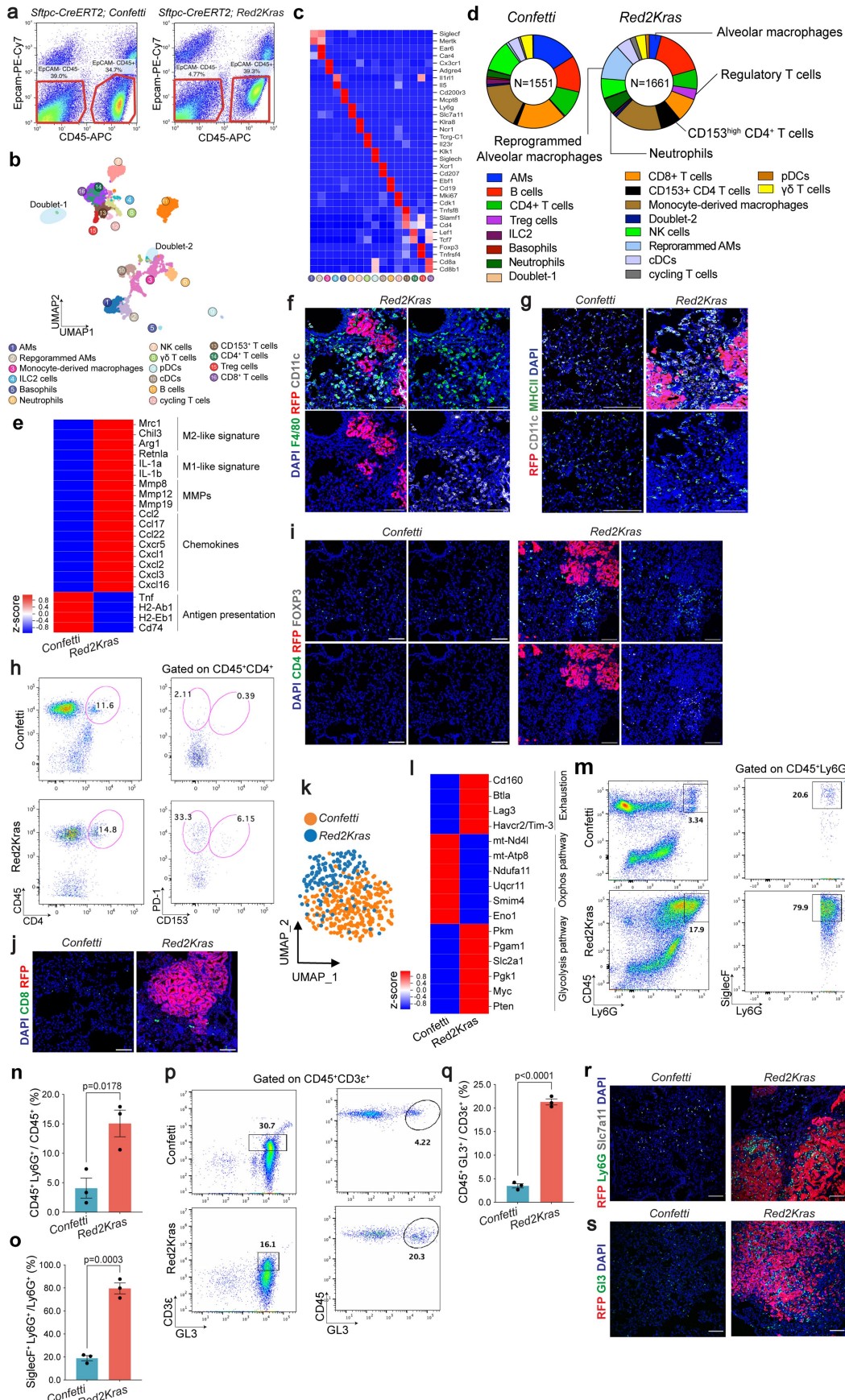

**Extended Data Fig. 2** | See next page for caption.

**Extended Data Fig. 2 | Dynamic immune cell landscape during early tumour development. a**, Fluorescence-Activated Cell Sorting (FACS) gating strategy for isolating immune cells (CD45$^+$) and both mesenchymal and endothelial cells (CD45$^-$EpCAM$^-$) from *Confetti* (top) and *Red2Kras* (bottom) lungs for single-cell RNA sequencing (scRNA-seq). Red boxes highlight the sorted populations used for library generation in each sample. **b**, UMAP plot showing all integrated single-cell transcriptomics of immune cells obtained from *Confetti* and *Red2Kras* datasets, annotated into the 16 identified populations. Cells are coloured and numbered by population. **c**, Heatmap plot showing key gene markers for each immune population annotated in (**b**). **d**, Distribution of each immune cell cluster in *Confetti* and *Red2Kras* lungs. Number of cells in the individual cluster is depicted in the figure. **e**, Heatmap plot showing the expression of genes altered in reprogrammed macrophages from *Red2Kras* lungs relative to alveolar macrophages from *Confetti* lungs, indicating enhanced potential for recruiting other immune cells. **f,g**, Representative confocal images showing alveolar macrophages and lineage-labelled AT2 cells in *Red2Kras* lungs at 4 weeks following oncogenic activation. DAPI (blue), F4/80 (pan macrophage, green), CD11c (alveolar macrophage, grey) and RFP (red, tumour) (**f**). DAPI (blue), CD11c (alveolar macrophage, grey), MHC-II (green) and RFP (red, tumour) (**g**). Scale bar, 100 μm. Images representative of *n* = 3 mice. **h**, Flow cytometry analysis of CD153$^+$PD-1$^+$CD4$^+$ T cells in *Confetti* and *Red2Kras* lungs at 4 weeks following oncogenic activation. **i**, Representative confocal images for regulatory T cells and lineage-labelled AT2 cells in *Confetti* and *Red2Kras* lungs at 4 weeks following oncogenic activation. DAPI (blue), CD4 (green), FOXP3 (grey) and RFP (red, tumour). Scale bar, 100 μm. Images representative of *n* = 3 mice. **j**, Representative confocal images for CD8$^+$ T cells and lineage-labelled AT2 cells in *Confetti* and *Red2Kras* lungs at 4 weeks following oncogenic activation. DAPI (blue), CD8 (green) and RFP (red, tumour). Scale bar, 100 μm. Images representative of *n* = 3 mice. **k**, UMAP plot showing all sub-clustered CD8$^+$ T cells annotated based on the dataset of origin. **l**, Heatmap plot showing exhaustion markers and metabolic signatures of CD8$^+$ T cells from (**k**). **m**, Flow cytometric analysis for neutrophils in *Confetti* and *Red2Kras* lungs, at 4 weeks following oncogenic activation. **n,o**, Quantification of the percentage of Ly6G$^+$ neutrophils (**n**) and Ly6G$^+$SiglecF$^+$ neutrophils (**o**) in *Confetti* (*n* = 3) and *Red2Kras* (*n* = 3) lungs. Data are mean ± s.e.m. Each dot represents a biologically independent mouse replicate. Data are mean ± s.e.m. *P* values were calculated using two-tailed unpaired t-test. **p**, Flow cytometric analysis for γδ T cells in *Confetti* and *Red2Kras* lungs, at 4 weeks following oncogenic activation. GL3 (TCR gamma/delta), CD3ε (T cells), and CD45 (pan immune cells). **q**, Quantification of the percentage of γδ T cells in *Confetti* (*n* = 3) and *Red2Kras* (*n* = 3) lungs. Data are mean ± s.e.m. Each dot represents a biologically independent mouse replicate. *P* values were calculated using two-tailed unpaired t-test. **r, s**, Representative confocal images for neutrophils (**q**), γδ T cells (**r**) and lineage-labelled AT2 cells in *Confetti* and *Red2Kras* lungs at 4 weeks following oncogenic activation. DAPI (blue), Ly6G (neutrophils, green), Slc7a11 (neutrophils, grey) and RFP (red, tumour) (**q**). DAPI (blue), Gl3 (TCR gamma/delta, green) and RFP (red) (**r**). Scale bar, 100 μm. Images representative of *n* = 3 mice.

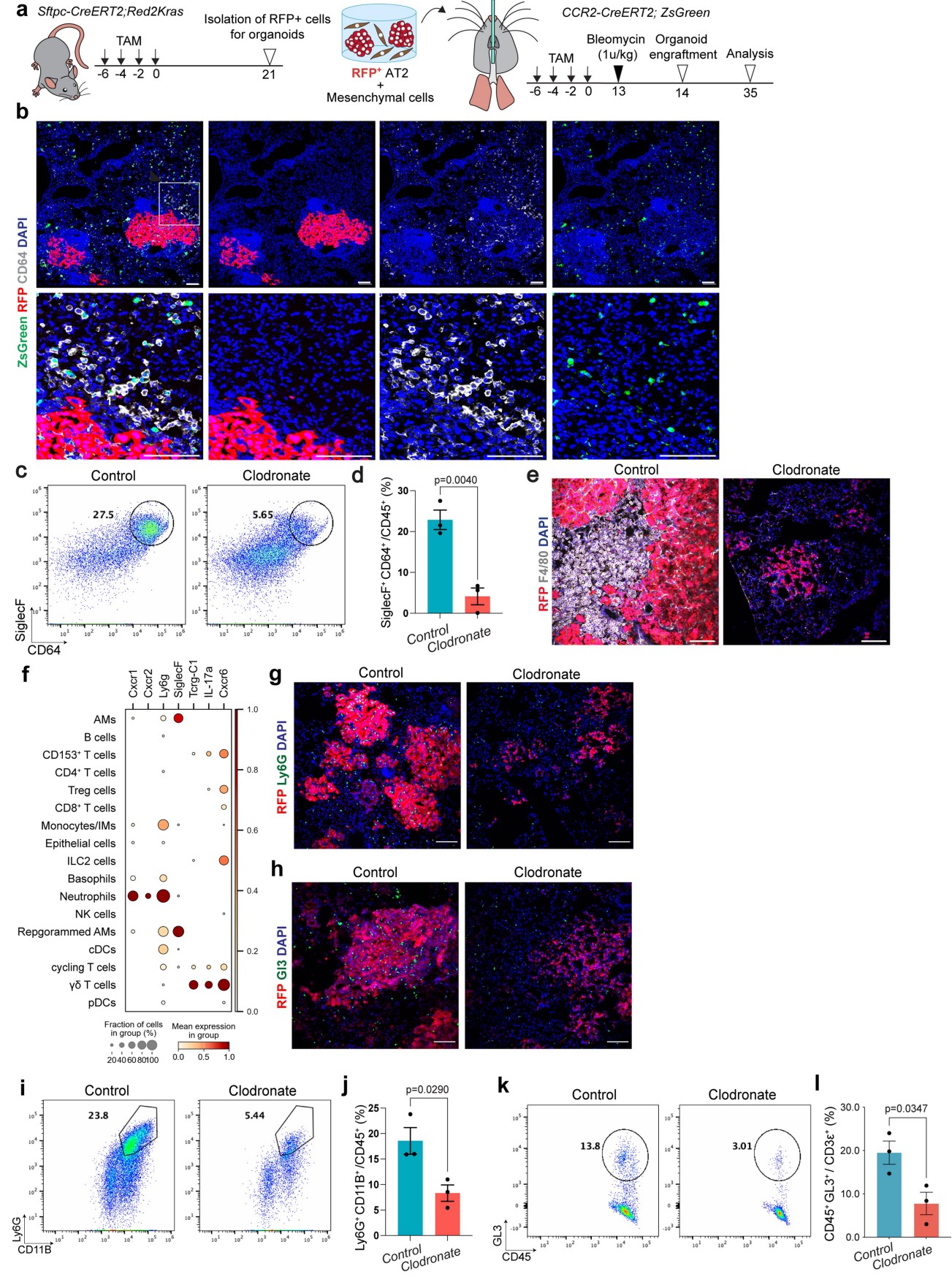

**Extended Data Fig. 3** | See next page for caption.

**Extended Data Fig. 3 | Reprogrammed resident alveolar macrophages establish the early tumour niche. a**, Experiment scheme of orthotopic engraftment of tumour organoids into the lungs of *CCR2-Cre^{ERT2};ZsGreen* mouse. **b**, Representative confocal images showing lineage^+ and lineage^- macrophages in the lungs of *CCR2-Cre^{ERT2};ZsGreen* mice after tumour organoid engraftment. DAPI (blue), ZsGreen (lineage-traced monocytes, green), CD64 (grey), and RFP (red, tumour). Scale bar, 100 μm. Insets (bottom) are enlargements of the dashed boxes (top). Scale bars, 50 μm. Images representative of *n = 3* mice. **c**, Flow cytometry analysis of alveolar macrophages in control and clodronate liposome-treated lungs. **d**, Quantification of alveolar macrophages in control (*n = 3*) and clodronate liposome-treated (*n = 3*) lungs. Data are mean ± s.e.m. Each dot represents a biologically independent mouse replicate. *P* values were calculated using two-tailed unpaired t-test. **e**, Representative confocal images of macrophages and lineage-labelled AT2 cells in control and clodronate liposome-treated lungs. DAPI (blue), F4/80 (grey) and RFP (red, tumour). Scale bar, 100 μm. Images representative of *n = 3* mice. **f**, Dotplot of key chemokine receptors in neutrophils and γδ T cells. Each dot is coloured based on the average expression and sized based on the percentage of cells expressing that gene, as shown. **g,h**, Representative confocal images of neutrophils (**g**), γδ T cells (**h**) and lineage-labelled AT2 cells in control and clodronate liposome-treated lungs. DAPI (blue), Ly6G (green) and RFP (red) (**g**). DAPI (blue), GL3 (green) and RFP (red) (**h**). Scale bar, 100 μm. Images representative of *n = 3* mice. **i**, Flow cytometry analysis of neutrophils from control and clodronate liposome-treated lungs, at 4 weeks following oncogenic activation. Ly6G (neutrophils), CD11B (leukocytes/granulocytes). **j**, Quantification of neutrophils from control (*n = 3*) and clodronate liposome-treated (*n = 3*) lungs. Data are mean ± s.e.m. Each dot represents a biologically independent mouse replicate. *P* values were calculated using two-tailed unpaired t-test. **k**, Flow cytometry analysis of γδ T cells from control and clodronate liposome-treated lungs, at 4 weeks following oncogenic activation. GL3 (TCR gamma/delta), CD3ε (T cells). **l**, Quantification of γδ T cells from control (*n = 3*) and clodronate liposome-treated (*n = 3*) lungs. Data are mean ± s.e.m. Each dot represents a biologically independent mouse replicate. *P* values were calculated using two-tailed unpaired t-test.

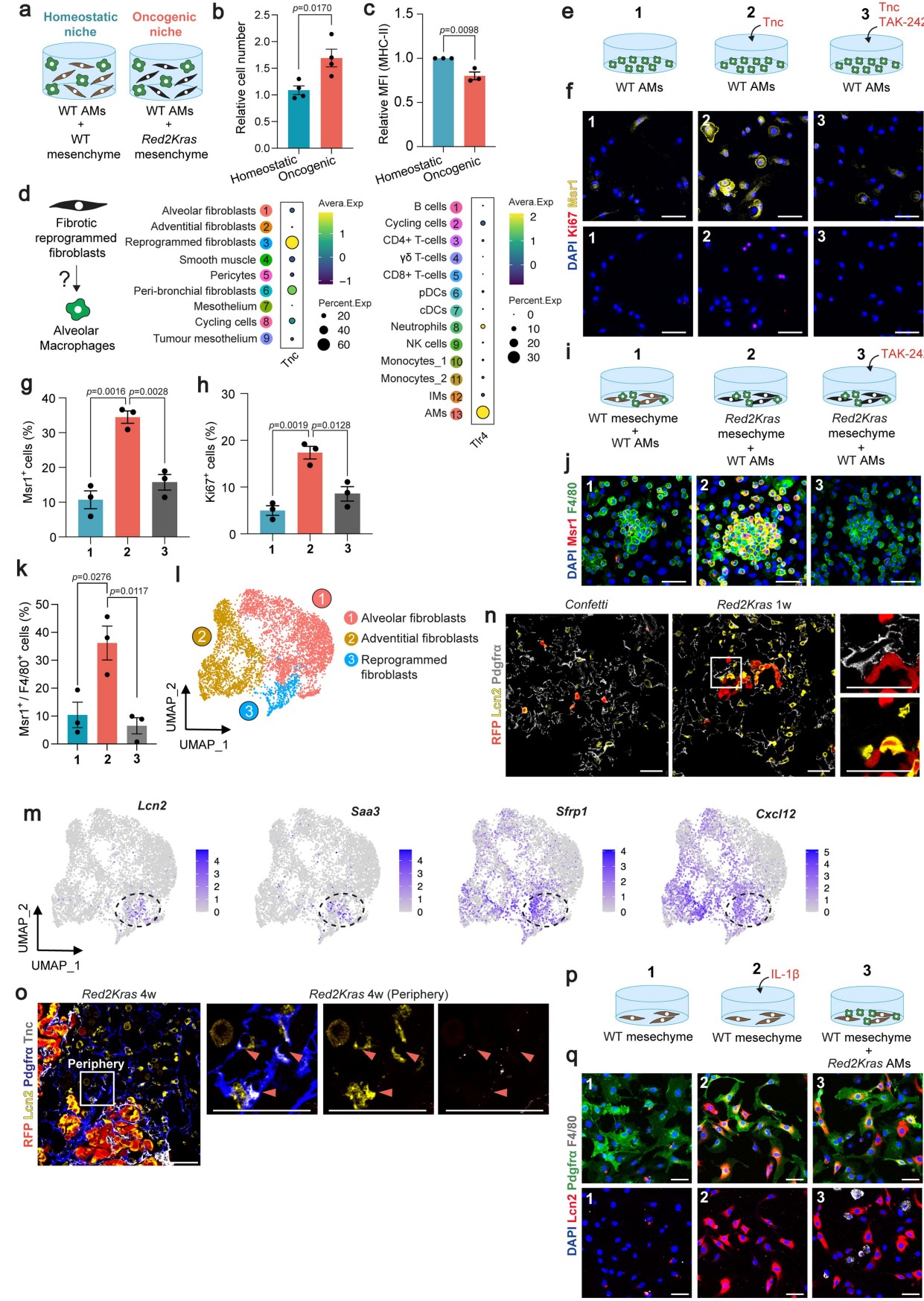

**Extended Data Fig. 4** | See next page for caption.

**Extended Data Fig. 4 | A Tnc-TLR4 axis induces alveolar macrophage reprogramming. a**, Schematic illustrating 3D Matrigel co-cultures of wildtype (WT) AMs with mesenchymal cells isolated from WT (homeostatic) or *Red2Kras* (oncogenic) lungs. **b**, Quantification of relative AMs numbers in the co-cultures from (**a**). Data are mean ± s.e.m. Each dot represents an independent experiment. Homeostatic ($n = 4$) and Oncogenic ($n = 4$). *P* values were calculated using two-tailed unpaired t-test. **c**, Quantification of the MFI for MHC-II of AMs in the co-cultures from (**a**). Data are mean ± s.e.m. Each dot represents an independent experiment. Homeostatic ($n = 3$) and Oncogenic ($n = 3$). *P* values were calculated using two-tailed unpaired t-test. **d**, DotPlots showing the expression of *Tnc* within mesenchymal cells and *TLR4* within all immune cell populations. Each dot is coloured based on the average expression and sized based on the percentage of cells expressing that gene. **e**, Schematic of WT AM culture conditions: 1. Untreated; 2. Tnc-treated; 3. Treatment with Tnc and TLR4 inhibitor (TAK-242). **f**, Representative confocal images for AMs from cultures in (**e**). DAPI (blue), Msr1 (yellow) and Ki67 (red). Scale bar, 50 μm. **g, h**, Quantification of the percentage of Msr1+ (**g**) and Ki67+ (**h**) AMs. Data are mean ± s.e.m. Each dot represents an independent experiment. Untreated ($n = 3$). Tnc ($n = 3$) and Tnc plus TAK-242 ($n = 3$). *P* values were calculated using two-tailed unpaired t-test. **i**, Schematic illustrating 2D co-cultures of WT AMs with mesenchymal cells isolated from WT or *Red2Kras* lungs. The TLR4 inhibitor TAK-242 was added to some of the co-cultures. **j**, Representative confocal images showing reprogrammed macrophages in the co-cultures from (**i**). DAPI (blue), Msr1 (red), F4/80 (green). Scale bar, 50 μm. **k**, Quantification of Msr1+ macrophages in the co-cultures from (**i**). Data are as mean ± s.e.m. Each dot represents an independent experiment. $n = 3$ for all experimental conditions. *P* values were calculated using two-tailed unpaired t-test. **l**, UMAP plot showing fibroblast subclusters annotated into three distinct populations (from Fig. 1b). Cells are coloured and numbered by population. **m**, Feature plots showing the expression of inflammatory markers within a sub-set of reprogrammed fibroblasts. **n**, Representative confocal images for inflammatory fibroblasts and lineage-labelled AT2 in *Red2Kras* lungs at 1 week following oncogenic activation. Mice received a single tamoxifen dose (0.1 mg/gbw). Lcn2 (yellow), Pdgfra (grey) and RFP (red, tumour). Scale bar, 40 μm. **o**, Representative confocal images for fibrotic and inflammatory fibroblasts and lineage-labelled AT2 in *Red2Kras* lungs at 4 week following oncogenic activation. Lcn2 (yellow), Pdgfra (blue), Tnc (grey) and RFP (red, tumour). Right panels are magnifications of the left panels. Orange arrows highlight Lcn2+ inflammatory fibroblasts with no Tnc expression. Scale bar, 50 μm. **p**, Schematic illustrating WT fibroblast cultures treated with IL-1β, or co-cultured with AMs isolated from *Red2Kras* lungs. **q**, Representative confocal images of the cultures from (**p**). DAPI (blue), Lcn2 (red), Pdgfra (green), F4/80 (grey). Scale bar, 50 μm.

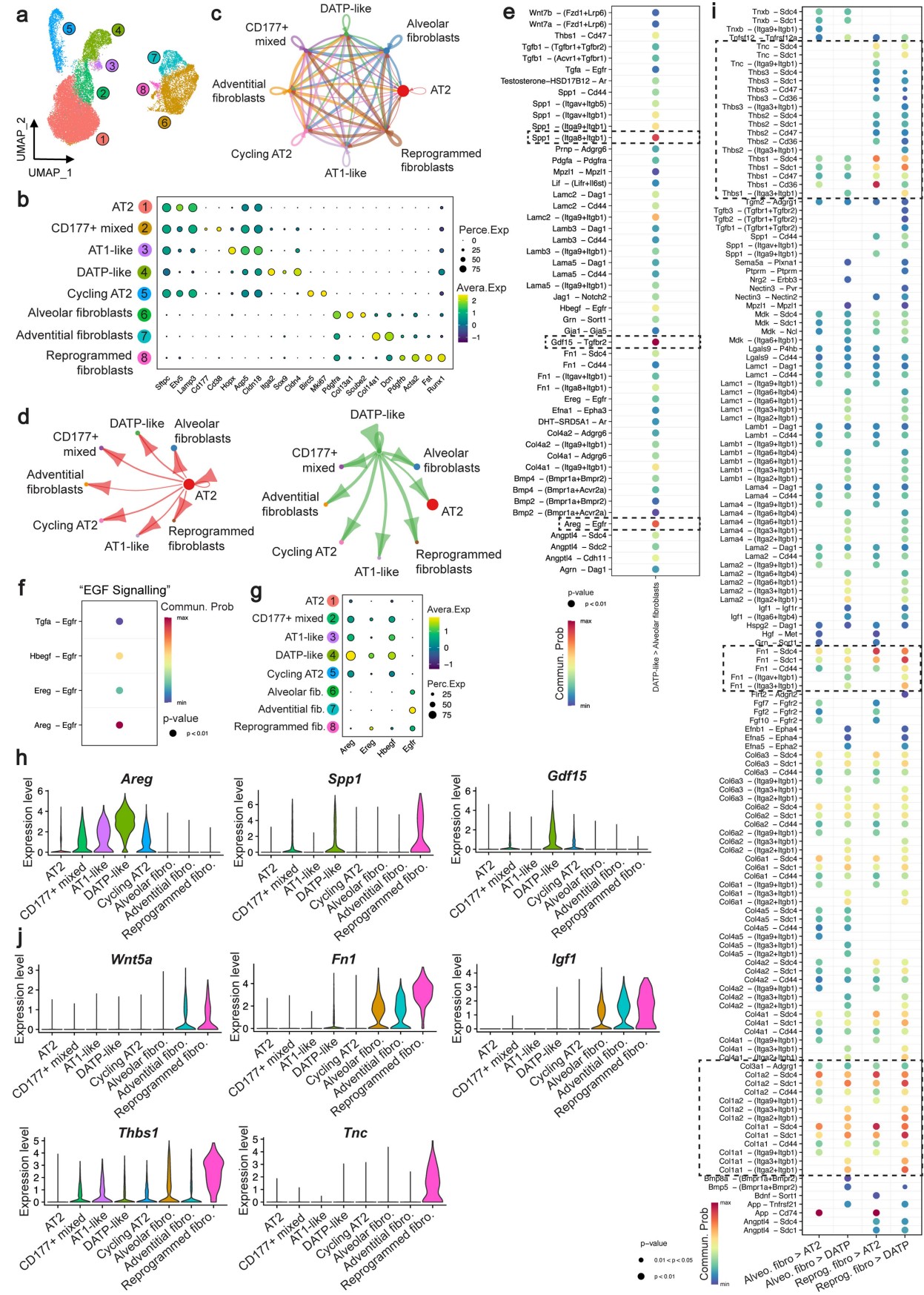

**Extended Data Fig. 5** | See next page for caption.

**Extended Data Fig. 5 | Regenerative mutant cell state serves as the signalling source for niche remodelling. a**, UMAP showing the integration of our single-cell transcriptomics datasets from lineage-labelled RFP[+] cells[7] and lung fibroblasts in *Red2Kras* and *Confetti* lungs. Cells and numbers are coloured by population. **b**, Dotplot showing the expression of key epithelial and fibroblast cell marker genes in the identified populations, annotated in **(a)**. Each dot is coloured based on the average expression and sized based on the percentage of cells expressing that gene. **c**, Aggregated cell-cell communication networks between epithelial and fibroblast states. The thickness of the connections represents the number of significant interactions identified by CellChat. **d**, Aggregated cell-cell communication networks with AT2 or DATP-like cells as the signalling source. The thickness of the connections represents the number of significant interactions identified by CellChat. **e**, All significant outgoing signals from DATP-like cells to alveolar fibroblasts identified by CellChat. Dashed lines highlight the top ligand-receptor pairs identified. Each dot is coloured according to the cellular communication probability. **f**, Outgoing EGF signalling from DATP-like mutant cells to alveolar fibroblasts highlighting Areg-EGFR as the pair with the highest integration probability. Each dot is coloured according to the cellular communication probability. **g**, Dotplot showing the expression of EGFR and EGFR-binding ligands in the cell populations annotated in (**a**). Each dot is coloured based on the average expression and sized based on the percentage of cells expressing that gene. **h**, Violin plot showing the expression of ligands identified as the top signals in the communication analysis from DATP-like cells to alveolar fibroblasts. **i**, All significant outgoing signals from fibroblasts to epithelial cells identified by CellChat. Dashed lines highlight the top ligand-receptor pairs identified. Each dot is coloured according to the cellular communication probability. **j**, Violin plot showing the expression of ligands identified as the top signals in the communication analysis from fibroblasts to epithelial cells.

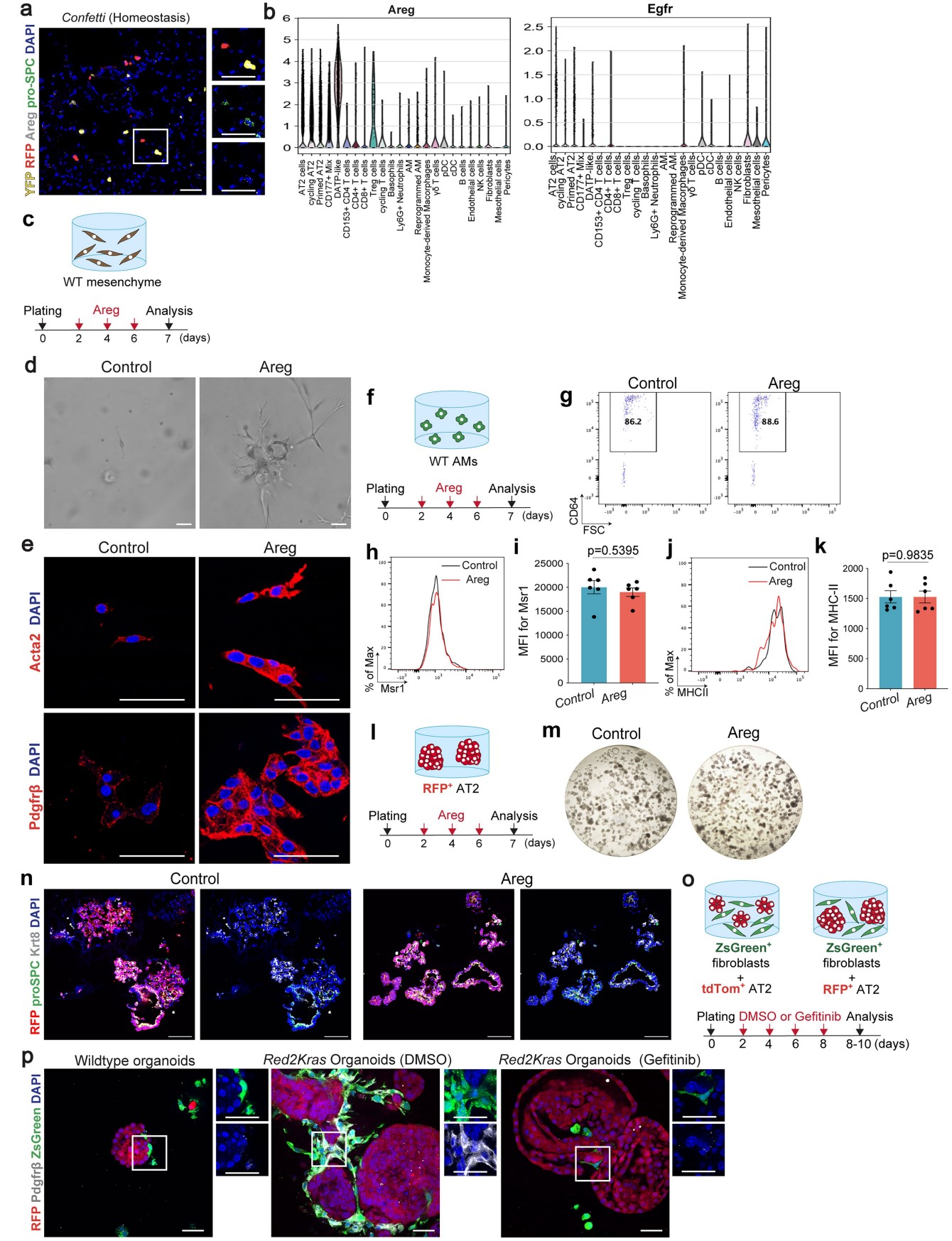

**Extended Data Fig. 6** | See next page for caption.

**Extended Data Fig. 6 | AREG–EGFR signalling reprograms fibroblasts but does not directly drive the reprogramming of tumour cells and macrophages. a**, Representative confocal images confirming an absence of detectable Areg expression in homeostatic lungs (*Confetti*). Right panels are the magnifications of left panel. DAPI (blue), pro-SPC (green), Areg (grey), YFP (yellow) and RFP (red). Scale bar, 50 μm. Image representative of *n* = 2 mice. **b**, Violin plot showing the expression of *Areg* and EGFR across epithelia, mesenchymal and immune cell populations. **c**, Experimental scheme to investigate the effect of Areg on wildtype mesenchymal cells. **d**, Representative brightfield images showing morphological changes of lung mesenchymal cells treated with Areg after 7 days in 3D Matrigel culture. Images are representative of *n* = 3 independent experiments. Scale bar, 50 μm. **e**, Representative confocal images showing the expression of reprogrammed fibroblast markers in lung mesenchyme cultures in (**c**). Top panel: DAPI (blue) and Acta2 (red). Bottom panel: DAPI (blue) and Pdgfrβ (red). Scale bar, 50 μm. **f**, Experimental scheme to investigate the effect of Areg in AMs. **g**, Flow cytometry analysis of AMs numbers after Areg treatment. **h**, Flow cytometry analysis showing Msr1 expression in AMs from (**f**). **i**, Quantification of the MFI for Msr1 in AMs from (**f**). Data are mean ± s.e.m. Each dot represents an independent experiment. Control (*n* = 6) and Areg (*n* = 6). *P* values were calculated using two-tailed unpaired t-test. n.s., not significant. **j**, Flow cytometry analysis showing MHCII expression on AMs from (**f**). **k**, Quantification of the MFI for MHCII in AMs from (**f**). Data are mean ± s.e.m. Each dot represents an independent experiment. Control (*n* = 6) and Areg (*n* = 6). *P* values were calculated using two-tailed unpaired t-test. n.s., not significant. **l**, Experimental scheme to investigate the effect of Areg on RFP⁺ tumour cells isolated from *Red2Kras* lungs. **m**, Bright-field image from (**l**). **n**, Representative confocal image of RFP⁺ tumour organoids from (**l**). DAPI (blue), Krt8 (grey), pro-SPC (green) and RFP (red). Images representative of *n* = 3 independent experiments. Scale bar, 100 μm. **o**, Schematic illustrating 3D organoid co-cultures of fibroblasts isolated from *Pdgfra-Cre^ERT2^; ZsGreen* lungs with lineage-labelled cells isolated from *Sftpc-Cre^ERT2^; tdTomato* or *Red2Kras* lungs. Gefitinib or DMSO was treated for the indicated duration. **p**, Representative confocal images showing reprogrammed fibroblasts and lineage-labelled cells in (**o**). Right panels are magnifications of left panel for each condition. DAPI (blue), ZsGreen (green), Pdgfrβ (grey), and RFP (red). Scale bar, 50 μm. Images of representative *n* = 3 independent experiments.

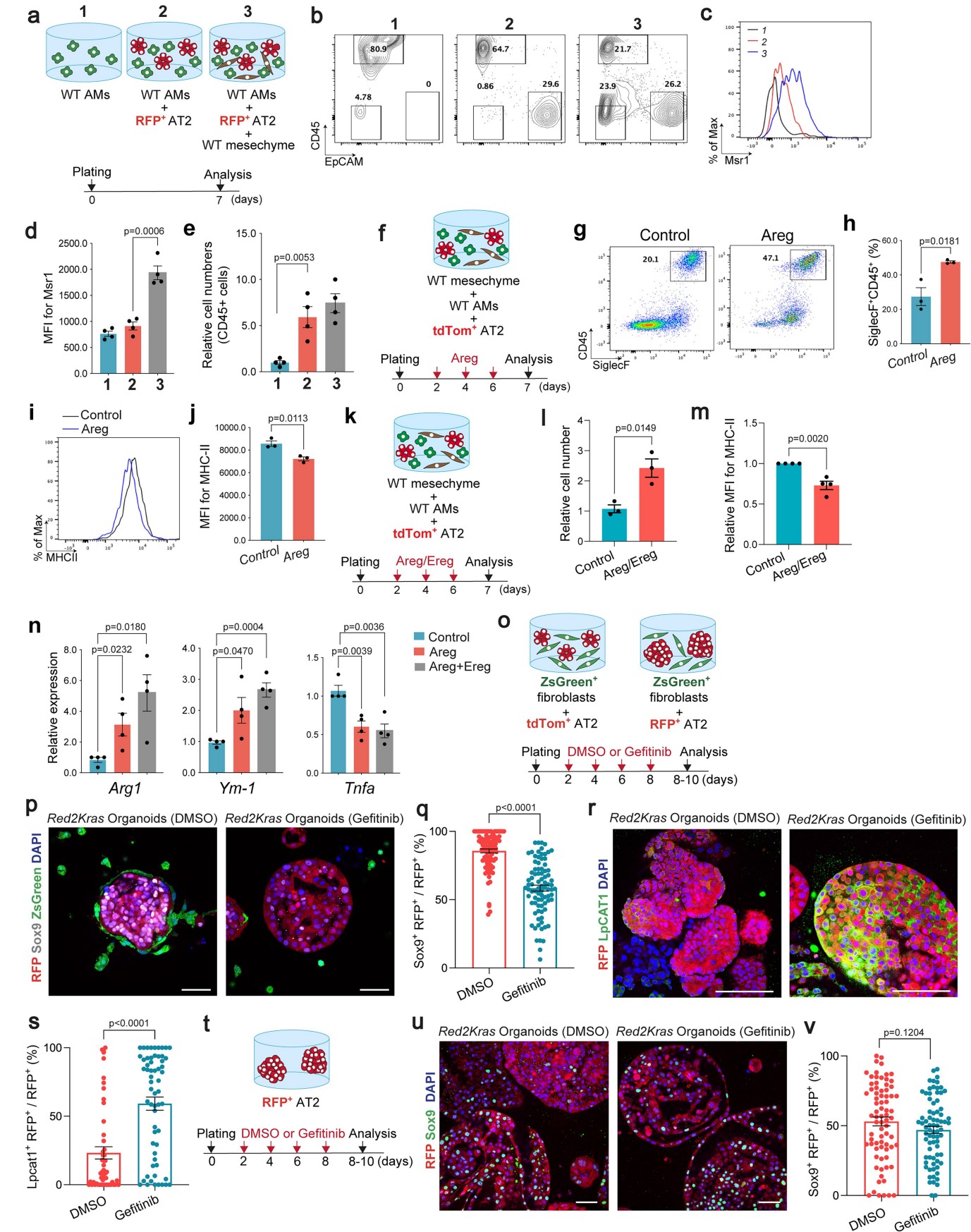

**Extended Data Fig. 7** | See next page for caption.

**Extended Data Fig. 7 | Areg–EGFR signalling mediates reciprocal reprogramming between fibroblasts and mutant AT2 cells. a**, Schematic illustrating 3D organoid co-cultures of wildtype AMs and mesenchymal cells with lineage-labelled RFP$^+$ cells isolated *Red2Kras* lungs. **b**, Representative flow cytometry plots showing the populations of macrophages (CD45$^+$EpCAM$^-$), tumour cells (CD45$^-$EpCAM$^+$), and mesenchymal cells (CD45$^-$EpCAM$^-$) gated from CD31$^-$ populations and stablished in (**a**). **c**, Flow cytometry analysis of Msr1 expression in AMs from (**a**). **d**, **e**, Quantification of the MFI for Msr1 (**d**) and relative cell numbers (**e**) of AMs obtained from (**a**). Data are mean ± s.e.m. Each dot represents one independent experiment. 1 (*n* = 4), 2 (*n* = 4) and 3 (*n* = 4). *P* values were calculated using two-tailed unpaired t-test. **f**, Experimental scheme to investigate the effect of Areg on AMs co-cultured with wildtype mesenchymal and AT2 cells. **g**, Flow cytometry analysis showing the proportion of AMs from (**f**). **h**, Quantification of frequency of AMs from (**f**). Data are mean ± s.e.m. Each dot represents one independent experiment. Control (*n* = 3) and Areg (*n* = 3). *P* values were calculated using two-tailed unpaired t-test. **i**, Flow cytometry analysis showing MHCII expression in AMs from (**f**). **j**, Quantification of the MFI for MHCII in AMs from (**f**). Data are mean ± s.e.m. Each dot represents one independent experiment. Control (*n* = 3) and Areg (*n* = 3). *P* values were calculated using two-tailed unpaired t-test. **k**, Experimental scheme to investigate the effect of Areg/Ereg on AMs co-cultured with wildtype mesenchymal and AT2 cells. **l**, Quantification of the relative cell numbers of AMs from (**k**). Data are mean ± s.e.m. Each dot represents one independent experiment. Control (*n* = 3) and Areg/Ereg (*n* = 3). *P* values were calculated using two-tailed unpaired t-test. **m**, Quantification of MFI for MHCII in AMs from (**k**). Data are mean ± s.e.m. Each dot represents one independent experiment. Control (*n* = 4) and Areg/Ereg (*n* = 4). *P* values were calculated using two-tailed unpaired t-test. **n**, qPCR analysis of *Arg1*, *Ym-1*, and *Tnfa* expression in AMs co-cultured with *wildtype* mesenchyme and AT2 cells treated with PBS, Areg alone, or Areg/Ereg. Data are mean ± s.e.m. Each dot represents one independent experiment. Control (*n* = 4), Areg (*n* = 4) and Areg/Ereg (*n* = 4). *P* values were calculated using two-tailed unpaired t-test. **o**, Schematic illustrating 3D organoid co-cultures of fibroblasts isolated from *Pdgfra-Cre$^{ERT2}$; ZsGreen* lungs with lineage-labelled cells isolated from *Sftpc-Cre$^{ERT2}$;tdTomato* or *Red2Kras* lungs. Gefitinib or DMSO was treated for the indicated duration. **p**, Representative confocal images showing lineage-labelled DATP-like mutant cells in 3D organoid co-cultures from (**o**). DAPI (blue), ZsGreen (green), Sox9 (grey), and RFP (red, tumour). Scale bar, 50 μm. **q**, Quantification of the percentage of Sox9 expressing cells within RFP$^+$ mutant organoids assessed in (**p**). Data are presented as mean ± s.e.m. Each dot represents one organoid from four independent experiments. DMSO (*n* = 4), Gefitinib (*n* = 4). *P* values were calculated using Mann–Whitney test. **r**, Representative confocal images showing lineage-labelled cells in 3D organoid co-cultures with *wild-type* mesenchymal cells and treated with Gefitinib or DMSO. DAPI (blue), LpCat1 (green) and RFP (red, tumour). Scale bar, 100 μm. **s**, Quantification of the percentage of LpCat1 expressing cells within RFP$^+$ mutant organoids. Data are presented as mean ± s.e.m. Each dot represents one organoid from three independent experiments. DMSO (*n* = 3), Gefitinib (*n* = 3). *P* values were calculated using Mann–Whitney test. **t**, Schematic illustrating 3D organoid cultures of mutant AT2 cells isolated from *Red2Kras* lungs. Gefitinib or DMSO was treated for the indicated duration. **u**, Representative confocal images showing the expression of DATP-like cell state marker in *Red2Kras* RFP$^+$ mutant organoids from (**t**). DAPI (blue), Sox9 (green) and RFP (red, tumour). Scale bar, 50 μm. **v**, Percentage of DATP-like cell states expressing Sox9 in *Red2Kras* RFP$^+$ mutant organoids from (**u**). Data are presented as mean ± s.e.m.; Each dot represents an individual organoid from four independent experiments. DMSO (*n* = 4), Gefitinib (*n* = 4). *P* values were calculated using two-tailed unpaired t-test.

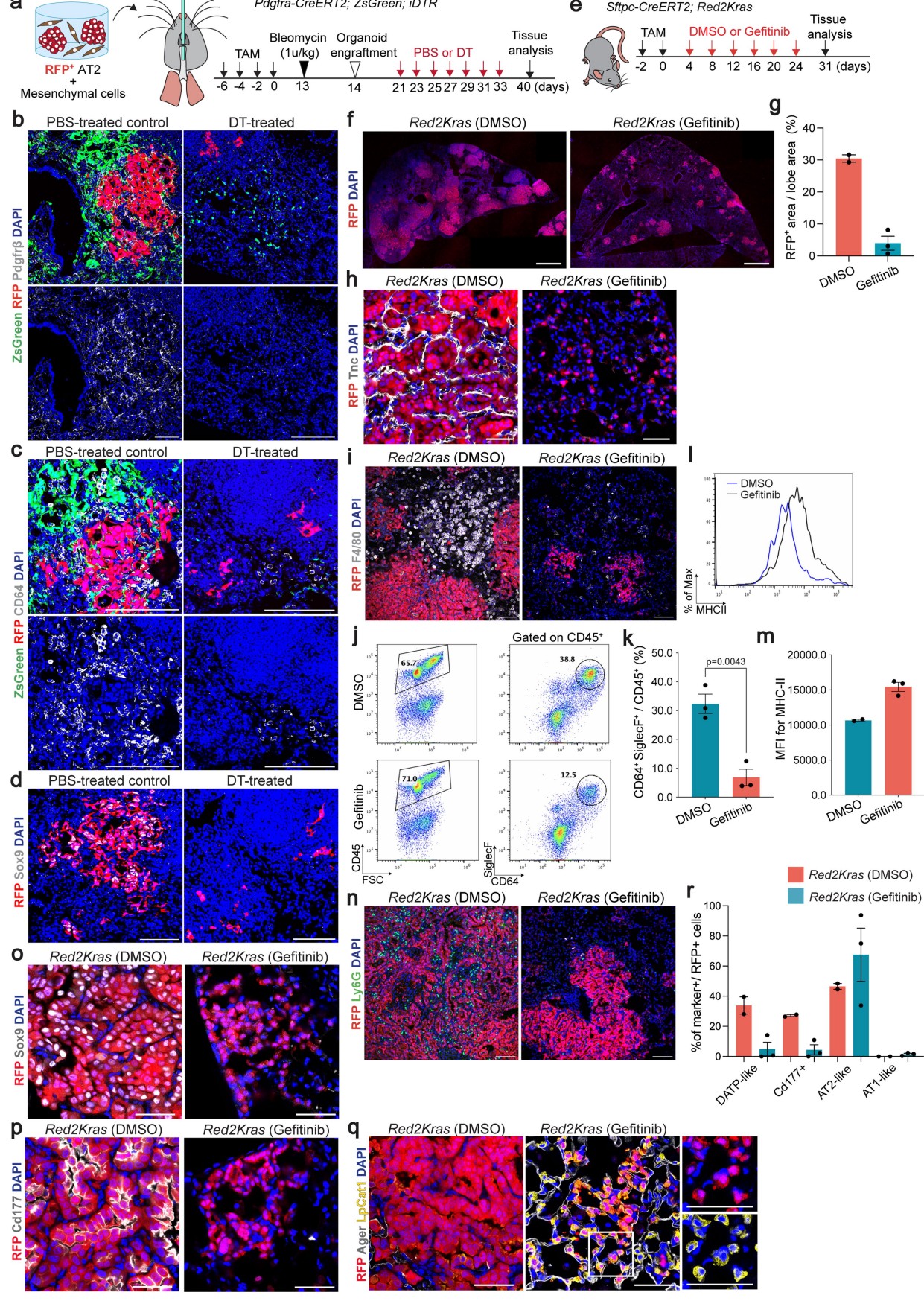

**Extended Data Fig. 8** | See next page for caption.

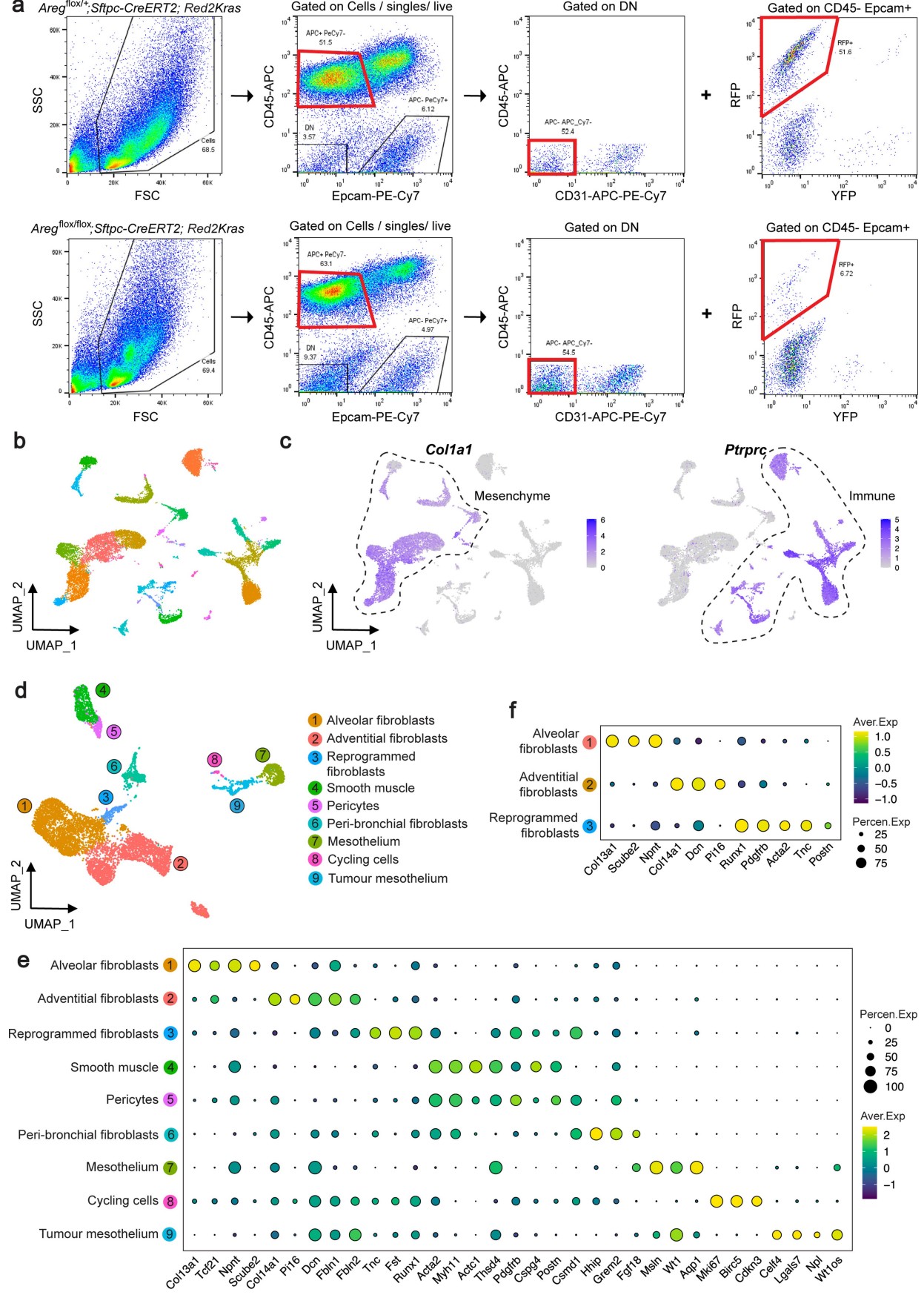

**Extended Data Fig. 8 | EGFR inhibition disrupts sustained mutant epithelial states. a**. Experiment scheme of orthotopic engraftment of tumour organoids into the lungs of *Pdgfra-Cre^ERT2^;ZsGreen;iDTR* mice. **b-d**. Representative confocal images for fibrotic fibroblasts (**b**), macrophage (**c**) and mutant epithelial cells (**d**) in PBS-treated and DTR-treated lungs. DAPI (blue), Pdgfrβ (grey), ZsGreen (green, lineage-traced fibroblasts) and RFP (red, tumour) (**b**). DAPI (blue), Cd68 (grey), ZsGreen (green, lineage-traced fibroblasts) and RFP (red, tumour) (**c**). DAPI (blue), Sox9 (grey), and RFP (red, tumour) (**d**). Images are representative of *n* = 2 mice. Scale bar, 100 μm. **e**, Experimental design for EGFR tyrosine kinase inhibitor administration. *Red2Kras* mice received two doses of tamoxifen and were treated with DMSO or Gefitinib (80 mg/kg) at the indicated timepoints. Lung tissues were collected for histological analyses. **f**, Representative tile scans of entire lung lobe cross-sections from *Red2Kras* animals treated with DMSO or Gefitinib. DAPI (blue) and RFP (red, tumour). Scale bar, 1,000 μm. **g**, Percentage of RFP⁺ mutant cell area relative to lobe area assessed in (**f**). Data are as mean ± s.e.m. Each dot represents a biologically independent mouse replicate. DMSO (*n* = 2), Gefitinib (*n* = 3). **h,i** Representative confocal images for reprogrammed fibroblasts, macrophages and lineage-labelled cells in DMSO or Gefitinib-treated lungs. DAPI (blue), Tnc (grey) and RFP (red, tumour) (**h**). Images representative of *n* = 2 (DMSO) and *n* = 2 (Gefitinib) mice. DAPI (blue), F4/80 (grey) and RFP (red, tumour) (**i**). Images representative of *n* = 2 (DMSO) and *n* = 3 (Gefitinib) mice. Scale bar, 50 μm. **j**, Flow cytometry analysis for AMs from *Red2Kras* lungs treated with DMSO or Gefitinib. **k**, Quantification of the percentage of AMs from DMSO- or Gefitinib-treated *Red2Kras* lungs. Data are mean ± s.e.m. Each dot represents a biologically independent mouse replicate. DMSO (*n* = 3) and Gefitinib (*n* = 3). *P* values were calculated using two-tailed unpaired t-test. **l**, Flow cytometry analysis for MHC-II expression on AMs from DMSO- or Gefitinib-treated *Red2Kras* lungs. **m**, Quantification of the MFI for MHC-II in AMs from DMSO- or Gefitinib-treated *Red2Kras* lungs. Data are mean ± s.e.m. Each dot represents a biologically independent mouse replicate. DMSO (*n* = 2) and Gefitinib (*n* = 3). **n**, Representative confocal images for neutrophils and lineage-labelled cells in DMSO- or Gefitinib-treated *Red2Kras* lungs. DAPI (blue), Ly6G (green) and RFP (red, tumour). Images representative of *n* = 3 biologically independent mice. Scale bar, 50 μm. **o-q**, Representative confocal images of the different epithelial states in DMSO- or Gefitinib-treated *Red2Kras* lungs. DAPI (blue), Sox9 (grey) and RFP (red, tumour) (**o**). DAPI (blue), Cd177 (grey) and RFP (red, tumour) (**p**). DAPI (blue), Ager (grey), LpCat1 (yellow) and RFP (red, tumour) (**q**). Scale bar, 50 μm. **r**, Quantification of the percentage of each cell state within the RFP⁺ mutant cell population in DMSO- or Gefitinib-treated *Red2Kras* lungs. Data are mean ± s.e.m. Each dot represents a biologically independent mouse replicate. DMSO (*n* = 2), Gefitinib (*n* = 3).

**Extended Data Fig. 9** | See next page for caption.

**Extended Data Fig. 9 | Transcriptional landscape of lung mesenchymal cells upon mutant cell-specific *Areg* deletion. a**, FACS gating strategy for isolating mesenchymal (CD45⁻CD31⁻EpCAM⁻), immune (CD45⁺EpCAM⁻), and lineage-labelled RFP⁺ mutant cells from *Areg*$^{flox/+}$ and *Areg*$^{flox/flox}$ lungs for scRNA-seq. Each sorted population, indicated by red boxes, was used for library generation. **b**, UMAP plot showing all immune and mesenchymal cell populations obtained after integration of *Areg*$^{flox/+}$ and *Areg*$^{flox/flox}$ datasets. Cells are coloured by cell cluster. **c**, Feature plot showing the expression of *Col1a1* (mesenchymal cells) and *Ptprc* (immune cells) in the UMAP from (**b**).

**d**, UMAP plot showing all sub-clustered mesenchymal cells, annotated into the nine identified population in the dataset. Cells are coloured and numbered by population. **e**, Dotplot showing the expression of key marker genes for mesenchymal cells annotated in (**d**). Each dot is coloured based on the average expression and sized based on the percentage of cells expressing that gene. **f**, Dotplot showing the expression of key marker genes for the sub-clustered fibroblast annotated in Fig. 4d. Each dot is coloured based on the average expression and sized based on the percentage of cells expressing that gene.

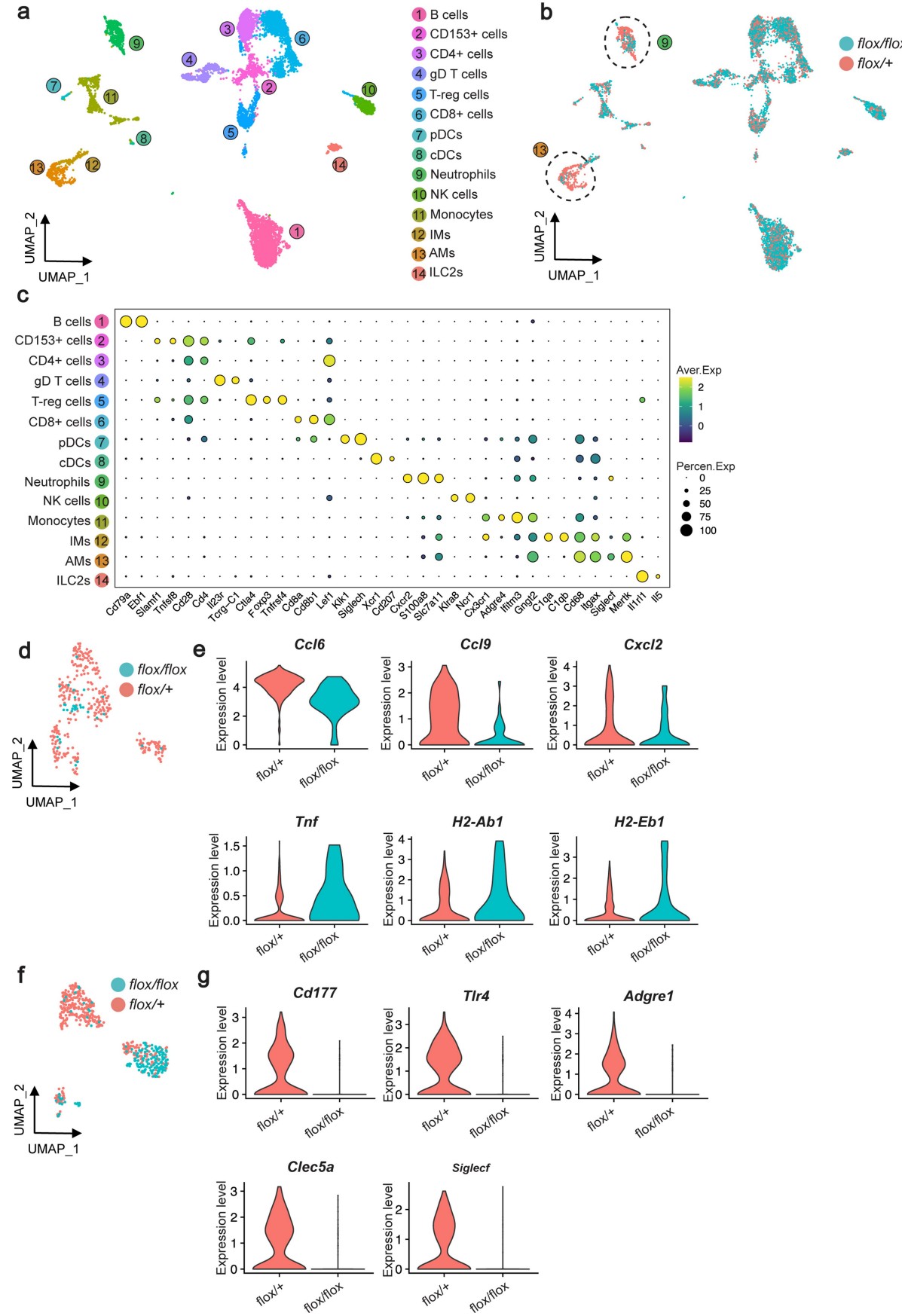

**Extended Data Fig. 10 | Transcriptional landscape of lung immune cells upon mutant cell-specific *Areg* deletion. a**, UMAP plot showing all sub-clustered immune cells, annotated into the fourteen identified populations in the dataset. Cells are coloured and numbered by population. **b**, UMAP plot showing the distribution of immune cells in *Areg*<sup>flox/+</sup> and *Areg*<sup>flox/flox</sup> datasets. Cells are coloured according to their dataset of origin. **c**, Dotplot showing the expression of key marker genes for immune cells annotated in (**a**). Each dot is coloured based on the average expression and sized based on the percentage of cells expressing that gene. **d**, UMAP plot showing the sub-clustering and distribution of AMs in *Areg*<sup>flox/+</sup> and *Areg*<sup>flox/flox</sup> datasets. Cells are coloured according to their dataset of origin. **e**, Violin plots comparing the expression of pro-inflammatory cytokines and MHC-II complex components in AMs between *Areg*<sup>flox/+</sup> and *Areg*<sup>flox/flox</sup> lungs. **f**, UMAP plot showing the distribution of neutrophils in *Areg*<sup>flox/+</sup> and *Areg*<sup>flox/flox</sup> lungs. Cells are coloured according to dataset of origin. **g**, Violin plots showing increased expression of genes associated with neutrophil activation in neutrophils from *Areg*<sup>flox/+</sup> lungs compared to *Areg*<sup>flox/flox</sup> lungs.

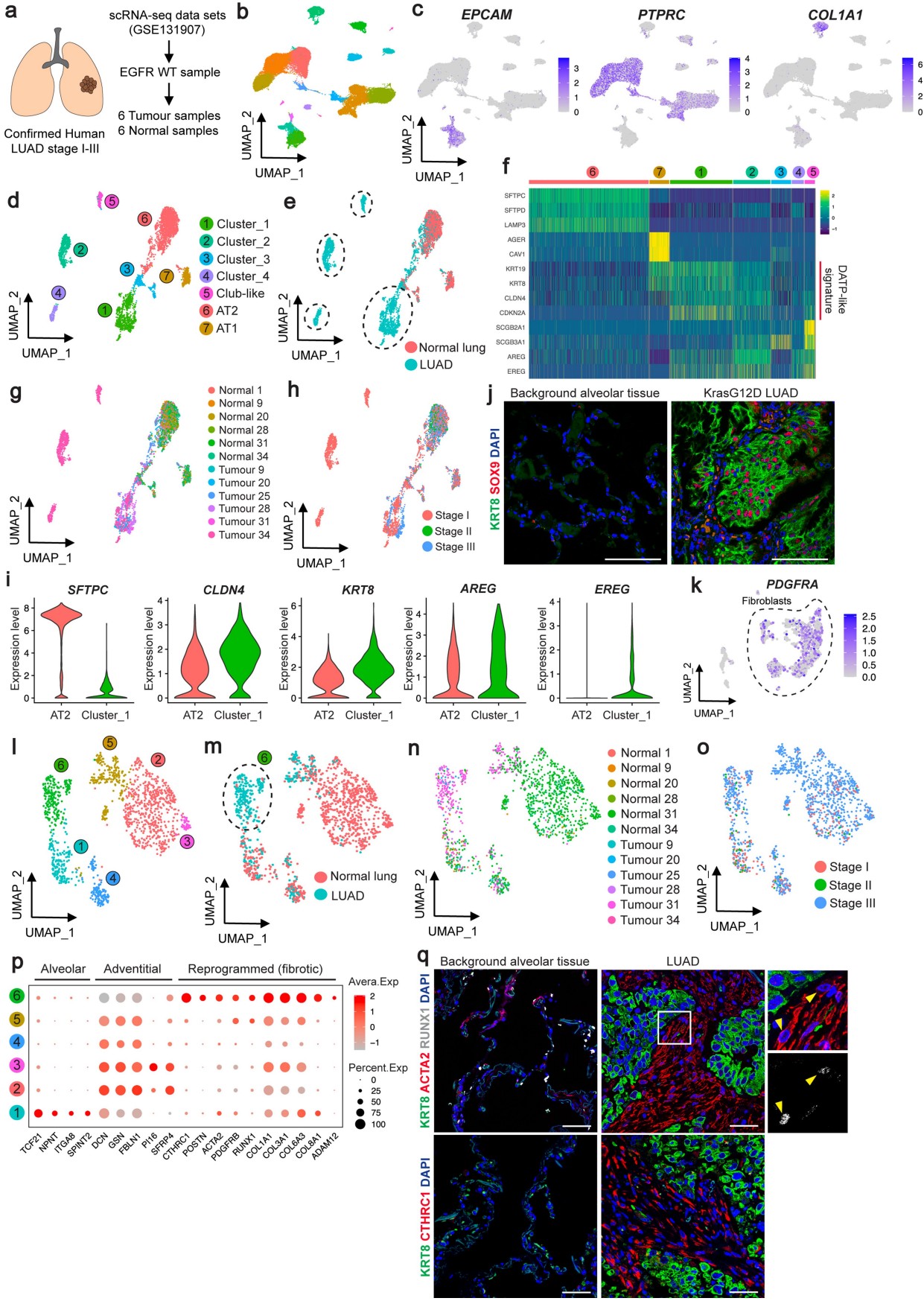

**Extended Data Fig. 11** | See next page for caption.

**Extended Data Fig. 11 | Early-stage human lung adenocarcinoma reveals distinct transcriptional reprogramming and fibrotic niche. a**, Schematic of the workflow for analysing the single-cell transcriptomics dataset of early-stage (I-III) human lung adenocarcinoma (LUAD) patient samples[45]. Patients with confirmed EGFR wildtype status were selected for analysis, with a total of six LUAD and six distant normal lung tissue samples (5 matched pairs). **b**, UMAP plot showing the integrated single-cell transcriptomics datasets of early-stage (I-III) human lung adenocarcinoma (LUAD) patient and matched distant normal lung tissue samples (5 matched pairs, 1 LUAD-only, and 1 normal-only sample). Cells are coloured by population. **c**, UMAP feature plots showing the expression of epithelial (*EPCAM*), immune (*PTPRC*) and mesenchymal (*COL1A1*) marker genes in the integrated UMAP from (**b**). **d**, UMAP plots showing all sub-clustered epithelial cells annotated into the seven distinct populations, colour-coded by cellular states. **e**, UMAP plot showing the distribution of epithelial cells, colour-coded by the group of datasets of origin (LUAD vs normal). **f**, Heatmap displaying expression of key marker genes distinguishing each cluster annotated in (**d**). **g**, UMAP plot showing the distribution of epithelial cells in normal and LUAD lung tissues. Cells are coloured according to dataset of origin (a total of twelve samples: six LUAD and six normal patient tissue samples). **h**, UMAP plot showing the distribution of epithelial cells in normal and LUAD lung tissues. Cells are coloured according to LUAD stage of dataset of origin. **i**, Violin plots depicting the expression of DATP-like cell marker genes and EGFR ligands in AT2 cells and cluster 1. **j**, Representative confocal images showing SOX9[+] intermediate epithelial cell states in early-stage KRAS[G12D] LUAD patient tissues. DAPI (blue), KRT8 (green) and SOX9 (red). Scale bar, 100 μm **k**, UMAP feature plot showing the expression of fibroblast marker gene *PDGFRA* in the UMAP of human mesenchymal cells. **l**, UMAP plot showing all sub-clustered fibroblasts annotated into six distinct populations. Cells are coloured and numbered by population. **m**, UMAP plot showing the distribution of fibroblasts in normal and LUAD lung tissues. Cells are coloured according to dataset of origin (LUAD vs normal). **n**, UMAP plots showing all sub-clustered fibroblasts colour-coded by dataset of origin. **o**, UMAP plot showing the distribution of fibroblasts, colour-coded by LUAD stage of dataset origin. **p**, Dotplot showing the expression of key fibroblast marker genes in each cluster annotated in (**l**). Each dot is coloured based on the average expression and sized based on the percentage of cells expressing that gene. **q**, Representative confocal images showing reprogrammed fibroblasts and DATP-like intermediate cells in matched background and LUAD patient tissues. DAPI (blue), RUNX1 (grey), ACTA2 (red), and KRT8 (green) (top panel). DAPI (blue), CTHRC1 (red), and KRT8 (green) (bottom panel). Yellow arrows highlight reprogrammed fibroblasts in close proximity to KRT8[+] intermediate cells. Scale bar, 50 μm.

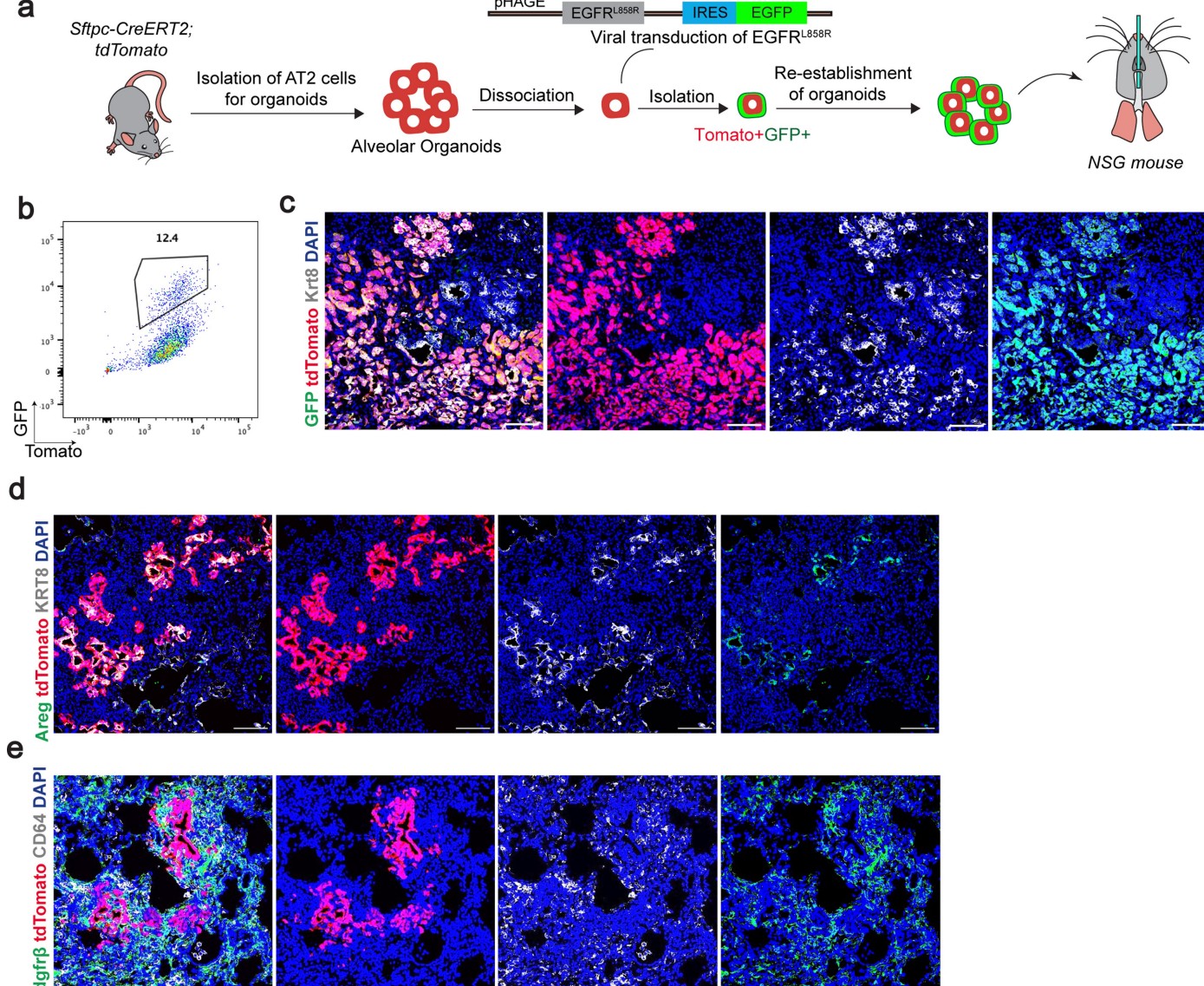

**Extended Data Fig. 12 | Establishment of pre-neoplastic niches driven by EGFR-mutant cells. a**. Experiment scheme of the 3D inducible LUAD model established by introducing EGFR^L858R mutation into primary AT2 cells, followed by orthotopic engraftment into NSG mouse lungs. **b**, Flow cytometry analysis showing the AT2 cells transduced with EGFR^L858R mutation. **c**, Representative confocal images showing the engraftment of *tdTom⁺EGFP⁺* mutant organoids. DAPI (blue), tdTomato (red), GFP (green) and Krt8 (grey). Images representative of *n = 2* mice. Scale bar, 100 μm. **d,e**, Representative confocal images showing Areg expression in Krt8⁺ epithelial cell states (**d**) and the associated remodelling of fibroblasts and macrophages surrounding tumour cells (**e**). DAPI (blue), tdTomato (red), Areg (green) and Krt8 (grey) (**d**). DAPI (blue), tdTomato (red), Pdgfrβ (green) and Cd68 (grey) (**e**). Images representative of *n = 2* mice. Scale bar, 100 μm.

# Reporting Summary

## Statistics

For all statistical analyses, confirm that the following items are present in the figure legend, table legend, main text, or Methods section.

| n/a | Confirmed | |
|---|---|---|
| ☐ | ☒ | The exact sample size (*n*) for each experimental group/condition, given as a discrete number and unit of measurement |
| ☐ | ☒ | A statement on whether measurements were taken from distinct samples or whether the same sample was measured repeatedly |
| ☐ | ☒ | The statistical test(s) used AND whether they are one- or two-sided<br>*Only common tests should be described solely by name; describe more complex techniques in the Methods section.* |
| ☒ | ☐ | A description of all covariates tested |
| ☐ | ☒ | A description of any assumptions or corrections, such as tests of normality and adjustment for multiple comparisons |
| ☐ | ☒ | A full description of the statistical parameters including central tendency (e.g. means) or other basic estimates (e.g. regression coefficient) AND variation (e.g. standard deviation) or associated estimates of uncertainty (e.g. confidence intervals) |
| ☐ | ☒ | For null hypothesis testing, the test statistic (e.g. *F*, *t*, *r*) with confidence intervals, effect sizes, degrees of freedom and *P* value noted<br>*Give P values as exact values whenever suitable.* |
| ☒ | ☐ | For Bayesian analysis, information on the choice of priors and Markov chain Monte Carlo settings |
| ☒ | ☐ | For hierarchical and complex designs, identification of the appropriate level for tests and full reporting of outcomes |
| ☒ | ☐ | Estimates of effect sizes (e.g. Cohen's *d*, Pearson's *r*), indicating how they were calculated |

*Our web collection on statistics for biologists contains articles on many of the points above.*

## Software and code

Policy information about availability of computer code

| Data collection | Leica Application Suite X (Confocal microscopy; version 2.0.0)<br>BD FACS Diva (Flow Cytometry 8.0.1)<br>FASTQ files of scRNA-seq data generated on the 10X Chromium platform were processed using the standard Cell Ranger pipeline. |
|---|---|
| Data analysis | Fiji/ImageJ (for image processing and analysis; 1.54p)<br>FlowJo (for Flow cytometry analysis; FlowJo v10.8.1)<br>GraphPad Prism (v.10.4.0 and v7.0)<br>R studio (for scRNA-seq analysis; version 4.4.2)<br>CellRanger (v.6.0.2; v7.2.0 and v8.0.0)<br>Scanpy pipeline (v.1.9.1)<br>Seurat package (v.5) |

For manuscripts utilizing custom algorithms or software that are central to the research but not yet described in published literature, software must be made available to editors and reviewers. We strongly encourage code deposition in a community repository (e.g. GitHub). See the Nature Portfolio guidelines for submitting code & software for further information.

## Data

Policy information about availability of data

All manuscripts must include a data availability statement. This statement should provide the following information, where applicable:

- Accession codes, unique identifiers, or web links for publicly available datasets
- A description of any restrictions on data availability
- For clinical datasets or third party data, please ensure that the statement adheres to our policy

Single-cell RNA sequencing datasets have been deposited in the Gene Expression Omnibus (GEO) under the following accession numbers: human alveolar organoids (GSE310335); mesenchymal (GSE316241) and immune (GSE316243) cells from Confetti and Red2Kras lungs; and tumours and stromal cells from Aregflox/+ and Aregflox/flox lungs (GSE316244). Additionally, publicly available datasets were used in this study and re-analyzed: Bleomycin injury scRNA-seq dataset (GSE132771); Human LUAD scRNA-seq dataset (GSE131907)

## Research involving human participants, their data, or biological material

Policy information about studies with human participants or human data. See also policy information about sex, gender (identity/presentation), and sexual orientation and race, ethnicity and racism.

| | |
|---|---|
| Reporting on sex and gender | All human specimens were obtained in de-identified form, and no donor sex or gender information was provided |
| Reporting on race, ethnicity, or other socially relevant groupings | No race, ethnicity or other socially constructed or socially relevant categorization was used in this manuscript. |
| Population characteristics | Samples included non-diseased background lung tissue and early-stage lung adenocarcinoma specimens as specified in the relevant figure legends. |
| Recruitment | No participants were recruited for this study. |
| Ethics oversight | All human specimens were obtained and approved under Research Tissue Bank Generic REC approval (Tissue Bank Project number T02233), MSKCC Institutional Review Board approval (IRB # 12-245), and Severance Hospital Review Board approval (IRB # 4-2019-0447, 4-2012-0685 and 4-2013-0770). |

Note that full information on the approval of the study protocol must also be provided in the manuscript.

# Field-specific reporting

Please select the one below that is the best fit for your research. If you are not sure, read the appropriate sections before making your selection.

☒ Life sciences ☐ Behavioural & social sciences ☐ Ecological, evolutionary & environmental sciences

For a reference copy of the document with all sections, see nature.com/documents/nr-reporting-summary-flat.pdf

# Life sciences study design

All studies must disclose on these points even when the disclosure is negative.

| | |
|---|---|
| Sample size | Sample size for animal and organoid experiments was made as large as possible and sufficient to determine statistical significance. For most experiments, a minimum of N=3 of biological replicates was used (except when stated otherwise in the method section of respective figure legends). |
| Data exclusions | No animals were excluded from the statistical analysis |
| Replication | For each experiment, several replicates were used (and stated in the respective figure legends). |
| Randomization | For all experiments both female and male mice were used. Mice were randomly assigned to experimental and control groups when possible For in vitro experiments, individual wells were randomly assigned as control or treated for all experiments. |
| Blinding | Blinding of both animal and organoid experiments was challeging since the same investigator was responsible for performing the experiment, collecting and processing the samples and analyze the data. To decrease unwanted biases whenever possible, mice samples were analyzed before confirming their experimental group or genotype. However, blinding was not performed when treatment effects on tumour volume were readily distinguishable between groups. |

# Reporting for specific materials, systems and methods

We require information from authors about some types of materials, experimental systems and methods used in many studies. Here, indicate whether each material, system or method listed is relevant to your study. If you are not sure if a list item applies to your research, read the appropriate section before selecting a response.

## Materials & experimental systems

| n/a | Involved in the study |
|---|---|
| ☐ | ☒ Antibodies |
| ☐ | ☒ Eukaryotic cell lines |
| ☒ | ☐ Palaeontology and archaeology |
| ☐ | ☒ Animals and other organisms |
| ☒ | ☐ Clinical data |
| ☒ | ☐ Dual use research of concern |
| ☒ | ☐ Plants |

## Methods

| n/a | Involved in the study |
|---|---|
| ☒ | ☐ ChIP-seq |
| ☐ | ☒ Flow cytometry |
| ☒ | ☐ MRI-based neuroimaging |

# Antibodies

Antibodies used

For FACS:
Anti-mouse CD326 (EpCAM); PE-Cy7; (Biolegend #118216); clone G.8.8
Anti-mouse CD45; APC; (BD Biosciences #559864); clone 30-F11
Anti-mouse CD31; APC; (BD Biosciences #551262); clone MEC 13.3
Anti-mouse CD31; APC-Cy7; (Biolegend #102533); clone MEC 13.3
Anti-mouse SiglecF; BV421; (Biolegend #155509); clone S17007L
Anti-mouse SiglecF; PE; (Biolegend #155505); clone S17007L
Anti-mouse CD64; PE; (Biolegend #139303); clone x54-5/7.1
Anti-mouse MHC-II; FITC; (eBioscience #11-5321-81); clone M5/114.15.2
Anti-mouse CD11b; Biotin; (Biolegend #101203); clone M1/70
Anti-mouse Ly6G; PE; (Biolegend #101207); clone 1A8
Anti-mouse Gl3; PE; (Biolegend #118107); clone LG.3A10
Anti-mouse CD204; APC; (Biolegend #154711); clone 1F8C33
Anti-human HTII-280 IgM; (Terrace Biotech # TB-27AHT2-280);
Anti-human CD45; APC; (Biolegend #368512); clone 2D1
Anti-human EpCAM; FITC; (Biolegend #324204); clone 9C4

For IF staining in mouse and Human tissue samples (primary antibodies):
Anti-Ager; Rat; (R&D Systems #MAB1179); clone 175410
Anti-Itga2; Rabbit; (Abcam #ab181548); clone EPR17338
Anti-cytokeratin 8 (Krt8); Rat; (DSHB #TROMA-I)
Anti-prosurfactant protein C; Rabbit; (Millipore #AB3786)
Anti-Sox9; Rabbit; (Abcam #Ab185230); clone EPR14335
Anti-α smooth muscle actin; Mouse; (Sigma #A5228); clone 1A4
Anti-α smooth muscle actin; Mouse; (R&D Systems #MAB1420); clone 1A4
Anti-Runx1/AML1; Rabbit; (Cell Signaling #8529); clone D4A6
Anti-Pdgfrβ; Rabbit; (Cell Signaling #3169); clone 28E1
Anti-F4/80; Rat; (Bio-rad #MCA497GA); clone A3-1
Anti-Ly6g; Rat; (Biolegend #127601/127605); clone 1A8
Anti-Gl3; Rat; (Biolegend #118101); clone GL3
Anti-Pdgfrα; Rabbit; (Cell Signaling #3174); clone D1E1E
Anti-Lipocalin-2/NGAL; Goat; (R&D Systems #AF1857)
Anti-Amphiregulin; Goat; (R&D Systems #AF989)
Anti-Amphiregulin; Rabbit; (Proteintech #16036-1-AP)
Anti-CTHRC1; Rabbit; (MaineHeatlh #Vli55); clone Vli55
Anti-Ki67; Rat; (Thermo Fisher #14-5698-82); clone S01A15
Anti-Tenascin C; Rat; (R&D Systems #MAB2138-SP); clone 578
Anti-MSR1; Rabbit; (Cell Signaling #91119T); clone E4H1C
Anti-CD68/SR-D1; Rabbit; (R&D Systems #MAB101141-SP); clone 2449D
Anti-LpCat1; Rabbit; (Proteintech #16112-1-AP);
Anti-CD177; Rabbit; (R&D Systems #MAB8186); clone 1171A
Anti-CD11c; Hamster; (BioLegend #117301); clone N418
Anti-Foxp3; Rabbit; (Cell Signaling #12653); clone D608R
Anti-CD64; Rat ; (BioLegend #161002); clone S18017D
Anti-Slc7a11; Rabbit; (ThermoFisher #PA1-16893); clone PA1-16893
Anti-MHCII; Rat; (ThermoFisher #16-5321-81); clone M5/114.15.2
Anti-PD-L1; Rabbit; (R&D Systems #MAB90781-100); clone 2096C

For IF staining in mouse and Human tissue samples (secondary antibodies):
Donkey anti-rat Alexa Fluor™ 647; (Invitrogen #A48272)
Donkey anti-rabbit Alexa Fluor™ 647; (Invitrogen #A31573)
Donkey anti-goat Alexa Fluor™ 647; (Invitrogen #A21447)
Donkey anti-mouse Alexa Fluor™ 647; (Invitrogen #A-31571)
Donkey anti-rabbit DyLight™ 755; (Invitrogen #SA5-10043)
Donkey anti-rat DyLight™755; (Invitrogen #SA5-10031)

Donkey anti-Rabbit Alexa Fluor™ Plus 405; (Invitrogen #A48258)
Donkey anti-Goat Alexa Fluor™ Plus 405; (Invitrogen #A48259)
Goat anti-Armenian Hamster Alexa Fluor® 647; (Jackson ImmunoResearch  #127-605-099)

| | |
|---|---|
| Validation | All antibodies were tested and validated by the manufactor (details are specified in supplier webpages and are accessible using catalog numbers indicated for each antibody above). Antibodies which were not previously validated in mouse or Human tissue were tested for this study and data presented in the submitted manuscript. |

Anti-Ager: validated for Immunohistochemistry, Western Blot. Tested reactivity: Mouse, Rat
Anti-Itga2: validated for use in Western Blot, Flow Cytometry, Flow Cytometry, Immunoprecipitation, Immunohistochemistry, Immunocytochemistry/immunofluorescence. KO validated for confirmed specificity. Tested reactivity: Human, Mouse, Rat.
Anti-Krt8: specificity for keratin K8 has been knock-out validated. Tested reactivity: Canine, Human, Mouse
Anti-Sox9: validated for use in Flow Cyt (Intra), ICC/IF, IHC-P, IP, WB. Tested reactivity: Human, mouse, rat samples.
Anti-a-smooth actin: Multiplex Immunofluorescence, Immunohistochemistry, Western Blot, Intracellular Staining by Flow Cytometry, Dual RNAscope ISH-IHC Compatible, Immunocytochemistry, Simple Western, CyTOF-ready; Tested reactivity: Human, Mouse, Rat
Anti-Runx1: Western Blotting. Tested reactivity: Human, Mouse.
Anti-Pdgfrβ: Western Blotting, Simple Western™, Immunoprecipitation, Immunohistochemistry (Paraffin), Immunofluorescence (Frozen), Immunofluorescence (Immunocytochemistry). Tested reactivity: Human, Mouse, Rat. This antibody may cross-react with PDGF receptor α when highly overexpressed. Nonspecific labeling in fixed frozen mouse colon has been observed by immunofluorescence.
Anti-F4/80: This product has been reported to work in the following applications. This information is derived from testing within our laboratories, peer-reviewed publications or personal communications from the originators: Flow Cytometry, Immuno-electron Microscopy, Immunofluorescence, Immunohistology, Western Blotting and Radioimmunoassays. Tested reactivity: Mouse.
Anti-Ly6g: Each lot of this antibody is quality control tested by immunofluorescent staining with flow cytometric analysis. Tested reactivity: Mouse.
Anti-Gl3: The GL3 antibody has been shown to be useful in identifying γ/δ T cells by flow cytometry and immunohistochemistry and depleting γ/δ T cells in vivo. Tested reactivity: Mouse.
Anti-Pdgfrα: Western Blotting, Immunoprecipitation, Immunohistochemistry (Paraffin), Immunofluorescence (Immunocytochemistry), Flow Cytometry (Fixed/Permeabilized). Tested reactivity: Human, Mouse.
Anti-Lipocalin-2/NGAL: Immunohistochemistry, Western Blot, Simple Western, Immunoprecipitation; Tested reactivity: Mouse.
Anti-Amphiregulin (mouse): Immunohistochemistry, Western Blot, ELISA Capture (Matched Antibody Pair), Neutralization. Tested reactivity: Mouse, Guinea Pig, Transgenic Mouse
Anti-Amphiregulin (Human): KD/KO validated. WB, IHC, IF/ICC, IF-P, FC (Intra), IP, ELISA. Tested reactivity: human, mouse.
Anti-Cthrc1: The antibody was raised against the whole molecule of human Cthrc1. It cross-reacts with rat and mouse Cthrc1. Western-blot, ELISA, IP.
Anti-Ki67: Immunohistochemistry (Paraffin) (IHC (P)), Immunohistochemistry (Frozen) (IHC (F)), Immunocytochemistry (ICC/IF). Dog, Cynomolgus. Tested reactivity: monkey, Human, Mouse, Non-human primate, Rat.
Anti-Tenascin C: Western Blot, Neutralization, Immunocytochemistry. Tested reactivity: Human, Mouse.
Anti-MSR1: Immunofluorescence (Frozen), Immunofluorescence (Immunocytochemistry), Flow Cytometry (Fixed/Permeabilized) and Flow Cytometry (Live). Tested reactivity: Mouse.
Anti-CD68: Multiplex Immunofluorescence, Immunohistochemistry, Intracellular Staining by Flow Cytometry, CyTOF-ready. Tested reactivity: Human, Mouse, Transgenic Mouse.
Anti-LpCat1: KD/KO validated. WB, IHC, IF/ICC, IP, CoIP, ELISA. Tested reactivity: human, mouse, rat.
Anti-CD177: Flow Cytometry; Tested reactivity: mouse.
Anti-Cd11c: Tested reactivity: Mouse; Each lot of this antibody is quality control tested by immunofluorescent staining with flow cytometric analysis.
Anti-Foxp3: IHC Leica Bond, Immunohistochemistry (Paraffin), Immunofluorescence (Frozen) and Flow Cytometry (Fixed/Permeabilized). Tested reactivity: Mouse, Monkey.
Anti-CD64: FC - Quality tested, IHC-F – Verified. Tested reactivity: Mouse and Human.
Anti-Slc7a11: Western Blot (WB), Immunohistochemistry (Paraffin) (IHC (P)), Immunocytochemistry (ICC/IF), Flow Cytometry (Flow), In Situ Hybridization (ISH). Tested reactivity: Human, Mouse, Rat
Anti-MHCII:  Flow Cytometry (Flow), Neutralization (Neu) and Functional Assay (Functional). Tested reactivity: Mouse.
Anti-PD-L1: Immunohistochemistry, Western Blot, Flow Cytometry; Tested reactivity: Mouse

# Eukaryotic cell lines

Policy information about cell lines and Sex and Gender in Research

| | |
|---|---|
| Cell line source(s) | Primary mouse lung cells were derived from wildtype or transgenic mouse lungs in the laboratory. Primary human lung cells were derived from fresh human lung tissues in the laboratory. HEK293T cells were obtained from ATCC (CRL-11268) and maintained according to the supplier's recommendations. |
| Authentication | All cell lines generated in-house were genotyped for authentication. HEK293T cells were not additionally authenticated. |
| Mycoplasma contamination | All cell lines used tested negative for Mycoplasma contamination. |
| Commonly misidentified lines (See ICLAC register) | No. |

# Animals and other research organisms

Policy information about <u>studies involving animals</u>; <u>ARRIVE guidelines</u> recommended for reporting animal research, and <u>Sex and Gender in Research</u>

| | |
|---|---|
| Laboratory animals | Sftpc-CreERT2 (Jax: 028054), R26R-Confetti (Jax: 013731), Pdgfra-CreERT2 (Jax: 032770), R26R-iDTR (Jax: 007900), NOD/Scid Il2rg null Tg (NSGj Jax: 005557), and Ai6/RCL-ZsGreen (Jax: 007906) animals were obtained from The Jackson Laboratory. AregloxP/loxP animals were kindly provided by Prof. Menna Clatworthy from the University of Cambridge, UK. Red2Kras mice were generated inhouse and previously described. CCR2-CreERT2 mice were kindly provided by Prof. Burkhard Becher (University of Zurich). All transgenic mouse strains were maintained on a C57BL or C57BL/6Brd-Tyr 597 c-Brd mixed background. Mice were housed under specific pathogen-free conditions in individually ventilated cages with a 12-hour light/12-hour dark cycle. Ambient temperature was maintained at 20–24 °C with relative humidity of 40–60%, in accordance with institutional animal care guidelines. |
| Wild animals | No wild animals were used in this study |
| Reporting on sex | Both male and female animals were used for all experiments perfomed. |
| Field-collected samples | No filed collected samples |
| Ethics oversight | Mouse studies in the UK were approved under UK Home Office Project Licences PC7F8AE82 and PP3176550, and experiments in the US and Korea were approved by the MSKCC Institutional Animal Care and Use Committee (IACUC) (protocol #24-04-003) and GIST IACUC (protocol # GIST-2022-043). All procedures complied with institutional and national guidelines. |

Note that full information on the approval of the study protocol must also be provided in the manuscript.

# Plants

| | |
|---|---|
| Seed stocks | n/a |
| Novel plant genotypes | n/a |
| Authentication | n/a |

# Flow Cytometry

## Plots

Confirm that:

☒ The axis labels state the marker and fluorochrome used (e.g. CD4-FITC).

☒ The axis scales are clearly visible. Include numbers along axes only for bottom left plot of group (a 'group' is an analysis of identical markers).

☒ All plots are contour plots with outliers or pseudocolor plots.

☒ A numerical value for number of cells or percentage (with statistics) is provided.

## Methodology

| | |
|---|---|
| Sample preparation | For isolation of lung cell, mice were culled by cervical dislocation and lungs cleared of blood via perfusion with 10ml of PBS. Lungs were inflated with 2-3ml of a Dispase solution (Fisher Scientific, 11553550) via intratracheal injection. When isolating mesenchymal cells, Collagenase I (GIBCO, 17100017) was added to the Dispase solution at 350U/ml before inflation. Lungs were carefully dissected out of the thoracic cavity and placed on a petri dish on ice. Individual lobes were separated from each lung, placed in a 50ml falcon and minced into small pieces. Cells were washes down with 3ml of PBS to the bottom of the falcon tube. When isolating epithelial cells only, 60ul of 100mg/ml Collagenase/Dispase solution was added per tube. Samples were places in a shaking incubator at 37ºC, 190 rpm, for 45 min. 7.5ul of 1% DNase I (Sigma, D4527) was added to each sample in the final 10 min of incubation. The resulting cell suspensions were filtered sequentially through 100 µm and 40 µm cell strainers and washed with 2 ml of 10% foetal bovine serum (FBS, 815 Pan-Biotech, P40- 37500) in PBS (PF10) to collect remaining cells. Samples were centrifuged at 800 rpm for 5 min at 4ºC. Supernatant was removed and pellets were resuspended in 1 ml of red blood cell lysis buffer (RBC buffer, made inhouse: 150 mM NH4Cl and 10 mM 567 KHCO3 in distilled H2O) for 60 seconds at RT. After lysis, 6 ml of Dulbecco's Modified Eagle Medium Nutrient Mixture F-12 820 (DMEM/F-12, Invitrogen, 11330057) was added to the tube to neutralize the RBC buffer. 500µL of filtered FBS was added slowly to the bottom of each tube to collect live cells. Tubes were centrifuged again at 800rpm for 5 min at 4ºC. Cell pellets were resuspended in PF10 and placed into separate 1.5 ml tubes for antibody staining. |

| | |
|---|---|
| Instrument | BD Influx™ cell sorter<br>Facsdiscover s8<br>Facsymphony s6 |
| Software | Raw FACS files were obtained from sorter and analysed using FlowJo |
| Cell population abundance | In the lung tissue, EPCAM+ cells comprise approximately 15% of the single cell suspension. For Red2Kras mice, RFP+ cells comprise approximately 10-30% of the epithelial populations (which is dependent on the time point used for collection, dose of tamoxifen, and mouse to mouse variability). |
| Gating strategy | The gating strategy for each experiment and population of interest is detailed in the method sections.<br>To isolate RFP+ mutant cells from Red2Kras animals the following strategy was used: Epcam+/CD45-/CD31- -> RFP+/YFP-.<br>To isolate lineage labelled fibroblasts the following strategy was used: Epcam-/CD45-/CD31- -> ZsGreen+<br>To isolate mesenchymal cells the following strategy was used: Epcam-/CD45-/CD31-<br>To isolate alveolar macrophages the following strategy was used: CD64+/CD45+ -> SiglecF+<br><br>Gating strategies used to isolate cells for scRNA-seq analyses are provided in the Supplementary Information |

☒ Tick this box to confirm that a figure exemplifying the gating strategy is provided in the Supplementary Information.

