## [Peer Review File · Nature]

Early fibrotic niches establish tumour-permissive microenvironments

Corresponding Author: Dr Joo-Hyeon Lee

Version 0:

Reviewer comments:

Referee #1

(Remarks to the Author)

In this work by Cardoso, Choi et al., authors investigate the sequential tissue remodeling of the lung during early oncogenesis, using a sophisticated KrasG12D mouse model that enables lineage tracing. Using multiple techniques, authors show that during pre-neoplastic stages, the lung mesenchymal undergoes remodeling, with a subset of alveolar fibroblasts – already present in homeostatic conditions – adopting a more fibrotic- and wound healing-like phenotype. Authors highlight differences in the immune-compartment, with a pre-tumoral and early-tumor niche characterized by presence of exhausted-like T cells and immunosuppressive granulocytes. The study further demonstrates a role for amphiregulin-EGFR axis in promoting the reprogramming of fibroblasts and alveolar macrophages, mediated by AT2 mutant pre-cancer cells.

This work focuses on an important aspect of tumor development, which is tumor initiation and the structural and cellular changes that tissues undergo during early oncogenesis. The study is linear and clear to follow, the methods are state-of-the-art and the Figures are elaborate, although discrepancies between the data and the interpretation of the data are not addressed enough in the manuscript. This work highlights important biological findings, such as the role of Areg-EGFR axis in establishing a pre-cancer environment that will facilitate tumor development. The first part of the work, however, is in parts redundant and lacks novelty. It is also not clear to what extent the events described are specific to Kras or more general to lung tumorigenesis. The gene signature of rewired fibroblasts during oncogenesis and the one of pericytes are often very similar, yet this point is not addressed in the text. Similarly, lung tissue staining often lacks a precise distinction between these cell subsets. Ex vivo and in vitro studies focused on macrophage rewiring could also be revised and enriched. Hence, this work could benefit from improvement.

Major comments:

1. Figure 1 and extended Fig 1 - Although the staining and the meta-analysis are convincing, they leave open discrepancies relating to the distinction between pericytes and fibroblasts that are not addressed in the text and that should be included.
 - Reprogrammed fibroblasts in Figure 1g and 1n are characterized by significant upregulation of Postn, Acta2 and Pdgfrb – among others. However, these markers are also appointed as pericytes markers as shown in Extended Data 1-d. This should be addressed and discussed.
 - Furthermore, the choice of protein markers to identify the reprogrammed fibroblasts in proximity of cancerous cells ex vivo (figure 1i and 1j) is only partly convincing, as either ACTA2+ PDGFRb+ or RUNX1+PDGFRb+ are used as markers to define these cells. Authors should use codex, multiplexed IF or a similar method to better distinguish reprogrammed fibroblasts from pericytes (e.g., as a triple positive RUNX1+ACTA2+PDGFRb+ cells). This will also allow the inclusion of additional cell-specific markers such as MCAM for pericytes and FAP and PDPN for fibroblasts.
2. Figure 3 – Continuing the previous point, when assessing the presence of fibroblast subsets and their spatial distribution in relation to AT2 oncogenic cells in figure 3a, 3b and 3d, similar comments made for figure 1i and 1j apply. The employment of multiplexed IF in this case could not only allow a better definition of RUNX1+PDGFRb+ cells as pathogenic fibroblasts and not pericytes, but could also allow to observe the spatial distribution of the reprogrammed fibroblasts in relation to macrophages, which is of great relevance, given recent evidence on the pro-tumoral effects of macrophage and CAFs interactions (PMID 35862581, PMID 30894509, PMID 39239526, PMID 37726308).
 - The authors should quantify their fluorescent staining panels. In particular, In Figure 3a and extended 3a they should

quantify the change in fibroblasts and not just macrophages.

- It is not clear why for this part of the research, only PDGFRb - and not RUNX1 - was used to identify reprogrammed fibroblasts, and this point should be addressed. What is the role of RUNX1 in these fibroblasts? It has been previously mentioned in the context of fibroblast reprogramming also in breast cancer, this study should be cited (PMID 36287637).
- In addition, although the in vitro-ex vivo assays shown in figure 3e-g are convincing, it would be interesting to see whether the loss of MHCII expression in AMs induced by Red2Kras-mesenchymal cells reflects a phenotypic rewiring (e.g., changes in CD204, 206, Arg expression) and/or a functional change in macrophages. For example, rewired macrophages could be co-cultured in presence of CD4 or CD8 T cells to assess T cell state and activation.

3. The authors describe 3 subsets of fibroblasts – Alveolar, Adventitial and Reprogrammed. They claim that Alveolar shift to reprogrammed, however the adventitial subset also changes and this change is not discussed. Moreover, Adventitial fibroblasts seem fewer in the kras than confeti in Fig 1d but increased in Fig 1E. Later on in Figure 5f the Adventitial change as well, and this is not discussed.

4. Figure 4 - The lineage-tracing experiments as well as the multi-culture/organoids studies offer mechanistic insights in a convincing manner. However, similarly to previous in vitro experiments, authors solely assess MHC-II expression on macrophages as a measure for their rewiring. As an overall reduction in macrophages is observed, it would be of great interest to assess other markers associated with a pro-tumorigenic / immune-suppressive phenotype in macrophages, as well as their functional activity in presence or absence of Gefitinib (for example in relation to their ability to alter T cell activation and/or migratory properties).

5. Figure 5 and extended data 6 - Although authors highlight a reduction of the Runx1 reprogrammed fibroblast population in AT2AregKO mice following transcriptomic analysis, the lung tissue staining was performed using only PDGFR-beta as a marker for this fibroblast subset. It is once again not clear why authors did not stain for RUNX1 as this is used as the main marker to define such population. This point should be addressed. It would be relevant to also assess MHC-II expression on macrophages in AT2AregKO mice compared to WT, to further strengthen the claims. And once again, staining should be quantified at least for some panels (Extended 6d-h).

6. Figure 6 and extended Figure 8 – The authors use patient samples with EGFR WT. Would patients with mutated EGFR signaling show a similar or different phenotype? Is the RAS signaling pathway specifically important also in patients or would other lung cancer organoids show a similar reprogramming?

Minor comment:

The authors should be consistent with markers in staining and should quantify at least some of the staining panels, particularly those in which staining is scarce even in the positively stained

(Remarks on code availability)

Referee #3

(Remarks to the Author)

In this manuscript the authors dissect the changes in the tumor microenvironment during early tumorigenesis with a special focus on fibroblasts and immune cells (alveolar macrophages mostly), as well as their interaction with mutant AT2 cells identifying Areg-EGFR axis as a key mechanism driving the remodeling of the tumor niche and contributing to tumor development. Combining several techniques including in vivo mouse models, scRNASeq analysis, immunofluorescence and organoid cultures, the authors explore the expansion and reprogramming of the cells surrounding Kras mutant AT2 cells, which is addressed in multiple ways using different models and strategies. Finally, the authors suggest that a similar remodeling of the tumor microenvironment occurs in human LUAD at some extent. If this is the case, findings derived from this work may contribute to the development of novel therapeutic approaches for early-stage lung cancer.

These novel and relevant findings are consistent with very recent papers published in the context of injury-repair including their own previous reports. In general, the conclusions are well supported by the data provided, but some concerns remain. The alterations in the tumor microenvironment are clear, but the sequential order of events need to be clarified in my view. Increasing the number of samples where indicated, and including additional time points may help to unravel the dynamics in the system.

Major comments

1. Based on previous publications, it is surprising that murine reprogrammed fibroblasts do not express Cthrc1, while in human is highly expressed in Fibroblasts 6 together with ACTA2, PDGFRB, etc. It would be interesting to stain for it in Red2Kras lungs together with Acta2, Pdgfrb and Runx1 to confirm whether reprogrammed fibroblasts really resemble fibrotic fibroblasts.

2. Adventitial fibroblasts increase 10% in Red2Kras lungs. Despite they do not give rise to reprogrammed fibroblasts according to the trajectory analysis, which is their contribution to tumor formation or to the remodeling of the environment? are they found in close proximity to RFP+ mutant cells, in the periphery or within the RFP+ tumors?

3. Since pericytes are defined by Pdgfrb and Postn expression, also expressed by reprogrammed fibroblasts, it would be necessary to stain for pericyte markers and discuss their difference to reprogrammed fibroblasts and their potential contribution in this context.

4. F4/80 is a general marker of macrophages including AMs but also interstitial macrophages (IMs). A combination with another marker specific of AMs should be shown in Fig. 2d to confirm the expansion of AMs in particular. Since different methods have been used to show macrophage expansion (flow cytometry and immunofluorescence), it needs to be clarified whether only AMs (CD64+ SiglecF+) expand or if it is an expansion of the entire macrophage population (F4/80+) and be consistent along the manuscript.
5. Adding the numbers of the expanded immune cells in Fig. 2f, j and k make 90-95% of all the CD45+ cells, is this consistent with the immune landscape observed in the lung tumor microenvironment? Other immune cell populations shown in Ext Data Fig 2d including B cells, CD4 and CD8+ T cells for instance, would be then reduced? This should be mentioned and discussed.
6. In their previous publication (Choi et al., 2020), the authors demonstrate the relevance of interstitial macrophages in the context of injury, producing IL1b that is essential for inducing AT2 cells to differentiate in order to accomplish regeneration. Here, reprogrammed AMs upregulate proinflammatory cytokines including Il1b and their location has changed. So, apart from expressing SiglecF, how different are they to IMs? Could they be acquiring an IM signature? And if so, could they support mutant cell states?
7. An image showing MHC-II co-stained with another AM marker should be shown. Is there a decrease in MHC-II+ AM numbers in addition to a reduction in the levels of MHC-II in AMs?
8. Along the manuscript, decrease in MHC-II levels is shown to demonstrate AM reprogramming. This could mean a reduction in their function not necessarily reprogramming. Thus, increase in proinflammatory cytokines or other changes are required to support that there is reprogramming in macrophages. In addition, in several experiments, the expansion in macrophages is reported, but their reprogramming is not shown (Fig 4h, i., Fig 5 i, j, Exp Data Fig 5 g,h, Exp Data Fig 6 f-h). The analysis must include both.
9. In the text (L183) is indicated that "Within 1 week post-induction, reprogrammed fibroblasts expressing Pdgfrb and Runx1 emerged in direct contact with nascent RFP+ tumours", then an image showing co-expression of Pdgfrb and Runx1 would be required in Figure 3a. In Extended Data Fig 3a, are Pdgfra+ Runx1+ cells also expressing Pdgfrb? How many of the Pdgfra+ fibroblasts around the tumor have been reprogrammed at 1w timepoint? Is there an increase comparable to that of the macrophages? Do both cell types follow a similar temporal expansion?
10. Figure 3b, 3c. Since this is very important for establishing the sequential order of events, the "n" needs to be increased to confirm an increase or not in macrophages in Red2Kras lungs at 1w after induction. Also, here it is analyzed only the expansion but not the reprogramming of macrophages that maybe occurs already 1w post-induction. Do they show MHC-II decrease and increase in proinflammatory cytokine expression?
11. In the co-culture experiment shown in Fig 3e, is there a change in the mesenchymal cells in terms of expansion, survival, or phenotypic change? What is the impact of Red2Kras vs WT AMs on WT mesenchymal cells?
12. In Fig. 3i, like in 3c, an increase in the "n" per group is needed to really find out which are the dynamics. Do Lcn2+ Pdgfra+ inflammatory cells express aSMA, Runx1 or Pdgfrb?
13. It seems that Areg-EGFR also regulate survival and/or proliferation of fibroblasts, not only reprogramming (Extended Data Fig 5b and Fig 4g, 4m). Please, provide this information.
14. It is not shown which cell types express EGFR and in which cell type is activated. Areg and Gefitinib treatments can affect AT2 cells and macrophages as well. Thus, it is necessary to clarify the effect of Areg on each cell type alone and how EGFR signaling inhibition affects them. Also, are there other cells in the lung expressing Areg apart from epithelial cells?
15. In Fig 4i. the analysis has to show the relative MFI for MHC-II of AMs from Confetti DMSO and Confetti Gefitinib together with the control and treated Red2kras AMs. It is assumed that AMs were isolated for this analysis since AT2 cells also express MHC-II but the detailed method of how this analysis has been performed is not clear. Please, explain further in the text or the figure legend.
16. In the experiment showed in Fig 4j, Areg is added together with Ereg, is Areg alone not enough to see an effect on macrophages? The effect of Areg in mesenchymal cells is very obvious (Extended Data Fig 5b). If reprogrammed fibroblasts drive AM changes, Areg alone should be enough. Please explain this point. For the experiments in Fig 4h and 4j, please provide images showing macrophage expansion and MHC-II expression.
17. "Notably, mutant organoids grown without fibroblasts exhibited no response to Gefitinib treatment". Which response is referring to? Do they grow? Do they generate DATP-like cells? In Fig 4m, which is the state of RFP+ cells? Are they AT2 cells expressing SPC? Or they advanced to another state, for instance cd177+ cells? Please, characterize them in more detail co-staining with other epithelial markers, and with Ki67
18. Similarly, a deeper characterization of RFP+ cells after Kras inhibition (MRTX1133, Fig 4o-q) and EGFR signaling inhibition (Gefitinib, Ext Data Fig 6) in vivo is required. Are AT2 and AT1 cell numbers recovered compared to a WT lung? What are RFP+ cells? Co-stain with different markers for the different epithelial states in these models and/or show the distribution of the clusters as in Figure 5n.

19. L296. "Moreover, Gefitinib treatment decreased the number of reprogrammed mutant cells compared to control (Extended Data Fig. 6i). These data demonstrate that EGFR activation is required for fibroblast reprogramming, which drives macrophage remodelling and supports tumour expansion in vivo". It is unknown whether the effect derived from Gefitinib treatment on mutant AT2 cells is direct or indirect due to lack of fibroblast reprogramming, and it is not demonstrated in this context that changes in macrophages are induced by fibroblast reprogramming. In vivo it is difficult to understand what happens first. Additional earlier timepoints during the course of the treatments or the genetic perturbations will help clarifying the changes in each cellular compartment contributing to dissect the mechanism.

20. In vivo administration of Gefitinib to Red2Kras since day 0, impedes the formation of DATP-like cells at day 4? or their production of Areg at day 7 when Pdgfrb+ Runx1+ fibroblast emerge? In vivo Gefitinib treatment results in a dramatic decrease in RFP+ cells (Ext Data Fig 6). However, in vitro experiments (Fig 4g, 4m) show that they stopped expressing DATP-like markers but the size of the organoid is similar to that of the control. Please, address these discrepancies. Would a longer treatment result in mutant cell death?

21. To demonstrate that reprogrammed fibroblasts support mutant AT2 cells, they should be directly targeted. Ablation of Pdgfrb+ fibroblast using a DTA system, or deletion of Egr in Pdgfrb+ fibroblast will help clarifying this point analyzing the effects on AM and on reprogrammed AT2. What happen to RFP+ DATP-like cells in this context? Is there tumor expansion? Do AMs expand or reprogram?

22. To better understand what happen to mutant AT2 cells when Areg is inhibited, a timecourse is required. This will help to understand the increase in RFP+ AT2+ cells at the expense of RFP+ DATP-like cells and AT1 cells, together with the changes in fibroblasts and macrophages. Is EGFR activated in alveolar fibroblasts adjacent to targeted AT2 cells? CD177+ cells do not change, what can this mean?

23. In human LUAD tissue (Fig 6), CTHRC1+, ACTA2+ fibroblasts are not found within the tumor but in the periphery. Do they express PDGFRB or Lcn2? Do they show EGFR activation? Do KRT8+ cells express DATP-like markers and produce AREG? How are AMs in this context?

24. Please, characterize RFP+ cells during the course of human organoid formation, confirm AREG-EGFR effects and incorporate AMs to the analysis.

Minor comments:

1. The gate in Fig. 2h needs to be the same for Confetti and for Red2Kras. Also, in 2i, flow panel for CD3 is not shown. Please, show all of them.

2. L215,216: "...suggests that fibrotic signals dominate over inflammatory cues during early tumour development, as recently described in regeneration". The authors recently demonstrated that Il1r1 is required for driving tumor formation via mutant AT2 cellular reprogramming (England et al, CSC 2025). Please, re-phrase or clarify what you mean exactly.

3. It is surprising to see that Pdgfra is still expressed in reprogrammed fibroblasts. Previously, it has been shown that Pdgfrb+ fibrotic fibroblasts stopped expressing Pdgfra 28 days after injury (Jones et al., Nature 2024). Could this be related to reversibility-persistence in the tissue of these fibroblasts? Just comment on these differences.

(Remarks on code availability)

The code is not available at the moment.

Referee #4

(Remarks to the Author)

In this study, the authors use mouse models and co-culture systems to investigate how the tumor and TME interact on a molecular and cellular level during tumor development to facilitate this process. The authors focus mostly on fibrotic niche formation and ligand/receptor interactions that may enable the TME to facilitate tumor development.

The models used in this study are very interesting and relevant. The overall area of study is important and timely. However, there are major limitations in terms of novelty and depth.

-Overall, the findings are somewhat underwhelming, as the involvement of the AREG-EGFR pathway in cancer/tumor microenvironment interactions is already well established.

-Despite the known implication of the AREG-EGFR pathway, the authors do not mention or investigate that macrophages can also produce AREG, which has been described previously. Furthermore, it remains unclear whether macrophages can also influence fibroblasts and induce the formation of the fibrotic fibroblast phenotype observed.

-The mechanisms by which reprogrammed fibroblasts enhance macrophage accumulation and tissue remodeling are not explored. The molecular pathways involved in this crosstalk should be addressed.

-It is also unclear whether tumor cells can directly influence macrophages, contributing to their recruitment, survival, or phenotypic polarization. This needs to be examined.

-The paper does not sufficiently explore how alterations in alveolar macrophages promote tumor development beyond a reduction in antigen presentation. An experiment involving co-culturing tumor cells with alveolar macrophages previously conditioned by fibroblasts would be informative.

-Regarding macrophage expansion, the study does not specify whether this results from the proliferation of resident alveolar macrophages, the recruitment and differentiation of monocytes, or both. Experimental clarification is needed.

-While the study highlights a pro-inflammatory environment, it does not show how this state benefits tumor progression, especially given that inflammation can have a dual or opposing effects on cancer development.

-How generalizable are the findings also remains to be demonstrated in this study.

(Remarks on code availability)

Version 1:

Reviewer comments:

Referee #1

(Remarks to the Author)

The authors have nicely addressed most of my comments, adding substantial mechanistic insights to their work.

Several remaining issues should be addressed before this manuscript is accepted for publication.

- In response to my comment on Figure 1, the authors added new staining and claim that "IF co-staining for $Pdgfr\alpha$ and $Pdgfr\beta$ in both Confetti and Red2Kras lungs showed reduced $Pdgfr\alpha$ expression in reprogrammed fibroblasts expressing $Pdgfr\beta$ located within RFP+ tumours (Revised Fig. 1i; reproduced in Rev1-Fig. 2b)." I still see a lot of PDGFRalpha staining in the representative image. Can the authors explain this? This should be quantified.

- The authors nicely show the spatiotemporal dynamics of reprogrammed fibrotic fibroblasts in new Figure 3. However there is no pValue in Reviewer Figure 3B – if these results are not statistically significant, the authors should clearly state it and not present this as a change.

- In many of the revised Figures (Reviewer Figure 3, 4, 5, 7) the authors show only 3 mice/Ns per group and do not mention in the Fig. legends whether these experiments have been repeated and how many times. Showing data from only 3 mice in only one experiment is not sufficient. This is also true for some of the new IF imaging experiments (for example Reviewer Fig 10) where the author show a representative image but do not specify how many mice or images were taken. This must be addressed.

- The authors should cite and discuss previous studies that have not been mentioned - PMID 34624218 and PMID: PMC9889778

Referee #3

(Remarks to the Author)

The revised manuscript incorporates substantial new analyses and experiments that fully address my previous concerns, clarifying the results and supporting the conclusions.

This study defines the temporal and spatial dynamics of fibroblasts and macrophages relative to mutant AT2 cells during early oncogenesis. The authors demonstrate that mutant AT2 cells initiate fibrotic reprogramming of adjacent alveolar fibroblasts through an Areg–EGFR signaling axis, a key mechanism driving tumour–niche co-evolution. These fibrotic fibroblasts subsequently remodel the immune compartment by driving alveolar macrophage (AM) expansion and phenotypic rewiring through the Tnc–TLR4 axis, establishing a permissive immunosuppressive niche that promotes immune-cell recruitment and the emergence of inflammatory fibroblasts, ultimately contributing to tumor growth. Importantly, disrupting these tumor-niche interactions halts mutant-cell reprogramming as well as stromal and immune remodelling. The development of inducible human LUAD organoids further supports that similar microenvironmental reprogramming occurs in human disease, highlighting a potentially targetable window for therapeutic intervention in early-stage lung cancer.

Overall, the findings are solid and rigorous, well supported by multiple mouse and human models, and provide compelling evidence for a spatiotemporal signalling axis coordinating early tumor-niche evolution.

Thus, I believe the manuscript is suitable for publication in Nature.

However, I would like to suggest some minor changes to further improve the readiness and comprehensiveness of the study:

1. It would help to indicate in Figure legend in Fig.2, Ext Data Fig. 2, 4, what the markers indicate, i.e: CD3, T cells. What is GL3? Or what is CD11c or CD11b labeling?
2. Extended Data Fig. 3 is not referenced in the manuscript
3. Pag. 321, Fig 4 I-o, the increase in AT2 cells is not significant so in the text, it can only be indicated a tendency.
4. A more clear description of how cell counts were conducted. It seems that at least 7 fields per mouse lung were counted, in n=3 in some cases. If so, nested t test should have been performed to keep into account the variability between the different fields in 1 mouse. Additionally, the statistics paragraph should include how group size (n) was calculated.

Referee #4

(Remarks to the Author)

The authors have adequately addressed my comments in the revised manuscript, which is improved.

Version 2:

Reviewer comments:

Referee #1

(Remarks to the Author)

The authors have addressed all my remaining concerns. Congratulations

Referees' comments:

Referee #1 (Remarks to the Author):

In this work by Cardoso, Choi et al., authors investigate the sequential tissue remodeling of the lung during early oncogenesis, using a sophisticated KrasG12D mouse model that enables lineage tracing. Using multiple techniques, authors show that during pre-neoplastic stages, the lung mesenchymal undergoes remodeling, with a subset of alveolar fibroblasts – already present in homeostatic conditions – adopting a more fibrotic- and wound healing-like phenotype. Authors highlight differences in the immune-compartment, with a pre-tumoral and early-tumor niche characterized by presence of exhausted-like T cells and immunosuppressive granulocytes. The study further demonstrates a role for amphiregulin-EGFR axis in promoting the reprogramming of fibroblasts and alveolar macrophages, mediated by AT2 mutant pre-cancer cells.

This work focuses on an important aspect of tumor development, which is tumor initiation and the structural and cellular changes that tissues undergo during early oncogenesis. The study is linear and clear to follow, the methods are state-of-the-art and the Figures are elaborate, although discrepancies between the data and the interpretation of the data are not addressed enough in the manuscript. This work highlights important biological findings, such as the role of Areg-EGFR axis in establishing a pre-cancer environment that will facilitate tumor development. The first part of the work, however, is in parts redundant and lacks novelty. It is also not clear to what extent the events described are specific to Kras or more general to lung tumorigenesis. The gene signature of rewired fibroblasts during oncogenesis and the one of pericytes are often very similar, yet this point is not addressed in the text. Similarly, lung tissue staining often lacks a precise distinction between these cell subsets. Ex vivo and in vitro studies focused on macrophage rewiring could also be revised and enriched. Hence, this work could benefit from improvement.

We thank the Reviewer for the thoughtful comments and the opportunity to clarify the novelty and significance of our fibroblast findings, which we have now further strengthened with additional molecular, spatial, mechanistic, and cross-model experiments. Although fibroblasts have been linked to tumorigenesis, our work advances this understanding by demonstrating that a fibrotic, pro-cancer fibroblast state emerges at the pre-neoplastic stage, is driven by Areg-EGFR signalling from nascent DATP-like mutant epithelial states, and plays a causal role in maintaining reprogrammed tumour cell states and early niche establishment for tumour development.

To incorporate the Reviewer's constructive feedback and reinforce our conclusions, we have now:

- Molecularly and spatially distinguished early fibrotic fibroblasts from pericytes, showing that they form discrete clusters and occupy distinct anatomical niches during early tumorigenesis.
- Expanded our analysis of fibroblast-immune interactions, adding *in vitro* and *in vivo* evidence that early fibrotic fibroblasts reprogramme macrophages via the Tnc-TLR4 axis, promoting immune recruitment.
- Demonstrated that Areg-dependent tumour and fibroblast reprogramming are present in an *EGFR*-driven model of lung tumorigenesis, supporting that early fibrotic niche formation is conserved beyond the Kras context.

Most importantly, we provide functional evidence that DATP-like mutant epithelial states act as early signalling hubs that initiate and coordinate tumour niche establishment. These states trigger fibroblast and immune remodelling that, in turn, reinforce epithelial transformation. Altogether, these revisions define a previously unrecognized temporal and mechanistic hierarchy in which nascent tumour states orchestrate multicellular niche reconstruction essential for tumour development.

Major comments:

1. Figure 1 and extended Fig 1 - Although the staining and the meta-analysis are convincing, they leave open discrepancies relating to the distinction between pericytes and fibroblasts that are not addressed in the text and that should be included.

- Reprogrammed fibroblasts in Figure 1g and 1n are characterized by significant upregulation of *Postn*, *Acta2* and *Pdgfrb* – among others. However, these markers are also appointed as pericytes markers as shown in Extended Data 1-d. This should be addressed and discussed.

- Furthermore, the choice of protein markers to identify the reprogrammed fibroblasts in proximity of cancerous cells *ex vivo* (figure 1i and 1j) is only partly convincing, as either ACTA2+ PDGFRb+ or RUNX1+PDGFRb+ are used as markers to define these cells. Authors should use codex, multiplexed IF or a similar method to better distinguish reprogrammed fibroblasts from pericytes (e.g., as a triple positive RUNX1+ACTA2+PDGFRb+ cells). This will also allow the inclusion of additional cell-specific markers such as MCAM for pericytes and FAP and PDPN for fibroblasts.

We agree with the Reviewer's point that some of the markers used to identify reprogrammed fibroblasts are also expressed by pericytes in the scRNA-seq data. However, our analysis reveals that reprogrammed fibroblasts form a transcriptionally and spatially distinct population. In the UMAP, reprogrammed fibroblasts and pericytes segregate into clearly separate clusters (**Revised Extended Fig. 1b**; reproduced in **Rev1-Fig. 1a**). Although both populations express *Pdgfrb*, reprogrammed fibroblasts emerging in *Red2Kras* lungs are uniquely defined by *Runx1*, *Runx2*, *Tnc* and *Fst*, which are not expressed by pericytes or other mesenchymal populations (**Revised Extended Fig. 1d**; reproduced in **Rev1-Fig. 1b**). To better delineate this population during early oncogenesis, we performed co-staining for *Runx1*, *Acta2* and *Pdgfrβ* in both *Confetti* and *Red2Kras* lungs, as suggested. No cells co-expressing all three markers were detected under homeostatic conditions, whereas *Runx1⁺Acta2⁺Pdgfrβ⁺* fibroblasts were readily observed adjacent to RFP⁺ mutant cells (**Revised Fig. 1j**; reproduced in **Rev1-Fig. 1c**). Furthermore, the revised manuscript now provides the mechanistic evidence that *Tnc⁺* fibrotic fibroblasts remodel macrophages through TLR4 signalling (**Revised Fig. 3i-l**; **Revised Ex Fig. 5**). Altogether, these results consistently support the emergence of a distinct, early reprogrammed fibroblast population during tumorigenesis. We have now included these data in our revised manuscript.

Reviewer1_Figure 1. Transcriptionally and spatially distinct fibrotic fibroblasts emerge during pre-cancer stage. **a**, UMAP plot showing all integrated scRNA-seq of mesenchymal cells obtained from *Confetti* and *Red2Kras* datasets, with the populations of “reprogrammed fibroblasts” and “pericytes” highlighted. **b**, Dotplot showing the expression of key marker genes distinguishing each mesenchymal cell cluster. Genes specific of the reprogrammed fibroblast population are highlighted in red box. Each dot is coloured based on the average expression and sized based on the percentage of cells expressing that gene. **c**, Representative confocal images showing reprogrammed fibroblasts and lineage-labelled *Kras*^{G12}-mutant AT2 cells in *Confetti* and *Red2Kras* lungs at 2 weeks post-induction. Runx1 (blue), Acta2 (yellow), Pdgfrβ (grey) and RFP (red, tumour). Orange arrows highlight reprogrammed fibroblasts. Scale bar, 50 μm. Images representative of *n* = 3 mice.

• Following up on this, the authors should add a marker in Fig 1i that decreases – of alveolar or adventitial fibroblasts.

ScRNA-seq analyses revealed a marked downregulation of alveolar fibroblast markers, including *Tcf21*, *Scube2*, *Npnt* and *Pdgfra*, in reprogrammed fibroblasts (**Revised Fig. 1g**; reproduced in **Rev1-Fig. 2a**). Consistently, IF co-staining for Pdgfra and Pdgfrβ in both *Confetti* and *Red2Kras* lungs showed reduced Pdgfra expression in reprogrammed fibroblasts expressing Pdgfrβ located within RFP⁺ tumours (**Revised Fig. 1i**; reproduced in **Rev1-Fig. 2b**). These new data have been incorporated into the revised manuscript.

Reviewer1_Figure 2. Transition of alveolar fibroblasts to reprogrammed fibroblasts. **a**, Dotplot showing the expression of key marker genes across fibroblast populations. Alveolar fibroblast markers are reduced in reprogrammed fibroblasts (highlighted by red box). Each dot is coloured by average expression and scaled by the percentage of cells expressing the gene. **b**, Representative confocal images of reprogrammed fibroblasts and lineage-labelled *Kras*^{G12D}-mutant AT2 cells in *Confetti* and *Red2Kras* lungs at 2 weeks post-induction. Pdgfr β (yellow), Pdgfra (grey) and RFP (red, tumours). Orange arrows indicate reprogrammed fibroblasts with decreased Pdgfra expression. Scale bar, 50 μ m. Images representative of $n = 3$ mice.

2. Figure 3 – Continuing the previous point, when assessing the presence of fibroblast subsets and their spatial distribution in relation to AT2 oncogenic cells in figure 3a, 3b and 3d, similar comments made for figure 1i and 1j apply. The employment of multiplexed IF in this case could not only allow a better definition of RUNX1+PDGFRb+ cells as pathogenic fibroblasts and not pericytes, but could also allow to observe the spatial distribution of the reprogrammed fibroblasts in relation to macrophages, which is of great relevance, given recent evidence on the pro-tumoral effects of macrophage and CAFs interactions (PMID 35862581, PMID 30894509, PMID 39239526, PMID 37726308).

- The authors should quantify their fluorescent staining panels. In particular, In Figure 3a and extended 3a they should quantify the change in fibroblasts and not just macrophages.
- It is not clear why for this part of the research, only PDGFRb - and not RUNX1 - was used to identify reprogrammed fibroblasts, and this point should be addressed. What is the role of RUNX1 in these fibroblasts? It has been previously mentioned in the context of fibroblast reprogramming also in breast cancer, this study should be cited (PMID 36287637).

We appreciate the Reviewer's comment, and the references provided. To better delineate the spatiotemporal dynamics of fibroblasts and immune cells contributing to tumour development, we traced RFP⁺ mutant cells and emerging tumour niches over time, from 1 to 8 weeks after oncogenic activation at clonal resolution. By 1 week post-induction, Pdgfr β ⁺Runx1⁺ fibroblasts emerged in direct contact with nascent RFP⁺ mutant cells, and by 2 weeks, nearly all expanding tumours were surrounded by fibrotic fibroblasts, which persisted throughout tumour growth (**Revised Fig. 3a, c**; reproduced in **Rev1-Fig. 3a, b**). These findings indicate that fibrotic reprogramming is initiated at the onset of tumorigenesis. In contrast, macrophage remodelling became prominent between 2 to 4 weeks, suggesting that fibroblast reprogramming precedes major macrophage expansion and phenotyping rewiring (**Revised Fig. 3b, d, e**). During this stage, we further demonstrated that Tnc⁺Pdgfr β ⁺ fibrotic fibroblasts reprogram macrophages through the Tnc-TLR4 axis (**Revised Fig. 3i-l**; **Revised Ex Fig. 5**). We then dissected the functional and spatial interactions between activated macrophages and inflammatory fibroblasts that dominate the tumour periphery at later stages (4 weeks onwards post-induction) (**Revised Fig. 3m, n**; **Revised Ex Fig. 6**). As suggested, the revised manuscript now incorporates a spatiotemporal reconstruction of these multicellular interactions at clonal resolution, along with comprehensive quantitative analyses.

Recent studies have identified both Runx1 and Runx2 as key regulators of fibrosis activation during lung fibrosis^{1,2}. Deletion of either factor impairs the transition of alveolar fibroblasts into a fibrotic state after injury, likely through modulation of Tgfβ1 signalling. These findings support a general role for Runx1/2 in driving the differentiation of alveolar fibroblasts into pathological states in response to profibrotic cues such as Tgfβ1 or Areg, depending on the context. Furthermore, as the Reviewer noted, Runx1 is also induced in fibroblasts transitioning into CAFs in breast cancer³, suggesting that similar transcriptional programs may govern fibroblast reprogramming across tissues. Thus, our data in the lung context may reflect a broader, conserved mechanism underlying stromal remodeling in diverse tumour types.

- In addition, although the *in vitro-ex vivo* assays shown in figure 3e-g are convincing, it would be interesting to see whether the loss of MHCII expression in AMs induced by *Red2Kras*-mesenchymal cells reflects a phenotypic rewiring (e.g., changes in CD204, 206, Arg expression) and/or a functional change in macrophages. For example, rewired macrophages could be co-cultured in presence of CD4 or CD8 T cells to assess T cell state and activation.

We thank the Reviewer for this important question, which was also raised by two other Reviewers. As suggested, we further analysed the phenotypic and functional features of reprogrammed macrophages emerging at the early stage of tumour development. Our scRNA-seq analysis revealed that AMs exhibited the most pronounced transcriptomic divergence in *Red2Kras* lungs, forming a discrete cluster distinct from homeostatic AMs and IM/monocyte-derived macrophages on the UMAP (**Revised Fig. 2a, b**; reproduced in **Rev1. Fig. 4a, b**). These reprogrammed AMs shared core transcriptomic features with resident alveolar macrophages based on canonical lineage markers, but were distinguished by elevated expression of specific genes such as *Msr1* and *Ch25h* (**Revised Fig. 2c**; reproduced in **Rev1. Fig. 4c**). Consistently, IF staining confirmed high *Msr1* (CD204) expression in the expanded AMs of *Red2Kras* lungs (**Revised Fig. 2j**; reproduced in **Rev1. Fig. 4e**). Importantly, co-culture of AMs with reprogrammed mesenchymal cells promoted AM proliferation and upregulated *Msr1* expression, reflecting fibroblast-mediated

phenotypic rewiring of AMs in *Red2Kras* lungs (**Revised Fig. 3j-l**; reproduced in **Rev1. Fig. 4f-i**). We identified Tenascin-C (Tnc), secreted by reprogrammed fibroblasts in *Red2Kras* lungs, as a key regulator of this process, acting through Toll-like receptor 4 (TLR4) signalling to drive AM expansion and phenotypic remodelling during early tumorigenesis (**Revised Fig. 3i-l**).

Reviewer1_Figure 4. Phenotypic rewiring of reprogrammed macrophages. **a, b**, UMAP visualization of macrophage subsets in *Confetti* and *Red2Kras* lungs. **c**, Dotplot showing expression of key markers across subsets. **d**, Heatmap showing chemokines upregulated in reprogrammed AMs relative to homeostatic AMs, indicating enhanced potential for recruiting other immune cells. **e**, Representative confocal images of reprogrammed AMs and lineage-labelled *Kras*^{G12D}-mutant AT2 cells in *Confetti* and *Red2Kras* lungs at 4 weeks following *Kras*^{G12D} induction. DAPI (blue), Msr1 (green), F4/80 (grey) and RFP (red, tumours). Images representative of $n = 3$ mice. **f**, Schematic illustrating co-cultures of wildtype AMs with mesenchymal cells isolated from *wildtype* (homeostatic) or *Red2Kras* (oncogenic) lungs. **g**, Representative confocal images of fibroblasts-AM co-cultures. DAPI (blue), Msr1 (yellow), F4/80 (green) and Ki67 (red). **h, i**, Percentage of Msr1⁺ (**h**) and Ki67⁺ (**i**) AMs in the co-cultures from (f, g). Data presented as mean \pm s.e.m.; Each dot represents one independent experiment. Homeostatic ($n = 3$) and oncogenic ($n = 4$). Statistical significance was determined using a two-tailed unpaired Student's t test. * $p < 0.05$; *** $p < 0.005$.

Beyond alterations in antigen presentation programs, reprogrammed AMs exhibited elevated expression of chemokines critical for immune cell recruitment. In particular, *Cxcl2*, a ligand for *Cxcr2*, was markedly upregulated (**Revised Fig. 2c, d**; reproduced in **Rev1. Fig. 4c, d**). Notably, consistent with previous studies⁴, we confirmed that *Cxcr2* is predominantly expressed on neutrophils, a key population in lung tumour development (**Revised Ex Fig. 4d**; reproduced in **Rev1. Fig. 5f**)⁵⁻⁸. To investigate the functional role of these macrophages during early tumour development, we specifically depleted alveolar macrophages via intratracheal administration of clodronate liposomes following oncogenic activation (**Revised Fig. 2k**; reproduced in **Rev1_Fig. 5a**). Ablation of macrophages significantly inhibited tumour growth (**Revised Fig. 2l, m**; reproduced in **Rev1_Fig. 5b-d**). Importantly, this was accompanied by a marked reduction in neutrophil expansion and infiltration – key contributors to lung tumour development – suggesting that macrophages promote tumour growth likely through *Cxcl2-Cxcr2*-dependent neutrophil recruitment (**Revised Fig. 2n, o**; **Revised Ex Fig. 4e**; reproduced in **Rev1_Fig. 5e,g**)^{4,6,8}. In addition, reprogrammed macrophages expressed elevated levels of *Cxcl16* (**Revised Fig. 2d**; reproduced in **Rev1_Fig. 4d**), whose receptor, *Cxcr6*, is enriched in $\gamma\delta$ -T cells, a population critical for lung tumour development (**Rev1_Fig. 5f**; **Revised Ex Fig. 4d**)⁷. Notably, $\gamma\delta$ -T recruitment was markedly impaired in the absence of macrophages (**Revised Fig. 2p, q**; **Revised Ex Fig. 4f**; reproduced in **Rev1_Fig. 5e,h**). These results demonstrate that AM reprogramming is essential for early tumour development by orchestrating the recruitment of neutrophils and $\gamma\delta$ T cells. These data have been incorporated into the revised manuscript.

Together, our findings demonstrate that macrophage reprogramming involves both phenotypic and functional changes, promoting the recruitment of neutrophils and $\gamma\delta$ T cells to establish a tumour-promoting oncogenesis.

Reviewer1_Figure 5. Macrophages are essential for early tumour development. **a**, Experimental scheme to deplete macrophages by clodronate liposome treatment via intratracheal root. **b**, Representative confocal images showing alveolar macrophage depletion after clodronate liposome administration. **c**, Flow cytometry analysis and quantification of alveolar macrophages following treatment. Each dot represents an independent mouse. Statistical significance was determined using a two-tailed unpaired Student's t test. $*p < 0.05$. **d**, H&E staining of lungs treated with control or clodronate liposome (left) and quantification of average tumour size (right). Each dot represents an independent tumour mass from 4 independent mice. Statistical significance was determined using a two-tailed unpaired Student's t test. $****p < 0.0001$. **e**, Representative confocal images showing reduced expansion of neutrophils and $\gamma\delta$ T cells. RFP (red, tumour), Ly6G (green, upper panel), GL3 (green, lower panel), and DAPI (blue), scale bar, 100 μm . **f**, Dotplot showing the expression of key chemokine receptors in neutrophils and $\gamma\delta$ T cells. **g**, Flow cytometry analysis (left) and quantification (right) of neutrophils. Each dot represents an independent mouse. **h**, Flow cytometry analysis (left) and quantification (right) of $\gamma\delta$ T cells. Each dot represents an independent mouse. Statistical significance was determined using a two-tailed unpaired Student's t test. $*p < 0.05$.

3. The authors describe 3 subsets of fibroblasts – Alveolar, Adventitial and Reprogrammed. They claim that Alveolar shift to reprogrammed, however the adventitial subset also changes and this change is not discussed. Moreover, Adventitial fibroblasts seem fewer in the *kras* than *confetti* in Fig 1d but increased in Fig 1E. Later on in Figure 5f the Adventitial change as well, and this is not discussed.

As the Reviewer noted, sub-clustering of the main fibroblast populations in our scRNA-seq dataset showed a relative increase in the proportion of adventitial fibroblasts in *Red2Kras* lungs compared with *Confetti* controls (43.9% versus 34%, respectively) (**Rev1-Fig. 6a**). However, re-clustering of adventitial fibroblasts revealed substantial overlap between *Red2Kras* and *Confetti* cells in UMAP space, with no cluster preferentially enriched in either dataset (**Rev1-Fig. 6b, c**). These data suggest that, although adventitial fibroblasts may be more abundant in *Red2Kras* lungs, their transcriptional identity remains largely unchanged relative to homeostasis.

To examine potential spatial changes during early oncogenesis, we stained for the adventitial fibroblast marker Pi16 in both *Confetti* and *Red2Kras* lungs. Consistent with previous reports, Pi16⁺ cells were localized around large arteries and airways within bronchovascular bundles in homeostasis, with no cells present in the distal alveoli (**Rev1-Fig. 6d**)⁹. A similar distribution was observed in *Red2Kras* lungs, with Pi16⁺ fibroblasts confined to the adventitial space (**Rev1-Fig. 6e**). Some RFP⁺ tumours arising near large airways and blood vessels were found in close proximity to Pi16⁺ fibroblasts (**Rev1-Fig. 6e**). To test whether spatial proximity to adventitial fibroblasts influences tumour development, we compared the cellular composition of RFP⁺ tumours in alveolar versus adventitial regions. Both regions contained Cd177⁺ and DATP-like mutant states, suggesting no clear association between adventitial fibroblasts and mutant cell reprogramming (**Rev1-Fig. 6f**).

Together, these results suggest that adventitial fibroblasts undergo no major transcriptional or spatial changes during early oncogenesis. Moreover, the variations in their representation in scRNA-seq datasets do not necessarily indicate significant expansion of adventitial fibroblasts *in vivo*.

Reviewer1_Figure 6. No noticeable phenotypic alterations in adventitial fibroblasts of *Red2Kras* lungs. **a**, Proportion of cells in each fibroblast cluster (from Revised Fig. 1e). Percentages smaller than 5% are omitted. **b**, UMAP plot of sub-clustered adventitial fibroblasts. **c**, UMAP plot showing the distribution of cells from *Confetti* and *Red2Kras* datasets. **d**, **e**, Representative confocal images showing the spatial location of PI16⁺ fibroblasts in adventitial and distal alveolar regions of *Confetti* (**d**) and *Red2Kras* (**e**) lungs. DAPI (blue), Pi16 (green) and RFP (red, tumour). Scale bar, 50 μ m. Images representative of $n = 3$ mice. **f**, Representative confocal images of RFP⁺ tumours emerging in alveolar or adventitial regions at 2 weeks post-induction (single tamoxifen dose, 0.1mg/gbw). DAPI (blue), Pi16 (green) and RFP (red, tumours). Scale bar, 50 μ m. Images representative of $n = 3$ mice.

4. Figure 4 - The lineage-tracing experiments as well as the multi-culture/organoids studies offer mechanistic insights in a convincing manner. However, similarly to previous in vitro experiments, authors solely assess MHC-II expression on macrophages as a measure for their rewiring. As an overall reduction in macrophages is observed, it would be of great interest to assess other

markers associated with a pro-tumorigenic / immune-suppressive phenotype in macrophages, as well as their functional activity in presence or absence of Gefitinib (for example in relation to their ability to alter T cell activation and/or migratory properties).

We thank the Reviewer for this valuable question. As detailed above, in addition to reduced MHC-II expression, we identified distinct features of reprogrammed AMs, notably the upregulation of Msr1 (**Revised Fig. 2; Rev1-Fig. 4**). Thus, we analyzed Msr1 expression together with MHC-II reduction as defining markers of AM reprogramming in our revised manuscript. To dissect the underlying mechanism, we established organoid co-cultures under three different conditions: (1) wildtype (WT) AM alone, (2) WT-AMs with RFP⁺ tumour cells, and (3) WT-AMs with RFP⁺ tumour cells and WT-mesenchymal cells (**Revised Ex Fig. 10a, b; reproduced in Rev1-Fig. 7a, b**). Importantly, Msr1 expression was selectively induced in AMs co-cultured with both mesenchymal and tumour cells, but not with tumour cell alone, and this induction was abolished by Gefitinib treatment (**Revised Ex Fig. 10c, d; reproduced in Rev1-Fig. 7c-e**). These data indicate that EGFR-dependent mutant signals mediated through mesenchymal cells are required to drive AM reprogramming. Consistent with this, stimulation of WT-AMs co-cultured with WT-mesenchyme and WT-AT2 cells using Areg or Areg/Ereg altered the expression of *Arg1*, *Ym-1* and *Tnfa* in AMs, confirming their phenotypic rewiring (**Revised Ex Fig. 10k, n; reproduced in Rev1-Fig. 7f**).

Collectively, these results demonstrate that fibrotic fibroblasts, induced by tumour cell-derived Areg/Ereg, drive macrophage phenotypic reprogramming. Significantly, as described above, we further identified Tnc, secreted by these fibrotic fibroblasts, as a key regulator of this process that acts through TLR4 signalling to promote AM expansion and phenotypic remodelling during early tumorigenesis (**Revised Fig. 3j-l**).

Reviewer1_Figure 7. Reprogramming of Msr1⁺ alveolar macrophages by AREG-EGFR signalling axis. **a**, Experiment scheme of organoid co-culture. **b**, Representative flow cytometry plots showing the populations of AMs (CD45⁺EpCAM⁻), tumour cells (CD45⁻EpCAM⁺), and mesenchymal cells (CD45⁻EpCAM⁻) within organoid co-cultures in **(a)**. Numbers adjacent to gates indicate the percentage of each population. **c**, Flow cytometry analysis of Msr1 expression in AMs (left) and quantification of MFI across three different co-culture conditions (right). Each dot represents an independent experiment. Statistical significance was determined using a two-tailed unpaired Student's t test. *** $p < 0.001$. **d**, Experiment scheme of organoid co-cultures treated with Gefitinib. **e**, Flow cytometry analysis of Msr1 expression in AMs (left) and quantification of MFI (right). Each dot represents an independent experiment. Statistical significance was determined using a two-tailed unpaired Student's t test. ** $p < 0.01$. **f**, qPCR analysis of *Arg1*, *Ym-1*, and *Tnfa* expression in AMs co-cultured with WT-mesenchymal and WT-AT2 cells treated with PBS, Areg alone, or Areg/Ereg. Each dot represents an independent experiment. Data are mean \pm s.e.m.; two-tailed unpaired Student's t test. * $p < 0.5$, ** $p < 0.01$, *** $p < 0.001$.

5. Figure 5 and extended data 6 - Although authors highlight a reduction of the Runx1 reprogrammed fibroblast population in AT2AregKO mice following transcriptomic analysis, the lung tissue staining was performed using only PDGFR-beta as a marker for this fibroblast subset. It is once again not clear why authors did not stain for RUNX1 as this is used as the main marker to define such population. This point should be addressed. It would be relevant to also assess MHC-II expression on macrophages in AT2AregKO mice compared to WT, to further strengthen the claims. And once again, staining should be quantified at least for some panels (Extended 6d-h).

We thank the Reviewer for this important question. As suggested, we performed additional co-staining for Runx1 and Pdgfr β in both *Areg-flox* and *Areg-flox/flox* lungs. Consistent with our transcriptomics data, Runx1⁺Pdgfr β ⁺ fibroblasts surrounding RFP⁺ tumours were markedly reduced upon *Areg* deletion in mutant AT2 cells (**Rev1-Fig. 8a**). We also confirmed a decrease in Tnc⁺Pdgfr β ⁺ fibrotic fibroblasts in *Areg-flox/flox* lungs. (**Revised Fig. 5m**; reproduced in **Rev1-Fig. 8f**). To further assess macrophage phenotypes in this context, we analysed MHC-II expression in AMs from *Areg-flox* versus *Areg-flox/flox* lungs. AMs from *Areg-flox* lungs showed features of reprogrammed AMs, including reduced MHC-II-associated gene expression and increased phenotypic rewiring genes such as *Msr1* (**Revised Ex Fig. 15e**; reproduced in **Rev1-Fig. 8b**). In contrast, AMs from *Areg-flox/flox* lungs largely reverted to a homeostatic state, restoring MHC-II expression and reducing *Msr1* expression. IF analysis further supported these findings, revealing a substantial decrease in Msr1⁺ macrophages in *Areg-flox/flox* lungs compared with controls (**Revised Fig. 5n-p**; reproduced in **Rev1-Fig. 8c-e**).

Together, our results demonstrate that DATP-like states arising from nascent mutant AT2 cells serve as a central signalling hub to initiate tumour niche reconstruction. Deletion of *Areg* in mutant AT2 cells prevents both fibroblast and AM reprogramming, thereby blocking early niche remodelling essential for tumour development. These data, together with quantitative analyses, have been included in our revised manuscript.

Reviewer1_Figure 8. Deletion of *Areg* in AT2 cells prevents niche remodelling during early tumorigenesis. **a**, Representative confocal images of reprogrammed fibroblasts in *Areg*-*flox*/⁺ and *Areg*-*flox*/*flox* lungs at 2 weeks post-induction. DAPI (blue), Pdgrfβ (grey), Runx1 (green) and RFP (red, tumours). Scale bar, 50 μm. **b**, DotPlot showing the expression of key reprogramming markers in AMs from *Areg*-*flox*/⁺ and *Areg*-*flox*/*flox* lungs. Each dot is coloured based on the average expression and sized based on the percentage of cells expressing that gene **c**, Representative confocal images of reprogrammed AMs in *Areg*-*flox*/⁺ and *Areg*-*flox*/*flox* lungs at 2 weeks post-induction. DAPI (blue), Msr1 (green), F4/80 (grey) and RFP (red, tumours). Scale bar, 50 μm. **d**, **e**, Quantification of AM expansion (**d**) and reprogramming (**e**) in *Areg*-*flox*/⁺ and *Areg*-*flox*/*flox* lungs. Data presented as mean ± s.e.m., each dot represents an individual mouse. *Areg*-*flox*/⁺ (*n* = 5) and *Areg*-*flox*/*flox* (*n* = 6). Statistical significance was determined using a two-tailed Mann Whitney test. ***p* < 0.01 (**d**) and a two-tailed unpaired Student's t test. ***p* < 0.01 (**e**). **f**, Representative confocal images of Tnc⁺Pdgrfβ⁺ reprogrammed fibroblasts in *Areg*-*flox*/⁺ and *Areg*-*flox*/*flox* lungs at 2 weeks post-induction. DAPI (blue), Pdgrfβ (grey), Tnc (green) and RFP (red, tumours). Scale bar, 50 μm.

6. Figure 6 and extended Figure 8 – The authors use patient samples with EGFR WT. Would patients with mutated EGFR signaling show a similar or different phenotype? Is the RAS signaling pathway specifically important also in patients or would other lung cancer organoids show a similar reprogramming?

We thank the Reviewer for this valuable comment. To assess whether *EGFR*-mutant tumours engage similar epithelial-stromal reprogramming mechanisms, we re-analysed publicly available single-cell transcriptomics datasets from patients with *EGFR*-mutant adenocarcinoma (LUAD) and matched normal lung tissues (4 paired samples; **Rev1-Fig. 9a**)¹⁰. Sub-clustering of EpCAM⁺ epithelial cells identified an *AREG*^{high} population specifically enriched in *EGFR*-mutant LUAD, relative to AT2 cells from normal lungs (**Rev1-Fig. 9b-e**). Notably, this *AREG*^{high} population

exhibited reduced expression of AT2 markers, such as *ETV5* and *LAMP3*, and increased expression of DATP-associated markers, such as *SOX9* and *ITGA2*, suggesting the emergence of a distinct, reprogrammed epithelial state in *EGFR*-mutant LUAD (**Rev1-Fig. 9f**). Furthermore, re-clustering of *COL1A1*⁺ mesenchymal cells revealed a fibrotic mesenchymal population characterised by *CTHRC1*, *RUNX1* and *RUNX2* expression, which was selectively enriched in *EGFR*-mutant LUAD specimens (**Rev1-Fig. 9b, c, g-i**). Importantly, both *AREG*-expressing tumour cells and these fibrotic mesenchymal states were present in stage-I *EGFR*-mutant LUAD, suggesting that fibrotic niche formation is initiated early and may similarly be driven by *EGFR*-mutant cells via the *AREG* – *EGFR* signalling axis (**Rev1-Fig. 9j, k**).

Reviewer1_Figure 9. Fibrotic niche signatures co-emerge with AREG-expressing tumour cells in early EGFR-mutant lung adenocarcinoma (LUAD). **a**, Schematic of the workflow for analysing scRNA-seq dataset of early-stage (I-III) human LUAD samples. Patients with confirmed *EGFR* mutations were selected, comprising 4 LUAD and 4 matched distant background lung samples. **b**, UMAP plot showing all integrated scRNA-seq from the *EGFR*-mutant LUAD dataset. **c**, Dotplot showing the expression of key marker genes distinguishing epithelial (*EpCAM*) and mesenchymal (*COL1A1*) cell clusters. **d**, UMAP plot of epithelial cell sub-clustering from (**b**), revealing 7 distinct populations, colour-coded by cellular states. **e**, UMAP plot showing epithelial cell distribution, colour-coded by dataset origin (*EGFR*-mutant LUAD vs normal). **f**, Heatmap showing marker gene expression in (**d**). **g**, UMAP plot of mesenchymal cell sub-clustering from (**b**), revealing 7 distinct populations, colour-coded by cellular states. **h**, UMAP plot showing mesenchymal cell distribution, colour-coded by dataset origin (*EGFR*-mutant LUAD vs normal). **i**, UMAP feature plots showing the expression of key marker genes for reprogrammed fibroblasts across mesenchymal clusters in (**g**). **j, k**, UMAP plots showing the distribution of epithelial cells (**j**) and mesenchymal cells (**k**), colour-coded by LUAD stage of dataset origin.

To further validate whether *EGFR* mutant cells can directly remodel their microenvironment, we developed a 3D inducible *EGFR*-mutant LUAD model by introducing EGFP reporter system to express *EGFR*^{L858R} specifically in primary mouse AT2 cells (**Revised Ex Fig. 18a**; reproduced in **Rev1-Fig. 10a**). Lineage-labelled AT2 cells were isolated, cultured as organoids, transduced with lentivirus at day 14 to induce *EGFR*^{L858R} expression alongside GFP, and subsequently orthotopically engrafted into NSG mouse lungs (**Revised Ex Fig. 18a, b**; reproduced in **Rev1-Fig. 10a, b**). After three weeks, *tdTom*⁺*EGFP*⁺ mutant cells formed expanding lesions enriched for *Krt8* and *Areg* expression, hallmark features of DATP-like states also observed in *Kras*^{G12D}-driven models (**Revised Ex Fig. 18c, d**; reproduced in **Rev1-Fig. 10c, d**). Notably, engraftment of *EGFR*^{L858R}-mutant cells induced *Pdgfrβ*⁺ fibrotic fibroblasts and macrophage expansion in adjacent niches (**Revised Ex Fig. 18e**; reproduced in **Rev1-Fig. 10c, d**).

Reviewer1_Figure 10. Establishment of pre-neoplastic niches driven by EGFR-mutant cells. **a.** Experiment scheme of the 3D inducible LUAD model established by introducing *EGFR*^{L858R} mutation into primary AT2 cells, followed by orthotopic engraftment into NSG mouse lungs. **b.** Flow cytometry analysis showing AT2 cells transduced with *EGFR*^{L858R} mutation. **c.** Representative confocal images showing that engrafted *tdTom*⁺*EGFP*⁺ mutant cells reprogram into Krt8⁺ cell states. **d, e.** Representative confocal images showing enriched Areg expression in Krt8⁺ mutant cell states (**d**) and the associated remodelling of fibrotic fibroblasts and macrophages in tumour regions (**e**). Tomato (red, tumours), Areg (green, d), Pdgfrβ (green, d), Krt8 (white, c, d), CD64 (white, e), and DAPI (blue), scale bar, 100 μm.

Together, these findings demonstrate that oncogenic *EGFR* mutation reprograms AT2 cells into Areg⁺ states that actively remodel surrounding fibroblasts and macrophages, establishing a fibrotic, pro-tumorigenic niche. This identifies an epithelial-stromal reprogramming axis as a conserved early mechanism of niche formation across LUAD oncogenic drivers, extending beyond *KRAS* mutations. Supporting the broader significance of this pathway, a recent preprint (accompanying manuscript by Skrupskelyte *et al.*)¹¹ shows that survival of carcinogen-induced oesophageal squamous tumours similarly relies on EGF-mediated crosstalk between Sox9⁺ nascent tumour cells and fibrotic fibroblasts, illustrating that analogous epithelial-fibroblast circuits shape tumour initiation in diverse tissues. Together, these observations underscore the generalisability of our findings and highlight a shared EGFR-dependent pathway orchestrating early tumour development across cancer types.

Minor comment:

The authors should be consistent with markers in staining and should quantify at least some of the staining panels, particularly those in which staining is scarce even in the positively stained

We thank the Reviewer for this valuable feedback. We have added the corresponding staining data and quantitative analyses, which are now included in the revised manuscript.

Referee #3 (Remarks to the Author):

In this manuscript the authors dissect the changes in the tumor microenvironment during early tumorigenesis with a special focus on fibroblasts and immune cells (alveolar macrophages mostly), as well as their interaction with mutant AT2 cells identifying Areg-EGFR axis as a key mechanism driving the remodeling of the tumor niche and contributing to tumor development. Combining several techniques including *in vivo* mouse models, scRNASeq analysis, immunofluorescence and organoid cultures, the authors explore the expansion and reprogramming of the cells surrounding Kras mutant AT2 cells, which is addressed in multiple ways using different models and strategies. Finally, the authors suggest that a similar remodeling of the tumor microenvironment occurs in human LUAD at some extent. If this is the case, findings derived from this work may contribute to the development of novel therapeutic approaches for early-stage lung cancer.

These novel and relevant findings are consistent with very recent papers published in the context of injury-repair including their own previous reports. In general, the conclusions are well supported by the data provided, but some concerns remain. The alterations in the tumor microenvironment are clear, but the sequential order of events need to be clarified in my view. Increasing the number of samples where indicated, and including additional time points may help to unravel the dynamics in the system.

We thank the Reviewer for insightful and constructive feedback. As suggested, we have expanded our analyses by increasing the number of samples and performing a more detailed examination of spatiotemporal dynamics underlying initial oncogenic niche remodelling. In addition, we have incorporated new, extensive *in vitro* and *in vivo* experiments investigating the interactions between mutant cells and their surrounding niche components, providing deeper mechanistic insight into how the pre-neoplastic microenvironment is established.

Major comments

1. Based on previous publications, it is surprising that murine reprogrammed fibroblasts do not express *Cthrc1*, while in human is highly expressed in Fibroblasts 6 together with ACTA2, PDGFRB, etc. It would be interesting to stain for it in Red2Kras lungs together with Acta2, Pdgfrb and Runx1 to confirm whether reprogrammed fibroblasts really resemble fibrotic fibroblasts.

We appreciate the Reviewer's comment. To address this point, we sub-clustered the population of reprogrammed fibroblasts from our *Confetti* vs *Red2Kras* scRNA-seq dataset and identified a small subset of fibrotic cells expressing *Cthrc1* within this dataset (**Rev3-Fig. 1a**). To validate this finding, we also performed co-staining for *Cthrc1* and *Pdgfrβ* in both *Confetti* and *Red2Kras* lungs. Consistent with the transcriptomic data, *Cthrc1* was detected in a subset of *Pdgfrβ*⁺ reprogrammed fibroblasts within RFP⁺ tumours (**Rev3-Fig. 1b**). Together, these data support that, similar to lung injury, a population of fibrotic *Cthrc1*⁺ cells arises during early oncogenesis.

Reviewer3_Figure 1. Emergence of fibrotic fibroblasts expressing Cthrc1 in Red2Kras lungs. a, Feature plots showing the expression of *Cthrc1* and other fibrotic marker genes in reprogrammed fibroblasts. **b,** Representative confocal images of reprogrammed fibroblasts and lineage-labelled *Kras*^{G12D}-mutant AT2 cells in *Confetti* and *Red2Kras* lungs at 2 weeks post-induction. DAPI (blue), Pdgfr β (green), Cthrc1 (grey) and RFP (red, tumours). Red arrows indicate Cthrc1⁺ reprogrammed fibroblasts. Scale bar, 50 μ m. Images representative of $n = 2$ mice.

2. Adventitial fibroblasts increase 10% in Red2Kras lungs. Despite they do not give rise to reprogrammed fibroblasts according to the trajectory analysis, which is their contribution to tumor formation or to the remodeling of the environment? are they found in close proximity to RFP+ mutant cells, in the periphery or within the RFP+ tumors?

We thank the Reviewer for raising this point, which was also noted by the Reviewer 1.

As detailed above, sub-clustering of the main fibroblast populations in our scRNA-seq dataset showed a relative increase in the proportion of adventitial fibroblasts in *Red2Kras* lungs compared with *Confetti* controls (43.9% versus 34%, respectively) (**Rev3-Fig. 2a**). However, re-clustering of adventitial fibroblasts revealed substantial overlap between *Red2Kras* and *Confetti* cells in UMAP space, with no cluster preferentially enriched in either dataset (**Rev3-Fig. 2b, c**). These data suggest that, although adventitial fibroblasts may be more abundant in *Red2Kras* lungs, their transcriptional identity remains largely unchanged relative to homeostasis.

To examine potential spatial changes during early oncogenesis, we stained for the adventitial fibroblast marker Pi16 in both *Confetti* and *Red2Kras* lungs. Consistent with previous reports, Pi16⁺ cells were localized around large arteries and airways within bronchovascular bundles in homeostasis, with no cells present in the distal alveoli (**Rev3-Fig. 2d**)⁹. A similar distribution was observed in *Red2Kras* lungs, with Pi16⁺ fibroblasts confined to the adventitial space (**Rev3-Fig. 2e**). Some RFP⁺ tumors arising near large airways and blood vessels were found in close proximity to Pi16⁺ fibroblasts (**Rev3-Fig. 2e**). To test whether spatial proximity to adventitial fibroblasts influences tumour development, we compared the cellular composition of RFP⁺ tumours in alveolar versus adventitial regions. Both regions contained Cd177⁺ and DATP-like

mutant states, indicating no clear association between adventitial fibroblasts and mutant cell reprogramming (**Rev3-Fig. 2f**).

Together, these results suggest that adventitial fibroblasts undergo no major transcriptional or spatial changes during early oncogenesis. Moreover, the variations in their representation in scRNA-seq datasets do not necessarily indicate significant expansion of adventitial fibroblasts *in vivo*.

Reviewer3_Figure 2. No noticeable phenotypic alterations in adventitial fibroblasts of *Red2Kras* lungs. **a**, Proportion of cells in each fibroblast cluster (from Revised Fig. 1e). Percentages smaller than 5% are omitted. **b**, UMAP plot of sub-clustered adventitial fibroblasts. **c**, UMAP plot showing the distribution of cells from *Confetti* and *Red2Kras* datasets. **d**, **e**, Representative confocal images showing the spatial location of PI16⁺ fibroblasts in adventitial and distal alveolar regions of *Confetti* (**d**) and *Red2Kras* (**e**) lungs. DAPI (blue), Pi16 (green) and RFP (red, tumours). Scale bar, 50 μ m. Images representative of $n = 3$ mice. **f**, Representative confocal images of RFP⁺ tumours emerging in alveolar or adventitial regions at 2 weeks post-induction (single tamoxifen dose, 0.1mg/gbw). DAPI (blue), Pi16 (green) and RFP (red, tumours). Scale bar, 50 μ m. Images representative of $n = 3$ mice.

3. Since pericytes are defined by *Pdgfrb* and *Postn* expression, also expressed by reprogrammed fibroblasts, it would be necessary to stain for pericyte markers and discuss their difference to reprogrammed fibroblasts and their potential contribution in this context.

We thank the Reviewer for raising this point, which was also noted by the Reviewer 1. As detailed above, although some of the markers used to identify reprogrammed fibroblasts are also expressed by pericytes in the scRNA-seq data, our analysis reveals that reprogrammed fibroblasts show a distinct transcriptional signature compared to pericytes, evident by the clear separation of their respective clusters in the UMAP (**Revised Ex Fig. 1c**; reproduced in **Rev3-Fig. 3a**). Although both populations express *Pdgfrβ*, reprogrammed fibroblasts emerging in *Red2Kras* lungs are additionally defined by a unique set of genes, including *Runx1*, *Runx2*, *Tnc* and *Fst*, that are not expressed by pericytes or other mesenchymal populations (**Revised Ex Fig. 1d**; reproduced in **Rev3-Fig. 3b**). To better delineate this population during early oncogenesis, we performed co-staining for *Runx1*, *Acta2* and *Pdgfrβ* in both *Confetti* and *Red2Kras* lung samples, as suggested. No mesenchymal cells expressing all three markers were detected under homeostatic conditions, whereas *Runx1*⁺*Acta2*⁺*Pdgfrβ*⁺ fibroblasts were clearly observed in close proximity to RFP⁺ mutant cells (**Revised Fig. 1j**; reproduced in **Rev3-Fig. 3c**). Altogether, these results consistently support the emergence of a distinct populations of reprogrammed fibroblasts during early tumorigenesis. We have now included these data in our revised manuscript.

Reviewer3_Figure 3. Transcriptionally and spatially distinct fibrotic fibroblasts emerge during the pre-cancer stage. **a**, UMAP plot showing all integrated scRNA-seq of mesenchymal cells obtained from *Confetti* and *Red2Kras* datasets, with the populations of “reprogrammed fibroblasts” and “pericytes” highlighted. **b**, Dotplot showing the expression of key marker genes distinguishing each mesenchymal cell cluster. Genes specific of the reprogrammed fibroblast population are highlighted in red box. Each dot is coloured based on the average expression and sized based on the percentage of cells expressing that gene. **c**, Representative confocal images showing reprogrammed fibroblasts and lineage-labelled *Kras*^{G12}-mutant AT2 cells in *Confetti* and *Red2Kras* lungs at 2 weeks post-induction. Runx1 (blue), Acta2 (yellow), Pdgfrβ (grey) and RFP (red, tumour). Orange arrows highlight reprogrammed fibroblasts. Scale bar, 50 μm. Images representative of *n* = 3 mice.

4. F4/80 is a general marker of macrophages including AMs but also interstitial macrophages (IMs). A combination with another marker specific of AMs should be shown in Fig. 2d to confirm the expansion of AMs in particular. Since different methods have been used to show macrophage expansion (flow cytometry and immunofluorescence), it needs to be clarified whether only AMs (CD64⁺ SiglecF⁺) expand or if it is an expansion of the entire macrophage population (F4/80⁺) and be consistent along the manuscript.

In response to the Reviewer's comment, we further characterised the macrophage populations expanded in *Red2Kras* lungs during early tumorigenesis. IF analysis revealed that F4/80⁺ macrophages co-expressed PDL-1 and CD11c – markers of alveolar, but not interstitial or monocyte-derived macrophages – were substantially expanded in the inter-tumour regions (Revised Fig. 2h, i; Revised Ex Fig. 2e; reproduced in Rev3_Fig. 4a, b).

Reviewer3_Figure 4. Expansion of reprogrammed alveolar macrophages in the inter-tumour region during early tumour development. a, b. Representative confocal images showing alveolar macrophage markers. DAPI (blue), F4/80 (green), RFP (red, tumour), PDL-1 (white) (a). DAPI (blue), F4/80 (green), RFP (red, tumour), CD11C (white) (b). Scale bar, 100 μ m.

To quantitatively analyse alveolar and interstitial macrophages, we performed flow cytometry in *Confetti* and *Red2Kras* lungs (Revised Fig. 2e, f; reproduced in Rev3_Fig. 5). While CD64⁺ macrophages were significantly increased in *Red2Kras* lungs, the proportions of SiglecF⁺ and SiglecF⁻ subsets within the CD64⁺ population remained comparable between *Confetti* and *Red2Kras* lungs. These data indicate that both alveolar and interstitial macrophages expand, with alveolar macrophages representing the predominant population increasing specifically within the inter-tumour regions during early tumour onset. These findings have been incorporated into the revised manuscript.

Reviewer3_Figure 5. Flow cytometric analysis showing alveolar and interstitial macrophage populations in *Confetti* and *Red2Kras* lungs (left). Percentage of alveolar and interstitial macrophages in *Confetti* (n=3) and *Red2Kras* (n=3) lungs (right). Each dot represents an individual mouse. Statistical significance was assessed using a two-tailed unpaired Student's t test. *p<0.05, **p<0.01.

5. Adding the numbers of the expanded immune cells in Fig.2f, j and k make 90-95% of all the CD45+ cells, is this consistent with the immune landscape observed in the lung tumor microenvironment? Other immune cell populations shown in Ext Data Fig 2d including B cells, CD4 and CD8+ T cells for instance, would be then reduced? This should be mentioned and discussed.

We thank the Reviewer for this important question. In response, we have thoroughly re-analysed the changes in immune cell populations during early tumour development. scRNA-seq analysis revealed a significant increase in subsets of T cells, including regulatory T cells and CD153⁺PD-1⁺CD4⁺ T cells, which was corroborated by IF staining and flow cytometry (**Revised Ex Fig. 2c, f, g**; reproduced in **Rev3_Fig. 6a-c**). Although the abundance and spatial distribution of CD8⁺ T cells and B220⁺ B cells were not significantly altered, CD8⁺ T cells upregulated exhaustion markers, such as *Cd160*, *Btla*, and *Havcr2*, and exhibited a metabolic shift from oxidative phosphorylation toward glycolysis, features characteristic of T-cell exhaustion, contributing to an immunosuppressive niche that favours tumour progression (**Revised Ex Fig. 2h-j**; reproduced in **Rev3_Fig. 6d, e**). Furthermore, $\gamma\delta$ T cells and neutrophils were also increased in *Red2Kras* lungs (**Revised Ex Fig. 2k-p**; reproduced in **Rev3_Fig. 6a, f, g**). Notably, spatial mapping revealed distinct immune-tumour interaction patterns: while macrophages predominantly accumulated in inter-tumour regions, neutrophils and $\gamma\delta$ T cells were confined to intra-tumoral areas, suggesting compartmentalised immune contributions during tumour formation (**Revised Ex Fig. 2q, r**; reproduced in **Rev3_Fig. 6f, g**). These data have been incorporated into our revised manuscript and discussed accordingly.

Reviewer3_Figure 6. Dynamic immune cell landscape during early tumour development. a, Frequency of immune cell subsets in *Confetti* and *Red2Kras* lungs. **b,** Representative confocal images showing expansion of regulatory T cells. RFP (red, tumours), CD4 (green), FOXP3 (white), and DAPI (blue), scale bar, 100 μ m. **c,** Flow cytometry of CD153⁺PD-1⁺CD4⁺ T cells in *Confetti* versus *Red2Kras* lungs. **d,** Representative confocal images showing B cells and CD8⁺ T cells with no significant changes. RFP (red, tumours), B220 (green, upper panel), CD8 (green, lower panel), CD3 (white, lower panel), and DAPI (blue), scale bar, 100 μ m. **e,** UMAP of CD8⁺ T cells (left) and heatmap of exhaustion markers and metabolic signatures (right). **f,** Representative confocal images showing lineage-labelled *Kras*^{G12D}-mutant AT2 cells and $\gamma\delta$ T cells (left), and quantification of the frequency (right). **g,** Representative confocal images showing lineage-labelled *Kras*^{G12D}-mutant AT2 cells and neutrophils (left), and quantification of the frequency (right). Scale bar, 100 μ m. Each dot represents an individual mouse. Statistical significance was assessed using a two-tailed unpaired Student's t test. * $p < 0.05$, **** $p < 0.0001$.

6. In their previous publication (Choi et al., 2020), the authors demonstrate the relevance of interstitial macrophages in the context of injury, producing IL1b that is essential for inducing AT2 cells to differentiate in order to accomplish regeneration. Here, reprogrammed AMs upregulate proinflammatory cytokines including Il1b and their location has changed. So, apart from expressing SiglecF, how different are they to IMs? Could they be acquiring an IM signature? And if so, could they support mutant cell states?

We thank the Reviewer for this important question. In response, we comprehensively analysed the phenotypic identity, origin, and function of reprogrammed AMs emerging during early tumour development.

scRNA-seq analysis revealed that AMs exhibited the most pronounced transcriptomic divergence in *Red2Kras* lungs, forming a discrete cluster distinct from homeostatic AMs and IM/monocyte-derived macrophages on the UMAP (**Revised Fig. 2a, b**; reproduced in **Rev3. Fig. 7a, b**).

Although these reprogrammed AMs retained core transcriptomic features with resident AMs based on canonical lineage markers, such as *SiglecF* and *MertK*, they were distinguished by elevated expression of genes such as *Msr1* and *Ch25h* (**Revised Fig. 2c**; reproduced in **Rev3. Fig. 7c**). They also adopted a mixed inflammatory profile, with upregulation of both pro-inflammatory mediators (e.g. *IL-1a*, *IL-1b*) and anti-inflammatory genes (e.g. *Mrc1*, *Chil3*, *Arg1*) relative to homeostatic AMs (**Revised Fig. 2d**; reproduced in **Rev3. Fig. 7d**). In parallel with alterations in antigen presentation programmes, they upregulated chemokines, including *Cxcl2* and *Cxcl16*, key drivers of neutrophil and $\gamma\delta$ -T cell recruitment (**Revised Fig. 2d**; reproduced in **Rev3_Fig. 7d**).

Reviewer3_Figure 7. Transcriptional signatures of reprogrammed macrophages. a, b, UMAP visualization of macrophage subsets in *Confetti* and *Red2Kras* lungs. **c**, Dotplot showing expression of key macrophage markers across subsets. **d**, Heatmap showing chemokines upregulated in reprogrammed macrophages relative to homeostatic alveolar macrophages, indicating enhanced potential for recruiting other immune cells.

To determine whether these reprogrammed AMs originate from monocytes, we orthotopically engrafted RFP⁺ mutant organoids derived from *Red2Kras* lungs into the lungs of *CCR2-Cre^{ERT2}; Rosa-ZsGreen* mice, allowing lineage tracing of monocyte-derived cells (**Revised Ex Fig. 3a**; reproduced in **Rev3_Fig. 8a**). In these engrafted tumours, most macrophages surrounding RFP⁺ tumour cells lacked *ZsGreen* labelling, indicating that reprogrammed AMs arise primarily from resident alveolar macrophages rather than recruited monocytes (**Revised Ex Fig. 3b**; reproduced in **Rev3_Fig. 8b**). Interestingly, reprogrammed AMs were enriched in inter-tumour regions, whereas lineage-labelled monocyte-derived macrophages – although a relatively small population – were found both within and around tumours, suggesting distinct spatial and functional roles at this stage.

Reviewer3_Figure 8. Reprogrammed macrophages expanding in the inter-tumour region originate primarily from resident alveolar macrophages during early tumour development. a, Experiment scheme of orthotopic engraftment of tumour organoids into the lungs of *CCR2-Cre^{ERT2};ZsGreen* mouse. **b,** Representative confocal images showing macrophages expanded around tumours. RFP (red, tumours), ZsGreen (lineage-traced monocytes, green), CD64 (white), and DAPI (blue), scale bar, 100µm. Insets (bottom) are enlargements of the dashed boxes (top). Scale bars,

To investigate the functional role of these reprogrammed AMs during early tumorigenesis, we selectively depleted alveolar macrophages by intratracheal administration of clodronate liposomes following oncogenic activation (**Revised Fig. 2k; Revised Ex Fig. 4a-c**; reproduced in **Rev3_Fig. 9a-c**). Ablation of macrophages significantly impaired tumour growth with reduced mutant cell reprogramming (**Revised Fig. 2l, m; Rev3_Fig. 9d, e**). Importantly, this was accompanied by a marked reduction in neutrophil and $\gamma\delta$ -T cell infiltration, key contributors to lung tumour development (**Revised Fig. 2n-q; Revised Ex Fig. 4e, f**; reproduced in **Rev3_Fig. 9f, h, i**)^{4,7} Given the selective expression of *Cxcr2* and *Cxcr6* in neutrophils and $\gamma\delta$ -T cells, respectively, these results suggest that reprogrammed AMs promote early tumour development likely through *Cxcl2*-*Cxcr2*-mediated neutrophil recruitment and *Cxcl16*-*Cxcr6*-mediated $\gamma\delta$ -T cell recruitment (**Revised Ex Fig. 4d**; reproduced in **Rev3_Fig. 9g**).

Reviewer3_Figure 9. Macrophages are essential for early tumour development. **a**, Experimental scheme to deplete macrophages by clodronate liposome treatment via intratracheal root. **b**, Flow cytometry analysis and quantification of AMs following treatment. Each dot represents an independent experiment. **c**, Representative confocal images showing AM depletion after clodronate liposome administration. **d**, H&E staining of lungs treated with control or clodronate liposome (left) and quantification of average tumour size (right). Each dot represents an independent tumour mass from 4 independent mice. **e**, Representative confocal images showing reduced reprogrammed tumour cells upon macrophage depletion. RFP (red, tumours), SPC (white, top panel), SOX9 (green, middle panel), CD177 (white, bottom panel), and DAPI (blue), scale bar, 100 μ m. **f**, Representative confocal images showing reduced expansion of neutrophils and $\gamma\delta$ -T cells. RFP (red, tumours), Ly6G (green, upper panel), GL3 (green, lower panel), and DAPI (blue), scale bar, 100 μ m. **g**, Dotplot showing the expression of key chemokine receptors in neutrophils and $\gamma\delta$ -T cells. **h**, Flow cytometry analysis (left) and quantification (right) of neutrophils. Each dot represents an independent experiment. **i**, Flow cytometry analysis (left) and quantification (right) of $\gamma\delta$ -T cells. Each dot represents an independent experiment.

Finally, lineage tracing of RFP⁺ mutant clones together with their surrounding niche cells over time revealed that the emergence of inflammatory fibroblasts within inter-tumour regions coincides temporally with AM expansion and reprogramming (**Revised Fig. 3m, n; Revised Ex Fig. 6a-c**). *In vitro*, stimulation with IL-1 β or exposure to *Red2Kras* AMs induced *Lcn2* expression in Pdgfa⁺ fibroblasts, suggesting that reprogrammed AMs may contribute to the remodeling of inflammatory fibroblast states (**Revised Ex Fig. 6e, f**).

Altogether, our results show that reprogrammed AMs acquire phenotypes distinct from IMs – they retain AM identity, cluster separately from IMs, and display a transcriptional programme not observed in IMs. Thus, they do not adopt an IM-like signature. Rather than directly reprogramming tumour cells (as IM-derived IL-1 β directly reprogrammes AT2 cells during regeneration, Choi et al. 2020), reprogrammed AMs facilitate tumour development primarily through niche remodelling, including recruiting pro-tumorigenic immune cells and inducing inflammatory fibroblasts within inter-tumour regions. Although they express increased IL-1 β relative to homeostatic AMs, IM/monocyte-derived macrophages and neutrophils express far higher IL-1 β and reside predominantly within tumour cores (**Revised Ex Fig. 2n; reproduced in Rev3-Fig. 10**). This spatial and transcriptional pattern suggests that neutrophil-derived IL-1 β is likely the dominant cue sustaining reprogrammed tumour cells, consistent with our recent finding that *Il1r1*-deficient *Kras*^{G12D}-mutant AT2 cells exhibit restricted tumour reprogramming¹².

Collectively, these findings delineate a coordinated functional compartmentalisation of the niche cell population. Reprogrammed AMs remodel the surrounding niche to enable tumour expansion, whereas fibrotic fibroblasts and IL-1 β -producing neutrophils reinforce tumour cell reprogramming within the tumour core.

7. An image showing MHC-II co-stained with another AM marker should be shown. Is there a decrease in MHC-II+ AM numbers in addition to a reduction in the levels of MHC-II in AMs?

Reviewer3_Figure 10. Higher expression of IL-1 β in neutrophils and monocyte-derived macrophages. Dotplot showing expression of key macrophage markers across subsets

As suggested, in the revised manuscript, we further characterised AM alterations during early tumour development. Consistent with flow cytometry analysis, IF staining for MHC-II and CD11c revealed reduced MHC-II expression in CD11c⁺ AMs in *Red2Kras* lungs compared to *Confetti* lungs (**Revised Fig. 2g, i**; reproduced in **Rev3_Fig. 11**). Notably, while the total number of AMs were increased, a substantial proportion of these cells exhibited features of reprogrammed AMs marked by *Msr1* expression, accompanied by reduced MHC-II levels (**Revised Fig. 2e-j**). These findings suggest that alveolar macrophages expand but undergo reprogramming that diminishes their antigen-presenting capacity.

8. Along the manuscript, decrease in MHC-II levels is shown to demonstrate AM reprogramming. This could mean a reduction in their function not necessarily reprogramming. Thus, increase in proinflammatory cytokines or other changes are required to support that there is reprogramming in macrophages. In addition, in several experiments, the expansion in macrophages is reported, but their reprogramming is not shown (Fig 4h, i., Fig 5 i, j, Exp Data Fig 5 g,h, Exp Data Fig 6 f-h). The analysis must include both.

We thank the Reviewer for this insightful comment. In addition to the reduction of MHC-II expression, we further characterised the phenotypic and functional changes of alveolar macrophages during early tumour development. As shown in **Rev3_Fig. 7**, reprogrammed macrophages displayed elevated expression of *Msr1* (*Cd204*), a marker associated with the anti-inflammatory characteristics of M2-like and immunoregulatory macrophages, identifying *Msr1* as a reliable indicator of macrophage reprogramming¹³. We have now included analyses of *Msr1* expression alongside MHC-II downregulation in our revised manuscript (**Revised Fig. 2c, 2j, 3b, 3k, 4j, 4f**). Furthermore, as described above, functional studies demonstrated that these macrophages play an essential role in establishing pre-neoplastic niches by recruiting immune populations such as neutrophils and $\gamma\delta$ -T cells, which are critical for tumour initiation. Together these results substantiate that the observed alterations reflect functional reprogramming rather than a mere reduction in macrophage activity. We have incorporated these results in our revised manuscript.

9. In the text (L183) is indicated that “Within 1 week post-induction, reprogrammed fibroblasts expressing *Pdgfrb* and *Runx1* emerged in direct contact with nascent RFP+ tumours”, then an image showing co-expression of *Pdgfrb* and *Runx1* would be required in Figure 3a. In Extended Data Fig 3a, are *Pdgfra*⁺ *Runx1*⁺ cells also expressing *Pdgfrb*? How many of the *Pdgfra*⁺

fibroblasts around the tumor have been reprogrammed at 1w timepoint? Is there an increase comparable to that of the macrophages? Do both cell types follow a similar temporal expansion?

We thank the Reviewer for the insightful feedback, which was also raised by Reviewer 1. To better delineate the spatiotemporal dynamics of fibroblasts and immune cells contributing to tumour development, we traced RFP⁺ mutant cells and emerging tumour niches over time, from 1 to 8 weeks after oncogenic activation at clonal resolution. To examine the onset of fibroblast reprogramming, we performed co-staining for Pdgfr β and Runx1 and quantified the proportion of RFP⁺ tumours containing fibrotic fibroblasts, as suggested. By 1 week post-induction, Pdgfr β ⁺Runx1⁺ fibroblasts emerged in direct contact with nascent RFP⁺ mutant cells, and by 2 weeks, nearly all expanding mutant tumours were surrounded by fibrotic fibroblasts, which persisted throughout tumour growth (**Revised Fig. 3a, c**; reproduced in **Rev3-Fig. 12a, c**). Consistently, co-staining for Pdgfr α and Pdgfr β showed reduced Pdgfr α expression in reprogrammed fibroblasts expressing Pdgfr β located within RFP⁺ tumours, indicative of a transition toward a fibrotic phenotype (**Revised Fig. 1i**). In contrast, macrophage remodelling became prominent between 2 to 4 weeks, suggesting that fibroblast reprogramming precedes major macrophage expansion and phenotyping rewiring (**Revised Fig. 3b, d, e**; reproduced in **Rev3-Fig. 12b, d, e**). These findings collectively support that fibrotic reprogramming is initiated at the onset of tumorigenesis and represents a key event in establishing the pre-neoplastic niche.

Reviewer3_Figure 12. Spatiotemporal dynamics of reprogrammed fibrotic fibroblasts and macrophages. **a**, Representative confocal images of reprogrammed fibrotic fibroblasts and lineage-labelled AT2 cells in *Confetti* and *Red2kras* lungs. Mice received a single tamoxifen dose (0.1mg/gbw). Pdgfr β (green), Runx1 (grey) and RFP (red, tumour). Yellow arrows indicate reprogrammed fibroblasts. **b**, Representative confocal images showing macrophages and lineage-labelled AT2 cells in *Confetti* and *Red2kras* lungs across the indicated time points. DAPI (blue), Msr1 (green), F4/80 (grey) and RFP (red, tumours). Scale bar, 50 μ m. Images representative of $n = 3$ mice per time point. **c**, Percentage of RFP $^+$ tumours containing fibrotic fibroblasts. Data are mean \pm s.e.m.; Each dot represents one mouse. *Red2kras* 1w ($n = 3$), 2w ($n = 3$), 4w ($n = 3$) and 8w ($n = 3$). **d, e**, Quantification of F4/80 $^+$ macrophages within RFP $^-$ stromal regions (**d**) and Msr1 $^+$ macrophages (**e**) from (**b**). Data represent mean \pm s.e.m.; each dot corresponds to an individual mouse. *Confetti* ($n = 3$), *Red2kras* 1w ($n = 6$), 2w ($n = 4$), 4w ($n = 4$) and 8w ($n = 3$). Statistical significance was determined by two-tailed unpaired Student's t test. * $P < 0.05$; ** $P < 0.01$; *** $P < 0.005$.

10. Figure 3b, 3c. Since this is very important for establishing the sequential order of events, the “n” needs to be increased to confirm an increase or not in macrophages in *Red2Kras* lungs at 1w after induction. Also, here it is analyzed only the expansion but not the reprogramming of macrophages that maybe occurs already 1w post-induction. Do they show MHC-II decrease and increase in proinflammatory cytokine expression?

We are grateful for this important and constructive feedback. As addressed above, we increased the number of biological replicates and performed a more detailed quantitative analyses to refine the temporal sequence of AM expansion and reprogramming following oncogenic activation. These updated analyses show that AM reprogramming, marked by the emergence of Msr1 $^+$ AMs, is first detected at 2 weeks post-induction, whereas a significant AM expansion becomes evident at 4 weeks (**Revised Fig. 3b, d, e**; reproduced in **Rev3-Fig. 12b, d, e**). This indicates that phenotypic reprogramming precedes major macrophage expansion during early tumour initiation. Furthermore, reprogrammed AMs also exhibited reduced MHC-II expression together with upregulation of chemokines involved in immune cell recruitment (**Revised Fig. 2a-j**). These findings support the notion that transcriptional and functional reprogramming of AMs initiates prior to their numerical expansion and aligns with the early establishment of a tumour-promoting niche.

11. In the co-culture experiment shown in Fig 3e, is there a change in the mesenchymal cells in terms of expansion, survival, or phenotypic change? What is the impact of *Red2Kras* vs WT AMs on WT mesenchymal cells?

In our original manuscript, we identified Lcn2 $^+$ Pdgfra $^+$ inflammatory fibroblasts emerging primarily at 4 weeks post-induction, coincident with expansion of reprogrammed AMs. To clarify their temporal and spatial relationship, we performed co-staining for Pdgfra, Lcn2 and F4/80 in *Red2Kras* lungs from 1 to 8 weeks after *Kras* activation at clonal resolution. As observed previously, Lcn2 $^+$ inflammatory fibroblasts appeared at the periphery of RFP $^+$ tumours at 4 and 8 weeks and were consistently positioned adjacent to clusters of expanded reprogrammed AMs, supporting potential crosstalk between two populations (**Revised Fig. 3m, n**; reproduced in **Rev3-Fig. 13a, b**). Given that *Red2Kras* AMs upregulate inflammatory cytokines, including *IL-1b*, in scRNA-seq data, we investigated whether IL-1 β or AM-derived signals could modulate fibroblast

states. Treatment of wildtype mesenchymal cells with IL-1 β for 48 hours induced Lcn2 expression in Pdgfra⁺ fibroblasts (**Revised Ex Fig. 6e, f**; reproduced in **Rev3-Fig. 13c, d**). Consistently, co-culture of wildtype mesenchymal cells with *Red2Kras* AMs also upregulated Lcn2, indicating that reprogrammed macrophages can directly activate inflammatory fibroblast states in the inter-tumour regions. Importantly, these Lcn2⁺Pdgfra⁺ fibroblasts did not express fibrotic markers such as Tnc and were rarely detected within tumour cores (**Revised Ex Fig 6d**). This supports a spatially compartmentalised organisation of fibroblast populations: fibrotic fibroblasts induced by tumour-derived Areg predominate within tumour centres, whereas inflammatory fibroblasts induced by AM-secreted IL-1 β localise to inter-tumour areas. This mirrors prior observations that fibrotic cues override inflammatory programmes in fibroblasts during injury repair, suggesting that these regulatory mechanisms are conserved in tumour-associated fibroblasts¹⁴.

Together, these analyses demonstrate that *Red2Kras* AMs directly drive inflammatory fibroblast activation, highlighting their specific and spatially restricted role in shaping the inter-tumour mesenchymal niche.

Reviewer3_Figure 13. Inflammatory fibroblasts driven by reprogrammed macrophages. a, Representative confocal images showing macrophages, inflammatory fibroblasts and lineage-labelled AT2 cells in *Red2Kras* lungs at 8 weeks after oncogenic activation. Lcn2 (yellow), Pdgfra (grey), F4/80 (green) and RFP (red, tumours). Scale bar, 50 μ m. **b**, Quantification of Lcn2⁺Pdgfra⁺ fibroblasts at both tumour periphery and inside. Data represent mean \pm s.e.m.; each dot corresponds to an individual mouse. *Confetti* ($n = 3$), *Red2Kras* 1w ($n = 3$), 2w ($n = 3$), 4w ($n = 3$) and 8w ($n = 3$). Statistical significance was determined by two-tailed unpaired Student's t test. * $P < 0.05$; ** $P < 0.01$; **c**, Schematic illustrating WT fibroblasts treated with IL-1 β or co-cultured with *Red2Kras* alveolar macrophages. **d**, Representative confocal images of fibroblasts treated with IL-1 β or co-cultured with *Red2Kras* AMs. Lcn2 (red), Pdgfra (green), F4.80 (grey) and DAPI (blue).

12. In Fig. 3i, like in 3c, an increase in the “n” per group is needed to really find out which are the dynamics. Do Lcn2+ Pdgfra+ inflammatory cells express aSMA, Runx1 or Pdgfrb?

We thank the Reviewer for this comment. As addressed above, we increased the number of biological replicates and performed a more detailed quantitative analyses to refine the temporal dynamics of inflammatory fibroblast emergence and their distinction from fibrotic fibroblasts following oncogenic activation (**Rev3-Fig. 12 and 13**). To more precisely define the inflammatory fibroblast population, we re-analysed our mesenchymal scRNA-seq dataset from *Red2Kras* and *Confetti* lungs. Re-clustering of reprogrammed fibroblasts revealed a distinct subset expressing inflammatory markers *Lcn2* and *Saa3* with minimal expression of fibrotic markers *Cthrc1*, *Pdgfrb*, *Acta2* and *Runx2* (**Rev3-Fig. 14a**). Consistently, IF co-staining for Pdgfra, Lcn2 and Tnc confirmed that Pdgfra⁺Lcn2⁺ fibroblasts at the tumour periphery lack Tnc expression (**Revised Ex Fig. 6d**; reproduced in **Rev3-Fig. 14b**). Together, these analyses strengthen our conclusion that inflammatory fibroblasts are transcriptionally and functionally distinct from fibrotic fibroblasts, supporting their separate roles within the evolving tumour microenvironment.

Reviewer3_Figure 14. Inflammatory fibroblasts represent a distinct population from fibrotic fibroblasts. **a**, FeaturePlot showing inflammatory (*Lcn2*, *Saa3*) and fibrotic marker (*Cthrc1*, *Pdgfrb*, *Acta2*, *Runx2*) expression in reprogrammed fibroblasts from the *Red2Kras* scRNA-seq dataset. Inflammatory fibroblasts, highlighted in red. **b**, Representative confocal images for inflammatory fibroblasts and lineage-labelled *Kras*^{G12D}-mutant AT2 cells in *Red2Kras* lungs at 4 weeks following oncogenic activation. Pdgfra (blue), Lcn2 (yellow), Tnc (grey) and RFP (red, tumour). Images representative of *n* =2. Orange arrows highlight inflammatory fibroblasts lacking Tnc expression.

13. It seems that Areg-EGFR also regulate survival and/or proliferation of fibroblasts, not only reprogramming (Extended Data Fig 5b and Fig 4g, 4m). Please, provide this information.

To address whether Areg-EGFR signalling regulates not only fibroblast reprogramming but also their proliferation or survival, we performed additional analyses. Co-staining for Runx1, Pdgfr β and Ki67 in *Red2Kras* lungs at 4 weeks post *Kras*^{G12D} induction revealed proliferative Pdgfr β ⁺Runx1⁺ fibrotic fibroblasts in close contact to RFP⁺ tumours (**Rev3-Fig. 15a**). To investigate whether tumour-derived Areg directly promotes fibroblast proliferation, we treated isolated WT mesenchymal cells with Areg *in vitro*. Areg-mediated EGFR activation enhanced both fibrotic reprogramming and cell proliferation in these fibroblasts (**Rev3-Fig. 15b**). Altogether, these findings indicate that Areg-EGFR signalling drives both the proliferative expansion and fibrotic reprogramming of lung fibroblasts.

Reviewer3_Figure 15. Areg drives fibroblast proliferation and fibrotic reprogramming. **a**, Representative confocal images showing proliferative, reprogrammed fibroblasts in *Red2Kras* lungs at 4 weeks post induction with one dose of 0.1mg/gbw. Ki67 (grey), Pdgfr β (cyan), Runx1 (green) and RFP (red, tumours). Scale bar, 25 μ m. Images representative of $n = 3$. **b**, Representative confocal images of WT-mesenchymal cells treated with Areg or PBS. Pdgfr β (red), Ki67 (green), DAPI (blue). Scale bar, 25 μ m. Images representative of $n = 3$.

14. It is not shown which cell types express EGFR and in which cell type is activated. Areg and Gefitinib treatments can affect AT2 cells and macrophages as well. Thus, it is necessary to clarify the effect of Areg on each cell type alone and how EGFR signaling inhibition affects them. Also, are there other cells in the lung expressing Areg apart from epithelial cells?

Following the Reviewer's comment, to identify *EGFR*-expressing populations, we re-analysed our immune and mesenchymal scRNA-seq datasets together with our previously published epithelial dataset¹². *EGFR* expression was detected in the subsets of fibroblasts, epithelial cells, and monocyte-derived macrophages (**Revised Ex Fig. 8c**; reproduced in **Rev3-Fig. 16a**). To directly assess their responsiveness to Areg, we treated wildtype AMs and RFP⁺ mutant cells with Areg *in vitro*, as we previously did for fibroblasts (**Revised Ex Fig. 8d, j**; reproduced in **Rev3-Fig. 16b, f**). Areg treatment had no effect on AM expansion or reprogramming, and did not alter proliferation or cell-state transitions of RFP⁺ tumour organoids (**Revised Ex Fig. 8b-h**; reproduced in **Rev3-Fig. 16b-h**). Consistently, Gefitinib treatment of RFP⁺ organoids also produced no detectable effects (**Revised Ex Fig. 11f-h**). These data indicate that fibroblasts are the primary Areg-responsive population.

Reviewer3_Figure 16. Impact of Areg-EGFR signalling on macrophage and tumour cell behaviours. **a**, Violin plot showing *EGFR* expression across epithelial cells, fibroblasts, and macrophages. **b**, Experimental scheme to investigate the effect of Areg on AMs. **c**, Flow cytometry analysis of AM expansion after Areg or PBS treatment. **d**, **e**, Flow cytometry showing Msr1 (**d**) and MHC-II (**e**) expression on AMs cultured with Areg (left) and quantification of Msr1 or MHC-II mean fluorescence intensity (MFI). Each dot represents individual experiment. *ns*, not significant. **f**, Experimental scheme to investigate the effect of Areg on RFP⁺ tumour cells isolated from *Red2Kras* lungs. **g**, Bright-field images of RFP organoids treated with PBS or Areg. **h**, Representative IF images of RFP⁺ tumour organoids in (**f**, **g**). RFP (red, tumours), Krt8 (white), pro-SPC (green), and DAPI (blue), scale bar, 100 μ m.

Next, we examined the cellular source of Areg. scRNA-seq analysis revealed detectable *Areg* expression in regulatory T cells (Treg) and subsets of macrophages (**Revised Ex Fig. 7f**; reproduced in **Rev3-Fig. 17a**). However, DATP-like reprogrammed cells expressed the highest expression levels. IF analysis confirmed strong Areg enrichment in DATP-like cells compared to Foxp3⁺ Tregs and F4/80⁺ macrophages (**Revised Ex Fig. 7g, h**; **Rev3-Fig. 17b, c**). Importantly, deletion of *Areg* specifically in RFP⁺ mutant cells significantly restrained tumour development, highlighting that tumour-derived Areg is the key functional source.

These data clarify which cell types express EGFR, which respond to Areg-EGFR-signalling, and which populations produce Areg, and the corresponding data have been included in the revised manuscript.

Reviewer3_Figure 17. DATP-like reprogrammed cells are the major source of Areg in the tumour microenvironment. **a**, Violin plot showing *Areg* expression across epithelial and immune cell populations. **b**, Representative IF images showing *Areg* expression in tumour cells and macrophages. RFP (red, tumours), F4/80 (white), *Areg* (green), and DAPI (blue), scale bar, 100 μ m. **c**, Representative IF images showing *Areg* expression in tumours and regulatory T cells. RFP (red, tumours), Foxp3 (white), *Areg* (green), and DAPI (blue), scale bar, 100 μ m.

15. In Fig 4i. the analysis has to show the relative MFI for MHC-II of AMs from Confetti DMSO and Confetti Gefitinib together with the control and treated Red2kras AMs. It is assumed that AMs were isolated for this analysis since AT2 cells also express MHC-II but the detailed method of how this analysis has been performed is not clear. Please, explain further in the text or the figure legend.

We apologize for this confusion. We have now clarified the experimental design and updated the figure legend accordingly.

For MHC-II analyses, AMs were isolated as CD64⁺SiglecF⁺ population gated on EpCAM⁻CD45⁺, while mesenchymal cells were sorted as EpCAM⁻CD31⁻CD45⁻ from wildtype lungs. These cells were then co-cultured with either WT *TdTomato*⁺ AT2 cells from *Sftpc-Cre^{ERT2}; TdTomato* lungs or *Red2Kras* RFP⁺ mutant AT2 cells from *Sftpc-Cre^{ERT2}; Red2Kras* lungs following tamoxifen treatment (**Revised Fig. 4f**; reproduced in **Rev3-Fig. 18a**). Detailed methods for these experiments have been included in the revised manuscript.

As suggested, we analysed relative MHC-II MFI in AMs from WT DMSO, WT Gefitinib, *Red2Kras* DMSO, and *Red2Kras* Gefitinib co-culture conditions. AMs co-cultured with *Red2Kras* AT2 cells and WT mesenchymal cells exhibited a pronounced reduction in MHC-II expression, which was restored by Gefitinib treatment (**Revised Fig. 4g, h**; reproduced in **Rev3-Fig. 18b, c**). Consistently, AM expansion and reprogramming, as indicated by *Msr1* upregulation, were also significantly reduced by Gefitinib (**Revised Fig. 4i, j**; reproduced in **Rev3-Fig. 18d-f**). In contrast, Gefitinib had no effect on AMs co-cultured with WT AT2 and WT mesenchymal cells.

Reviewer3_Figure 18. Remodelling of alveolar macrophages via EGFR signalling. **a**, Schematic illustrating 3D organoid co-cultures of wild-type alveolar macrophages and mesenchymal cells with lineage-labelled cells isolated from *Sftpc-Cre^{ERT2}*; *TdTomato* or *Red2Kras* lungs. **b**, Flow cytometry analysis showing MHCII expression in alveolar macrophages from each condition in **(a)**. **c**, Quantification of MFI for MHCII expression in each condition from **(a)**. Each dot represents individual experiment. Data are mean \pm s.e.m.; two-tailed unpaired Student's t test. * $p < 0.5$, ** $p < 0.01$, *** $p < 0.001$. **d**, Flow cytometry analysis of *Msr1* expression in macrophages cultured with RFP⁺ AT2 cells and WT mesenchyme with DMSO or Gefitinib. **e**, Quantification of MFI for MHCII expression in each condition from **(d)**. Each dot represents individual experiment. DMSO (n=4) and Gefitinib (n=4). Data are mean \pm s.e.m.; two-tailed unpaired Student's t test. ** $p < 0.01$.

16. In the experiment showed in Fig 4j, Areg is added together with Ereg, is Areg alone not enough to see an effect on macrophages? The effect of Areg in mesenchymal cells is very obvious (Extended Data Fig 5b). If reprogrammed fibroblasts drive AM changes, Areg alone should be enough. Please explain this point. For the experiments in Fig 4h and 4j, please provide images showing macrophage expansion and MHC-II expression.

As suggested, we directly tested the effect of Areg alone on AMs. In co-cultures of WT-AMs with WT-mesenchymal cells and WT-AT2 cells, Areg treatment alone was sufficient to promote macrophage expansion and reduce MHC-II expression (**Revised Ex Fig. 10f-j**; reproduced in **Rev3-Fig. 19a-c**). Furthermore, Areg alone induced changes in key reprogramming-associated genes, including *Arg1*, *Ym-1* and *Tnfa*, at levels comparable to the combined Areg and Ereg condition (**Revised Ex Fig. 10n**; reproduced in **Rev3-Fig. 19d**). These results indicate that Areg by itself is sufficient to drive the expansion and reprogramming of AMs in the context of mesenchymal-AT2 co-cultures. Flow cytometry data demonstrating macrophage expansion and MHC-II downregulation in response to Areg have been included in the revised manuscript.

Reviewer3_Figure 19. Effect of Areg on the reprogramming of alveolar macrophages co-cultured with fibroblasts and AT2 organoids. **a**, Experimental scheme to investigate the effect of Areg on alveolar macrophages co-cultured with fibroblasts and AT2 organoids. **b**, Flow cytometry analysis showing the expansion of alveolar macrophages (left) and quantification of frequency (right). Each dot represents individual experiment. **c**, Flow cytometry showing MHCII expression on alveolar macrophages cultured with Areg (left) and quantification of MHC mean fluorescence intensity (MFI). Each dot represents individual experiment. **d**, qPCR analysis of *Arg1*, *Ym-1*, and *Tnfa* expression in macrophages co-cultured with fibroblasts and AT2 organoids treated with PBS, Areg alone, or Areg/Ereg. Data are mean \pm s.e.m.; two-tailed unpaired Student's t test. * $p < 0.05$, ** $p < 0.01$, *** $p < 0.001$.

17. “Notably, mutant organoids grown without fibroblasts exhibited no response to Gefitinib treatment”. Which response is referring to? Do they grow? Do they generate DATP-like cells? In Fig 4m, which is the state of RFP+ cells? Are they AT2 cells expressing SPC? Or they advanced to another state, for instance cd177+ cells? Please, characterize them in more detail co-staining with other epithelial markers, and with Ki67

We apologize for the confusion and thank the Reviewer for raising this important point. Our intention was to convey that mutant AT2 organoids cultured without fibroblasts do not exhibit any detectable phenotypic change in response to Gefitinib, in contrast to co-cultures containing fibroblasts. Below, we provide clarification and additional characterization as requested.

To determine whether mutant AT2 cells directly promote fibroblast remodelling through EGFR activation, we first established organoid co-cultures of wildtype alveolar fibroblasts with RFP+ mutant AT2 cells and treated them with Gefitinib. Fibroblasts co-cultured with mutant AT2 cells acquired clear fibrotic phenotypes, which were markedly inhibited by Gefitinib, suggesting that tumour-derived Areg activates fibroblast EGFR to induce fibrotic reprogramming (**Revised Ex Fig. 9a, b**). We then investigated whether blocking this mutant AT2-fibroblast signalling affects the epithelial reprogramming of mutant cells themselves. Consistent with our previous findings and *in vivo* data, RFP+ mutant AT2 cells formed tumour organoids enriched for Sox9+ DATP-like states¹². However, Gefitinib treatment in co-cultures substantially reduced Sox9+ DATP-like cells and shifted the population toward Lpcat1+ AT2 cells, coinciding with inhibition of fibroblast remodelling (**Revised Ex Fig. 11a-e**; reproduced in **Rev3-Fig. 20a, b**). Notably, organoid number and size were unaffected, suggesting that fibrotic fibroblasts primarily sustain mutant epithelial reprogramming rather than promote proliferation (**Rev3-Fig. 20c, d**).

To determine whether Gefitinib acts directly on mutant AT2 cells, we treated RFP+ tumour organoids without fibroblasts. In this setting, Gefitinib had no detectable effect: organoids retained enriched Sox9+ DATP-like states at levels comparable to DMSO controls (**Revised Ex Fig. 11f-h**). Thus, the “no response” refers specifically to the lack of change in DATP-like state composition or organoid growth in the absence of fibroblasts. These findings support an indirect mechanism whereby Areg from mutant cells acts primarily via fibroblasts to generate the cues required to sustain epithelial reprogramming.

Finally, we have incorporated new *in vivo* analyses showing that both Gefitinib-treated *Red2Kras* lungs and Areg-deficient lungs exhibit reduced tumour growth, a marked loss of Sox9+ DATP-like and CD177+ populations, and an increase in Lpcat1+ and Sftpc+ AT2 cells with limited niche

remodelling (**Revised Fig. 5; Revised Ex Fig. 13**; see the details below as well). Altogether, these data demonstrate that Areg-EGFR signalling is essential for tumour initiation by driving fibroblast reprogramming and sustaining mutant epithelial reprogramming. These clarifications and new data have been incorporated into the revised manuscript.

Reviewer3_Figure 20. Areg-EGFR signalling mediates reciprocal reprogramming between fibroblasts and mutant AT2 cells. **a**, Representative confocal images of RFP⁺ mutant organoids co-cultured with fibroblasts and treated with DMSO or Gefitinib. RFP (red, tumours), Lpcat1 (green), DAPI (blue). Scale bar, 50µm **b**, Quantification of the percentage of Lpcat1⁺ cells shown in (a). Data represent mean \pm s.e.m.; each dot corresponds to an individual organoid, from a total of 3 independent experiments. Statistical significance was determined by two-tailed Mann Whitney test. **** $P < 0.001$. **c**, Representative images of RFP organoids grown in co-culture and treated with either DMSO or Gefitinib. Dotted line represents the edge of the Matrigel dome. **d**, Quantification of 2D surface area of organoids from (c). Each dot represents an individual organoid.

18. Similarly, a deeper characterization of RFP⁺ cells after Kras inhibition (MRTX1133, Fig 4o-q) and EGFR signaling inhibition (Gefitinib, Ext Data Fig 6) *in vivo* is required. Are AT2 and AT1 cell numbers recovered compared to a WT lung? What are RFP⁺ cells? Co-stain with different markers for the different epithelial states in these models and/or show the distribution of the clusters as in Figure 5n.

We thank the Reviewer for this critical feedback. To further define the epithelial states of RFP⁺ mutant cells following Kras inhibition, we performed IF staining for multiple lineage markers. MRTX1133 treatment resulted in a pronounced reduction of both DATP-like and Cd177⁺ reprogrammed states, accompanied by an increase in both AT2 and AT1 cell populations (**Revised Fig. 4l-p**; reproduced in **Rev3-Fig. 21a-e**). Notably, the AT2-to-AT1 cell ratio in MRTX1133-treated lungs approximated 2:1, closely resembling that observed in homeostatic lungs (**Revised Fig. 4o, p**; reproduced in **Rev3-Fig. 21d, e**)¹⁵. These findings indicate that Kras inhibition restores a homeostatic-like phenotypic balance between AT2 and AT1 cells. Importantly, as shown in our original submission, targeting reprogrammed tumour cells also reversed niche remodelling, leading to a marked reduction in fibrotic fibroblasts and diminished macrophage expansion and reprogramming (**Revised Fig. 4q-t**; reproduced in **Rev3-Fig. 21f-i**). Together, these data support that DATP reprogramming cooperates with the surrounding niche to sustain tumour-promoting microenvironments.

Reviewer3_Figure 21. Kras inhibition restores epithelial phenotypic homeostasis and restore niche remodelling. **a**, Experimental design for *Kras*^{G12D}-specific inhibitor administration. *Red2Kras* mice received two doses of tamoxifen to induce tumour formation. Four weeks post induction, mice were treated with MRTX1133 (15mg/Kg) twice daily for 10 days. Untreated samples were collected prior to treatment initiation. **b-d**, Representative confocal images showing lineage-labelled cells and epithelial cell states in MRTX1133-treated and untreated lungs. **(b)** DAPI (blue), Sox9 (yellow), RFP (red, tumours). Scale bar, 50 μ m. **(c)** DAPI (blue), Cd177 (yellow), RFP (red, tumours). Scale bar, 50 μ m. **(d)** DAPI (blue), Ager (yellow), Lpcat1 (grey), RFP (red, tumours). Scale bar, 50 μ m. **e**, Quantification of epithelial cell states among RFP⁺ cells from untreated and MRTX1133-treated mice. Data represent mean \pm s.e.m.; each dot corresponds to an individual mouse. Untreated ($n = 3$), MRTX1133 ($n = 3$). Statistical significance was determined by two-tailed unpaired Student's t test. * $P < 0.05$; ** $P < 0.01$; *** $P < 0.005$. **f, g**, Representative confocal images showing lineage-labelled cells with reprogrammed fibrotic fibroblasts **(f)** and macrophages **(g)** in MRTX1133-treated and untreated lungs. **h, i**, Quantification of macrophage expansion **(h)** and reprogramming **(i)** in MRTX1133 treated and untreated lungs. Data presented as mean \pm s.e.m., each dot represents an individual mouse. Untreated ($n = 3$), MRTX1133 ($n = 3$). Statistical significance was determined using a two-tailed unpaired Student's t test. * $P < 0.05$; ** $P < 0.01$.

Next, we further characterised the epithelial states of RFP⁺ mutant cells following EGFR inhibition. Gefitinib treatment in *Red2Kras* lungs markedly reduced tumour growth, accompanied by a decrease in reprogrammed epithelial cell states, including Sox9⁺ DATP-like and Cd177⁺ populations, and a concomitant increase in AT2 cells (**Revised Fig. 5a-c; Revised Ex Fig. 13f-i; reproduced in Rev3-Fig. 22a-d**). Furthermore, Gefitinib treatment suppressed fibrotic fibroblast reprogramming as well as macrophage remodelling (**Revised Fig. 5d, e; Revised Ex Fig. 13a-e**). Collectively, these findings demonstrate that inhibition of Areg-EGFR signalling disrupts tumour cell-driven fibrotic niche remodelling required to sustain reprogrammed epithelial states and shape the associated immune landscape.

Reviewer3_Figure 22. EGFR inhibition disrupts sustained mutant epithelial states. a-c, Representative confocal images showing cellular states of RFP⁺ mutant cells in DMSO- and Gefitinib-treated *Red2Kras* lungs. **(a)** DAPI (blue), Sox9 (grey), RFP (red, tumours). Scale bar, 50 μ m; **(b)** DAPI (blue), Cd177 (grey), RFP (red, tumours). Scale bar, 50 μ m; **(c)** DAPI (blue), Ager (grey), LpCAT1 (yellow), RFP (red, tumours). Scale bar, 50 μ m. **d,** Quantification of epithelial cell states within RFP⁺ populations in DMSO- and Gefitinib-treated *Red2Kras* lungs. Data represent mean \pm s.e.m.; each dot corresponds to an individual mouse. DMSO ($n = 2$), Gefitinib ($n = 3$). Statistical significance was determined by two-tailed unpaired Student's t test. * $P < 0.05$.

19. L296. “Moreover, Gefitinib treatment decreased the number of reprogrammed mutant cells compared to control (Extended Data Fig. 6i). These data demonstrate that EGFR activation is required for fibroblast reprogramming, which drives macrophage remodelling and supports tumour expansion *in vivo*”. It is unknown whether the effect derived from Gefitinib treatment on mutant AT2 cells is direct or indirect due to lack of fibroblast reprogramming, and it is not demonstrated in this context that changes in macrophages are induced by fibroblast reprogramming.

In vivo it is difficult to understand what happens first. Additional earlier timepoints during the course of the treatments or the genetic perturbations will help clarifying the changes in each cellular compartment contributing to dissect the mechanism.

We appreciate the Reviewer's comment. We agree that dissecting temporal relationships *in vivo* is challenging given the complexity of the tumour microenvironment, and a full perturbation time-course would not reliably establish causality, as it would inevitably interfere with the earliest tumour–fibroblast interactions at tumour onset. To overcome these limitations, we combined clonal-resolution spatiotemporal analyses *in vivo* with mechanistic multi-cell-type organoid co-cultures, allowing us to dissect the sequence of events with both spatial fidelity and functional precision.

As detailed in our revised manuscript and responses above, Pdgfr β ⁺Runx1⁺ fibroblasts appeared as early as 1 week post-oncogenic activation, in proximity to nascent RFP⁺ mutant cells, whereas Msr1⁺ macrophage expansion and reprogramming emerged from 2-4 weeks. These data indicate that fibroblast remodelling precedes macrophage changes during early tumorigenesis (**Revised Fig. 3**). scRNA-seq analyses show that Areg is produced primarily by DATP-like mutant cells, while EGFR is expressed across fibroblasts, macrophages, and epithelial cells. Functionally, Areg directly induces fibrotic reprogramming in fibroblasts, but not in macrophages or tumour cells, and this response is abolished by Gefitinib (**Revised Fig. 4; Revised Ex Fig. 8-10**), demonstrating that mutant cells act first through EGFR-dependent fibroblast activation.

Importantly, Areg-reprogrammed fibroblasts then drive macrophage expansion and phenotypic rewiring via the Tnc–TLR4 axis (**Revised Fig. 3**), establishing that macrophage changes arise downstream of fibroblast remodelling. *In vivo*, Gefitinib treatment reduces tumour burden and Sox9⁺ DATP-like states while restoring Lpcat1⁺ AT2 identity in parallel with inhibition of fibrotic fibroblast remodelling. By contrast, Gefitinib has no effect on mutant AT2 organoids cultured without fibroblasts, confirming that EGFR inhibition acts indirectly, through loss of fibroblast-derived cues.

Together, these integrated *in vivo* and *in vitro* analyses define a sequential and mechanistically coupled cascade in which DATP-like mutant epithelial states act as the initiating signalling hub, secreting Areg to activate fibroblast EGFR and trigger fibrotic reprogramming. These reprogrammed fibroblasts, in turn, sustain mutant epithelial plasticity and drive macrophage remodelling. Our combined approach provides a robust mechanistic framework demonstrating that fibroblast reprogramming arises downstream of mutant epithelial signalling and then functionally instructs both macrophage remodelling and the maintenance of reprogrammed epithelial states.

20. *In vivo* administration of Gefitinib to Red2Kras since day 0, impedes the formation of DATP-like cells at day 4? or their production of Areg at day 7 when Pdgfrb⁺ Runx1⁺ fibroblast emerge? *In vivo* Gefitinib treatment results in a dramatic decrease in RFP⁺ cells (Ext Data Fig 6). However, *in vitro* experiments (Fig 4g, 4m) show that they stopped expressing DATP-like markers but the size of the organoid is similar to that of the control. Please, address these discrepancies. Would a longer treatment result in mutant cell death?

We thank the Reviewer for this thoughtful question. In our *in vivo* experiments, Gefitinib was administered at day 6 after first tamoxifen induction, a time point when DATP-like states and Areg expression are already detectable (**Revised Fig. 5a**; **Revised Fig. 4d**). Blocking EGFR signalling at this stage prevents the emergence of Pdgfrb⁺Runx1⁺ fibrotic fibroblasts (normally evident by day 7), thereby interrupting the Areg-EGFR-driven epithelial-stromal circuit required to induce and sustain DATP-like states. Consequently, DATP-like cells are markedly reduced, with a sequential reversal of niche remodelling: immune and fibroblast features regress toward homeostatic states, and Lpcat1⁺ AT2 identity is restored. This coordinated loss of both epithelial and stromal reprogramming likely explains the pronounced reduction in RFP⁺ tumour expansion observed *in vivo*. These findings align with our prior *in vivo* clonal analyses, which showed mutant cell expansion proceeds through stochastic fate transitions, but that sustaining reprogrammed states requires ongoing niche support¹².

As addressed above, our *in vitro* data reinforce these conclusions. In fibroblast co-cultures, EGFR inhibition abolishes the DATP-like programme in mutant AT2 cells, leading to dominance of Lpcat1⁺ AT2 states and preventing fibrotic fibroblast reprogramming. Importantly, organoid size remains unchanged even as cell identity was reset, suggesting that loss of EGFR signalling uncouples epithelial reprogramming from proliferation and underscoring the instructive, rather than proliferative, role of reprogrammed fibroblasts.

21. To demonstrate that reprogrammed fibroblasts support mutant AT2 cells, they should be directly targeted. Ablation of Pdgfrb⁺ fibroblast using a DTA system, or deletion of Egfr in Pdgfrb⁺ fibroblast will help clarifying this point analyzing the effects on AM and on reprogrammed AT2. What happen to RFP⁺ DATP-like cells in this context? Is there tumor expansion? Do AMs expand or reprogram?

We thank the Reviewer for this important question. To investigate the functional role of fibroblasts in tumour development, we orthotopically engrafted RFP⁺ tumour organoids into the lungs of *Pdgfra-Cre^{ERT2}; Rosa-ZsGreen; Rosa-DTR* mice, enabling both lineage-tracing of Pdgfa⁺ resident fibroblasts and their selective depletion upon intratracheal diphtheria toxin (DT) administration

(**Revised Ex Fig. 12a**; reproduced in **Rev3-Fig. 23a**). Two weeks after organoid engraftment, DT was administered every other day for 14 days, and tissues were collected 19 days after first injection. In PBS-treated controls, lineage-labelled fibroblasts were found adjacent to RFP⁺ tumour cells and exhibited elevated Pdgfr β expression, consistent with tumour-induced fibrotic reprogramming of resident Pdgfra⁺ fibroblasts (**Revised Ex Fig. 12b**; reproduced in **Rev3-Fig. 23b**). In contrast, DT-mediated fibroblast depletion markedly reduced the expansion of lineage-labelled Pdgfr β ⁺ fibroblasts, impeded tumour growth, diminished reprogrammed Sox9⁺ mutant cells, accompanied by reduced macrophage expansion around RFP⁺ tumour regions (**Revised Ex Fig. 12c, d**; reproduced in **Rev3-Fig. 23c, d**). These data demonstrate that fibrotic reprogrammed fibroblasts originated from resident Pdgfra⁺ fibroblasts are critical for both tumour development and the establishment of the associated immune niches.

Reviewer3_ Figure 22. Fibroblasts form a critical niche that supports tumour development and immune remodelling. **a.** Experiment scheme of orthotopic engraftment of tumour organoids into the lungs of *Pdgfra-Cre^{ERT2}; Rosa-ZsGreen; Rosa-DTR* mice. **b-d.** Representative confocal images showing fibroblast reprogramming (**b**), macrophage expansion (**c**), and mutant epithelial cell reprogramming (**d**) in PBS-treated controls, all of which were markedly impeded in DT-treated lungs. RFP (red, tumours), ZsGreen (lineage-traced fibroblasts), Pdgfr β (white, b), CD68 (white, c), Sox9 (white, d) and DAPI (blue), scale bar, 100 μ m. Images representative of $n = 2$ mice.

22. To better understand what happen to mutant AT2 cells when Areg is inhibited, a timecourse is required. This will help to understand the increase in RFP+ AT2+ cells at the expense of RFP+ DATP-like cells and AT1 cells, together with the changes in fibroblasts and macrophages. Is EGFR activated in alveolar fibroblasts adjacent to targeted AT2 cells? CD177+ cells do not change, what can this mean?

We thank the Reviewer for these thoughtful comments. While a full *in vivo* time-course might provide additional granularity, it would not reliably establish causality because oncogenic activation and niche engagement occur asynchronously across clones, making it difficult to align perturbation timing with specific stage-matched cellular transitions. Instead, we addressed these questions using a combination of spatiotemporal clonal analyses, epithelial- and fibroblast-focused perturbations using organoid co-culture systems, and refined scRNA-seq interrogation, which together provide a coherent mechanistic framework.

In response to the Reviewer's question about *Cd177*⁺ cells, we carefully re-analyzed our scRNA-seq datasets from *Areg-flox/+* and *Areg-flox/flox* lungs to evaluate the *Cd177*⁺ population. We noted that original annotated *CD177*⁺ cluster contained only a small subset of cells expressing *Cd177* and its associated marker *Cd38*, suggesting over-clustering had obscured accurate comparison (**Rev3-Fig. 24a, b**). To resolve this, we re-clustered all RFP⁺ cells using a higher resolution and re-annotated epithelial states using the same marker genes. This produced a smaller, more distinct *Cd177*⁺ population with uniform *Cd177* and *Cd38* expression, while preserving the distribution of other states (**Rev3-Fig. 24c**). Using this refined annotation, we observed an almost two-fold reduction in the proportion of *Cd177*⁺ cells in *Areg-flox/flox* lungs compared to *Areg-flox/+* controls (6.7% vs. 11.4%), together with a marked loss of DATP-like states and a reciprocal increase in *Lpcat1*⁺ AT2 cells. These results confirm that Areg deletion shifts mutant AT2 cells away from reprogrammed states and back toward AT2 identity.

Altogether, these integrated analyses demonstrate that loss of *Areg* disrupts the mutant epithelial-fibroblast signalling axis, leading to failure of fibroblast EGFR activation, prevention of fibrotic niche formation, and consequent destabilisation of reprogrammed tumour epithelial states and immune cell remodelling. We have incorporated these clarifications and refined analyses into the revised manuscript.

Reviewer3_ Figure 24. Areg deletion in mutant AT2 cells reduces reprogrammed cell states and restores AT2 cell population in *Red2Kras* lungs. **a**, Original UMAP plot showing all lineage-labelled RFP⁺ cells isolated from *Areg*^{flox/+} and *Areg*^{flox/flox} lungs, clustered into five distinct cell states (first Manuscript version). Cells are coloured and numbered by state. **b**, FeaturePlot showing the expression of *Cd177* and *Cd38* within the cluster of *Cd177*⁺ cells. **c**, New UMAP of lineage-labelled RFP⁺ cells showing a specific *Cd177*⁺ cluster with more uniform expression of *Cd177* and *Cd38*. Cells are coloured and numbered by state. **d**, Percentage of RFP⁺ cells distributed across each cluster defined in (c). Percentages smaller than 5% are omitted from the bar plots.

23. In human LUAD tissue (Fig 6), CTHRC1⁺, ACTA2⁺ fibroblasts are not found within the tumor but in the periphery. Do they express PDGFRB or Lcn2? Do they show EGFR activation? Do KRT8⁺ cells express DATP-like markers and produce AREG? How are AMs in this context?

In response to the Reviewer's comment, we performed additional analyses on confirmed early-stage KRAS^{G12D} LUAD and matched background samples. IF staining confirmed the presence of KRT8⁺ DATP-like cells that co-express SOX9 and show high AREG expression (**Revised Fig. 6g, h; Revised Ex Fig. 16h**; reproduced in **Rev3-Fig, 25a, b**). Within the same samples, ACTA2⁺ fibrotic fibroblasts were spatially located inside tumour regions and closely associated with mutant KRT8⁺ cells (**Revised Fig. 6h**; reproduced in **Rev3-Fig, 25c**). scRNA-seq analysis further identified a distinct fibroblast cluster enriched for *ACTA2*, *PDGFRB*, *CTHRC1*, and *RUNX1*, confirming their fibrotic identity (**Revised Ex Fig. 17a-e**). In addition, we also detected separate fibroblast clusters (cluster 2 and cluster 5) exhibiting inflammatory features, characterized by high expression of *CXCL14*, *CCL2* and *SFRP4*, marker genes previously associated with inflammatory fibroblasts in IPF lungs (**Rev3-Fig. 25d-f**)¹⁴. Of note, these inflammatory markers were also

upregulated in matched background tissues within this scRNA-seq datasets, likely reflecting sampling from peri-tumoral regions during surgical resection. This observation aligns with our prior *in vivo* study, which demonstrated widespread reprogramming of wildtype epithelial cells across *Red2Kras* lungs even during early onsets¹². Furthermore, it also supports our current findings that inflammatory fibroblasts are enriched in peri-tumour areas, while fibrotic fibroblasts are primarily localised within the tumour itself, consistent with spatial patterns seen in *Red2Kras* lungs.

Reviewer3_ Figure 25. Characterisation of early human lung adenocarcinoma. a-c, Representative confocal images of early stage *KRAS*^{G12D} LUAD tissue and matched background lung. **(a)** DAPI (blue), KRT8 (green), SOX9 (red). **(b)** DAPI (blue), KRT8 (green), AREG (red). **(c)** DAPI (blue), KRT8 (green), ACTA2 (red). Scale bar, 50 μ m. **d,** UMAP plot showing all sub-clustered fibroblasts annotated into six distinct populations. Cells are coloured and numbered by population. **e,** FeaturePlots showing the expression of key inflammatory fibroblasts markers. **f,** UMAP plot showing the distribution of fibroblasts in normal and LUAD lung tissues. Dotted lines highlight a cluster with inflammatory fibroblast identity. Cells are coloured according to dataset of origin.

As requested, we further analysed macrophage populations within the human EGFR WT LUAD scRNA-seq dataset. Sub-clustering of all CD68⁺ cells identified several macrophage subsets, among which cluster_1 was specifically enriched in LUAD-derived samples (**Rev3-Fig, 26a-c**). Gene expression analysis revealed that cluster_1 presented an alveolar macrophage identity characterised by high *MERTK*, *MARCO* and *FCGR1A* expression (**Rev3-Fig, 26d**). While reprogramming-associated genes such as *MSR1* and *MRC1* were not exclusive to cluster_1, this cluster uniquely expressed multiple cytokines and inflammatory mediators, including *CCL3*, *CCL2*, *CCL13*, and showed enrichment of *CXCL2*, a neutrophil recruitment factor also upregulated in reprogrammed macrophages in our *Red2Kras* model (**Rev3-Fig, 26e**). Importantly, cluster_1 was detected in stage I LUAD samples, supporting the emergence of a reprogrammed alveolar macrophage-like state early in human tumour development (**Rev3-Fig, 26f**).

Reviewer3_ Figure 26. Emergence of reprogrammed macrophage states during early human LUAD. **a**, FeaturePlot showing the expression of the broad macrophage marker *CD68* in the UMAP dataset for all cells obtained from early *EGFR-WT* LUAD and matched non-tumour tissues. **b**, UMAP plot showing all sub-clustered macrophages annotated into nine distinct populations. Cells are coloured and numbered by population. **c**, UMAP plot showing the distribution of macrophages in normal and LUAD lung tissues. Dotted lines highlight a cluster specific of the LUAD dataset. Cells are coloured according to dataset of origin. **d, e**, Feature plots showing the expression of alveolar macrophage markers (**d**) and several inflammatory chemokines (**e**). **f**, UMAP plot showing the distribution of macrophages. Cells are coloured according to the LUAD Stage.

24. Please, characterize RFP+ cells during the course of human organoid formation, confirm AREG-EGFR effects and incorporate AMs to the analysis.

We thank the Reviewer for this valuable suggestion. As suggested, we analyzed the human alveolar organoid transduced with *KRAS*^{G12D}, representing – to our knowledge – the first human primary AT2-derived organoid model enabling direct reconstruction of early LUAD transitions (**Revised Fig. 6i**; reproduced in **Rev3-Fig. 27a**). This system, combined with our scRNA-seq profiling, provides a highly valuable and unprecedented experimental platform for dissecting the earliest steps of human LUAD initiation, which has remained technically inaccessible to patient samples.

Induction of *KRAS*^{G12D} led to a marked enlargement of organoids and increased *KRAS* expression (**Revised Fig. 6j, k**; reproduced in **Rev3-Fig. 27b, c**). To further characterise the cellular transitions of human AT2 (hAT2) cells upon *KRAS* activation, we performed scRNA-seq analysis of *KRAS*^{G12D}-hAT2 organoids and control hAT2 organoids. While control hAT2 organoids were largely composed of *SFTPC*⁺ AT2 cells, *KRAS* activation reprogrammed hAT2 cells toward distinct transitional states characterised by reduced *SFTPC* expression, consistent with our observations in *Red2Kras* lungs (**Revised Fig. 6l-n**; reproduced in **Rev3-Fig. 27d, e**). Notably, *KRAS*^{G12D}-hAT2 organoids also retained subsets of *SFTPC*⁺*SCGB3A2*⁺ cells, corresponding to AT0 cells, and *SFTPC*⁻*SCGB3A2*⁺*SCGB1a1*⁻ cells, corresponding to TRB-SCs, both of which have been reported to originate from AT2 cells in human distal lungs^{16,17}. Notably, we identified a distinct population of *AREG*^{high} cells co-expressing *SOX9*, *KRT8*, and *ITGA2*, indicative of DATP-like states, demonstrating that this transitional programme is conserved in human cells (**Revised Fig. 6n**; reproduced in **Rev3-Fig. 27e**).

To determine whether AREG-EGFR signalling mediates the reprogramming of fibroblasts, we co-cultured *KRAS*^{G12D}-hAT2 cells with human lung fibroblasts isolated from normal donors. This co-culture induced the expansion of *PDGFRB*⁺ fibroblasts around tumour cells, which was abrogated by Gefitinib treatment (**Revised Fig. 6o**; reproduced in **Rev3-Fig. 27f**). These findings indicate that *KRAS*^{G12D}-driven reprogramming of AT2 cells into DATP-like states is conserved during early tumorigenesis between mouse and human lungs, and that this epithelial transition contributes to fibroblast activation via the AREG-EGFR axis.

We also attempted co-cultures incorporating alveolar macrophages derived from human donors. However, this approach was technically challenging, as MHC/HLA compatibility requires isolation of alveolar macrophages from the same donor source, and collecting sufficient BALF from normal donors was not feasible.

Reviewer3_ Figure 27. *KRAS*^{G12D}-driven reprogramming of human AT2 cells drives fibrotic niches. **a**, Schematics of inducible viral constructs and experimental design for establishing inducible *KRAS*^{G12D}-human alveolar organoid. **b**, RFP expression in human alveolar organoids; control versus *KRAS*^{G12D}-transduced organoids. **c**, qPCR analysis showing increased *KRAS* expression upon doxycycline treatment. **d**, UMAP visualization showing diverse cell states derived from human AT2 cells following *KRAS*^{G12D} induction. **e**, Dotplot showing key marker genes defining the indicated cell populations. **f**, Representative IF images showing expansion of PDGFRβ⁺ fibroblasts around *KRAS*^{G12D}-expressing organoids and suppression of fibrotic remodelling upon EGFR inhibition. RFP (tumour organoid), PDGFRβ (green), PDGFRα (white), and DAPI (blue), scale bar, 100 µm.

Minor comments:

1. The gate in Fig. 2h needs to be the same for Confetti and for Red2Kras. Also, in 2i, flow panel for CD3 is not shown. Please, show all of them.

We thank the Reviewer for the comment. We have now ensured that the gating strategy in original Fig. 2h (**Revised Ex Fig. 2k**) is identical for both *Confetti* and *Red2Kras* lungs. In addition, we have included the previously missing flow cytometry panel for CD3 in original Fig. 2i (**Revised Ex Fig. 2o**). The revised figure now displays all relevant panels, and the gating and analysis were applied consistently across all samples. These changes have been incorporated into the revised manuscript.

2. L215,216: "...suggests that fibrotic signals dominate over inflammatory cues during early tumour development, as recently described in regeneration". The authors recently demonstrated that *Il1r1* is required for driving tumor formation via mutant AT2 cellular reprogramming (England et al, CSC 2025). Please, re-phrase or clarify what you mean exactly.

We thank the Reviewer for raising this important point and apologize for any confusion. To clarify, our statement that "fibrotic signals dominate over inflammatory cues" refers specifically to fibroblast fate decisions, not to the role of IL-1 β in mutant epithelial reprogramming.

As elegantly demonstrated by Tsukui et al., strong fibrotic cues such as TGF β convert inflammatory fibroblasts into *Cthrc1*⁺ fibrotic fibroblasts while simultaneously repressing inflammatory gene expression during tissue repair and remodeling¹⁴. Consistent with this model, our data show a spatially compartmentalised organisation of fibroblast states during early tumour development: fibrotic fibroblasts, induced by tumor-derived Areg, localise predominantly to tumor cores, whereas inflammatory fibroblasts are found in peri-tumoral regions, likely in response to signals from reprogrammed alveolar macrophages mediated by IL-1 β . Thus, within fibroblasts, elevated levels of Areg-mediated fibrotic signaling suppress IL-1 β -driven inflammatory programs, explaining the exclusion of inflammatory fibroblasts from tumour cores.

Importantly, this fibroblast-specific regulatory hierarchy does not conflict with the role of IL-1 β in tumour cells. As the Reviewer noted, and as shown in our prior study^{12,18}, IL-1R1 signalling is essential for maintaining the reprogrammed state of mutant AT2 cells, and IL-1 β from neutrophils/IMs contributes to sustaining tumour-cell reprogramming within tumour cores.

In summary, our data indicate that fibrotic cues decisively govern the fate and function of fibroblasts by overriding inflammatory signals, whereas IL-1 β remains indispensable for epithelial (tumour) cell reprogramming. We have clarified this distinction in the revised manuscript to address the Reviewer's concern.

3. It is surprising to see that *Pdgfra* is still expressed in reprogrammed fibroblasts. Previously, it has been shown that *Pdgfrb*⁺ fibrotic fibroblasts stopped expressing *Pdgfra* 28 days after injury (Jones et al., Nature 2024). Could this be related to reversibility-persistence in the tissue of these fibroblasts? Just comment on these differences.

We apologise for this confusion. As we detailed above and in our revised manuscript, *Pdgfra* expression is downregulated while *Pdgfr β* expression is upregulated, consistent with a transition from homeostatic to fibrotic fibroblast states (**Revised Fig. 1i**). These observations align with our IF and scRNA-seq analyses and support the notion of fibroblast reprogramming during early tumorigenesis. We have clarified this point in the revised manuscript and updated the relevant figures accordingly.

Referee #3 (Remarks on code availability):

The code is not available at the moment.

All transcriptomic datasets generated in this study are in the process of being deposited in a public repository. Accession numbers will be shared once available and can be provided upon request.

Referee #4 (Remarks to the Author):

In this study, the authors use mouse models and co-culture systems to investigate how the tumor and TME interact on a molecular and cellular level during tumor development to facilitate this process. The authors focus mostly on fibrotic niche formation and ligand/receptor interactions that may enable the TME to facilitate tumor development.

The models used in this study are very interesting and relevant. The overall area of study is important and timely. However, there are major limitations in terms of novelty and depth.

We thank the Reviewer for insightful and constructive feedback. In response, we have substantially expanded the molecular, spatial, mechanistic, and cross-model scope of our study. Our enhanced findings establish that nascent DATP-like mutant epithelial states serve as early signalling hubs, orchestrating the formation of a fibrotic, pro-tumorigenic niche via Areg–EGFR-mediated reprogramming of fibroblasts. This early fibrotic niche plays a causal role in sustaining reprogrammed tumour cell states and promoting immune remodeling through a Tenascin-C–TLR4 signalling axis, generating a multicellular epithelial–fibroblast–immune circuit. Using clonal-resolution time-course imaging, single-cell transcriptomics, organoid co-cultures, and lineage tracing, we delineate the sequential and cooperative interactions among these compartments. Importantly, these mechanisms are validated in human organoids and early-stage LUAD tissues, are conserved across distinct LUAD oncogenic drivers, and are observed in other cancer contexts, demonstrating a generalisable and previously unrecognized framework for EGFR-dependent niche initiation and early tumorigenesis. We believe that these comprehensive new analyses and mechanistic insights now fully address the Reviewer’s concerns and clarify the novelty and significance of our work.

-Overall, the findings are somewhat underwhelming, as the involvement of the AREG-EGFR pathway in cancer/tumor microenvironment interactions is already well established.

We thank the Reviewer for this comment. While the role of the Areg-EGFR axis in cancer and the microenvironment is increasingly appreciated, previous studies have largely focused on Areg produced by immune cells, particularly regulatory T cells, during tumour progression¹⁹⁻²⁵. In contrast, our study reveals that mutant epithelial cells themselves are an unrecognized and functionally pivotal source of Areg at the very onset of tumorigenesis. Specifically, we identify Areg expression within *Kras*^{G12D}-induced reprogrammed AT2-derived epithelial cell states, which act as early signalling hubs that initiate and coordinate niche reconstruction before overt tumour formation. We further demonstrate that Areg-EGFR signalling emanating from these nascent mutant epithelial states drives the emergence of a fibrotic, pro-cancer fibroblasts at the pre-neoplastic stage, and that this fibroblast state in turn plays a causal role in maintaining reprogrammed tumour cell states and establishing the early tumour-promoting niche. Importantly, AT2-specific deletion of Areg markedly impairs tumour growth and niche remodelling programs. These findings uncover a previously unappreciated mutant epithelial-driven mechanism of early tumour initiation through sequential reprogramming of fibroblasts and the immune ecosystem.

-Despite the known implication of the AREG-EGFR pathway, the authors do not mention or investigate that macrophages can also produce AREG, which has been described previously. Furthermore, it remains unclear whether macrophages can also influence fibroblasts and induce the formation of the fibrotic fibroblast phenotype observed.

As the Reviewer pointed out, Areg expression has been reported in immune cells, particularly in regulatory T cells and macrophages. To examine this in our system, we re-analysed *Areg* expression across our scRNA-seq datasets. Consistent with previous studies, *Areg* transcripts were detected in regulatory T cells and subsets of macrophages (**Revised Ex Fig. 7f**; reproduced in **Rev4-Fig. 1a**). However, *Krt8*⁺ DATP-like reprogrammed mutant cells exhibited markedly higher *Areg* expression than any other cell population. We further validated these findings by IF staining, which confirmed strong AREG expression in RFP⁺ tumour cells, with comparatively low expression in FOXP3⁺ Tregs and F4/80⁺ macrophages (**Rev4-Fig. 1b, c**). Functionally, as discussed above, selective deletion of *Areg* in AT2 cells significantly decreased tumour growth, underscoring that tumour-derived Areg is the dominant driver of early tumour initiation.

Reviewer4 Figure 1. DATP-like reprogrammed epithelial cells are the primary source of Areg in early *Kras*^{G12D} tumours. **a**, Violin plot of *Areg* expression across epithelial and immune cell populations. **b**, Representative IF images showing Areg expression in RFP⁺ tumour cells and macrophages. RFP (red, tumours), F4/80 (white), Areg (green), and DAPI (blue), scale bar, 100 μm. **c**, Representative IF images showing Areg expression in RFP⁺ tumour cells and regulatory T cells. RFP (red, tumours), Foxp3 (white), Areg (green), and DAPI (blue), scale bar, 100 μm.

In our original manuscript, we identified Lcn2⁺Pdgfra⁺ inflammatory fibroblasts primarily emerging at 4 weeks post *Kras* activation, coincident with macrophage expansion. To further dissect their interplay during tumour development, we performed co-staining for Pdgfra, Lcn2 and F4/80 in *Red2Kras* lungs from 1 to 8 weeks after induction. As observed previously, Lcn2⁺ inflammatory fibroblasts appeared at the periphery of RFP⁺ tumours at 4 and 8 weeks and were consistently positioned adjacent to clusters of expanded reprogrammed AMs, supporting potential crosstalk between two populations (**Revised Fig. 3m, n**; reproduced in **Rev4-Fig. 2a, b**). Given that *Red2Kras* AMs upregulate inflammatory cytokines, including *IL-1b*, in scRNA-seq data, we investigated whether IL-1β or AM-derived signals could modulate fibroblast states. Treatment of wildtype mesenchymal cells with IL-1β for 48 hours induced Lcn2 expression in Pdgfra⁺ fibroblasts

(Revised Ex Fig. 6e, f; reproduced in Rev4-Fig. 2c, d). Consistently, co-culture of wildtype mesenchymal cells with *Red2Kras* AMs also upregulated Lcn2, indicating that reprogrammed macrophages can directly activate inflammatory fibroblast states in the inter-tumour regions. Importantly, these Lcn2⁺Pdgfra⁺ fibroblasts did not express fibrotic markers such as Tnc and were rarely detected within tumour cores (Revised Ex Fig 6d). This supports a spatially compartmentalised organisation of fibroblast populations: fibrotic fibroblasts induced by tumour-derived Areg predominate within tumour centres, whereas inflammatory fibroblasts induced by AM-secreted IL-1 β localise to inter-tumour areas. This mirrors prior observations that fibrotic cues override inflammatory programmes in fibroblasts during injury repair, suggesting that these regulatory mechanisms are conserved in tumour-associated fibroblasts¹⁴.

Together, these analyses demonstrate that *Red2Kras* AMs directly drive inflammatory fibroblast activation, highlighting their specific and spatially restricted role in shaping the inter-tumour mesenchymal niche.

Reviewer4_Figure 2. Inflammatory fibroblasts driven by reprogrammed macrophages. a, Representative confocal images showing macrophages, inflammatory fibroblasts and lineage-labelled AT2 cells in *Red2Kras* lungs at 8 weeks after oncogenic activation. Lcn2 (yellow), Pdgfra (grey), F4/80 (green) and RFP (red). Scale bar, 50 μ m. **b**, Quantification of Lcn2⁺Pdgfra⁺ fibroblasts at both tumour periphery and inside. Data represent mean \pm s.e.m.; each dot corresponds to an individual mouse. *Confetti* ($n = 3$), *Red2Kras* 1w ($n = 3$), 2w ($n = 3$), 4w ($n = 3$) and 8w ($n = 3$). Statistical significance was determined by two-tailed unpaired Student's t test. * $p < 0.05$; ** $p < 0.01$. **c**, Schematic illustrating WT fibroblasts treated with IL-1 β or co-cultured with *Red2Kras* alveolar macrophages. **d**, Representative confocal images of fibroblasts treated with IL-1 β or co-cultured with *Red2Kras* AMs. Lcn2 (red), Pdgfra (green), F4.80 (grey) and DAPI (blue).

-The mechanisms by which reprogrammed fibroblasts enhance macrophage accumulation and tissue remodeling are not explored. The molecular pathways involved in this crosstalk should be addressed.

We thank the Reviewer for this important feedback. In response, we identified Tenascin-C (Tnc) as a key factor enriched in reprogrammed fibroblasts in our scRNA-seq analysis (**Revised Ex Fig. 5a**; reproduced in **Rev4. Fig. 3a**). Tnc is an ECM protein previously shown to promote tissue repair, angiogenesis, and tumor progression via modulation of immune cells^{26,27}. Notably, Toll-like receptor 4 (TLR4), a known Tnc-binding receptor, is selectively and highly expressed in AMs in scRNA-seq analysis, suggesting a potential molecular axis for fibroblast-macrophage crosstalk (**Revised Ex Fig. 5a**; reproduced in **Rev4. Fig. 3a**). IF staining further confirmed that Tnc is highly expressed in Pdgfr β ⁺ reprogrammed fibroblasts in direct contact with RFP⁺ mutant cells in *Red2Kras* lungs (**Revised Fig. 3i**; reproduced in **Rev4. Fig. 3b**). Specifically, Tnc⁺Pdgfr β ⁺ fibroblasts were observed contacting Cd68⁺ macrophages at the tumour periphery at 4 weeks following *Kras* activation, coinciding with major macrophage remodeling. To investigate their functional interactions, AMs isolated from wildtype lungs were treated with Tnc *in vitro*, which induced both proliferation and reprogramming of AMs, as indicated by increased *Msr1*, a marker of macrophage reprogramming, and *Ki67* expression (**Revised Ex Fig. 5b-e**; reproduced in **Rev4. Fig. 3c-f**). Notably, this effect was abrogated by the TLR4 inhibitor TAK-242, indicating a Tnc-TLR4-dependent mechanism for AM expansion and reprogramming. Furthermore, as shown in our original submission, co-culture of wildtype AMs with *Red2Kras*-derived mesenchymal cells, but not wildtype mesenchyme, induced AM reprogramming, which was abrogated by TLR4 inhibition (**Revised Fig. 3j-l**; reproduced in **Rev4. Fig. 3g-i**). These results identify Tnc as a reprogrammed fibroblast-derived cue that drives AM expansion and phenotypic remodeling through TLR4 signalling during early tumorigenesis. These new data have been incorporated in our revised manuscript.

Reviewer4_Figure 3. Tumour-induced reprogrammed fibroblasts remodel alveolar macrophages via Tnc-TLR4 axis. a, Dotplots showing the expression of Tnc in the population of reprogrammed fibroblasts and TLR4 in AMs. Each dot is coloured based on the average expression and sized based on the percentage of cells expressing that gene. **b**, Representative confocal image showing Tnc-expressing fibroblasts in the proximity of expanded macrophages in *Red2Kras* lungs at 4 weeks post-induction. Cd68 (yellow), Pdgfr β (green), TnC (grey) and RFP (red, tumour). Scale bar 50 μ m. Images representative of $n = 3$. **c**, Schematic illustrating cultures of wildtype AMs treated with Tnc or Tnc + TAK-242. **d**, Representative confocal images of Msr1⁺ AMs in culture after treatment with Tnc or Tnc+TLR4 inhibitor. DAPI (blue), Msr1 (yellow), Ki67 (red). Scale bar, 100 μ m. **e**, **f** Quantification of the percentage of reprogrammed (**e**) and proliferation (**f**) macrophages from (**d**). Data represent mean \pm s.e.m.; each dot corresponds to an independent experiment. Untreated ($n = 3$), Tnc-treated ($n = 3$) and Tnc+TAK-242 treated ($n = 3$). Statistical significance was determined by two-tailed unpaired Student's t test. * $p < 0.05$; ** $p < 0.01$. **g**, Schematic illustrating co-cultures of wildtype AMs with mesenchymal cells isolated from *wildtype* or *Red2Kras* lungs with treatment of TAK-242. **h**, Representative confocal images of Msr1⁺ AMs in the cultures from (**g**). DAPI (blue), Msr1 (red), F4.80 (green). Scale bar, 100 μ m. **i**, Quantification of the percentage of reprogrammed macrophages from (**h**). Data represent mean \pm s.e.m.; each dot corresponds to an independent experiment. WT ($n = 3$), *Red2Kras* mesenchyme ($n=4$) and *Red2Kras* mesenchyme + TAK-242 treated ($n = 4$). Statistical significance was determined by two-tailed unpaired Student's t test. * $p < 0.05$, ** $p < 0.01$.

-It is also unclear whether tumor cells can directly influence macrophages, contributing to their recruitment, survival, or phenotypic polarization. This needs to be examined.

To determine whether tumour cells can directly modulate AM phenotype or act primarily through fibroblasts, we established organoid co-cultures under three conditions: (1) wildtype (WT) AM alone, (2) WT-AMs with RFP⁺ tumour cells, and (3) WT-AMs with RFP⁺ tumour cells and WT-mesenchymal cells (**Revised Ex Fig. 10a, b**; reproduced in **Rev4-Fig. 4a, b**). Msr1 induction occurred only in the presence of both tumour cells and mesenchymal cells, but not tumour cells alone, demonstrating that mesenchymal cells are required for AM reprogramming (**Revised Ex Fig. 10c, d**; reproduced in **Rev4-Fig. 4d**). By contrast, AM expansion occurred in response to tumour cells even without mesenchymal cells, although was comparable to the condition including fibroblasts, suggesting that tumour cells may also contribute to AM expansion independently (**Revised Ex Fig. 10e**; reproduced in **Rev4-Fig. 4c**). Together, these findings delineate distinct yet cooperative roles of tumour cells and fibroblasts in shaping the alveolar macrophage compartment during early tumorigenesis, and demonstrate that mutant cell-driven fibrotic reprogramming of fibroblasts is essential for AM phenotypic remodelling.

Reviewer4_Figure 4. Reprogramming of alveolar macrophages is mediated by reprogrammed fibroblasts. **a**, Experiment scheme of organoid co-culture. **b**, Representative flow cytometry plots showing the populations of macrophages (CD45⁺EpCAM⁻), tumour cells (CD45⁻EpCAM⁺), and mesenchymal cells (CD45⁻EpCAM⁻) within organoid co-cultures. Numbers adjacent to gates indicate the percentage of each population. **c**, Quantification of relative macrophage numbers compared to AM alone controls. Each dot represents an independent experiment; statistical significance was assessed using a two-tailed unpaired Student's t-test. **d**, Flow cytometry analysis of MSR1 expression in AMs (left) and quantification of MFI across three different co-culture conditions (right). Each dot represents an independent experiment; statistical significance was assessed using a two-tailed unpaired Student's t-test. *** $p < 0.001$, ns, not significant.

-The paper does not sufficiently explore how alterations in alveolar macrophages promote tumor development beyond a reduction in antigen presentation. An experiment involving co-culturing tumor cells with alveolar macrophages previously conditioned by fibroblasts would be informative.

We thank the Reviewer for this important question, which was also raised by other Reviewer. In response, we further characterised reprogrammed AMs emerging during early tumour development. Our scRNA-seq analysis revealed that AMs exhibited the most pronounced transcriptomic divergence in *Red2Kras* lungs, forming a discrete cluster distinct from homeostatic AMs and IM/monocyte-derived macrophages on the UMAP (**Revised Fig. 2a, b**; reproduced in **Rev4_Fig. 5a, b**). Although these reprogrammed AMs retained core transcriptomic features with resident AMs based on canonical lineage markers, such as *SiglecF* and *MertK*, they were distinguished by elevated expression of genes such as *Msr1* and *Ch25h* (**Revised Fig. 2c**;

reproduced in **Rev4_Fig. 5c**). Notably, as detailed above, co-culture of AMs with reprogrammed mesenchymal cells promoted AM proliferation and upregulated *Msr1* expression, reflecting fibroblast-mediated phenotypic rewiring of AMs, through *Tnc*-TLR4 axis during early tumorigenesis.

Reviewer4_Figure 5. Phenotypic rewiring of reprogrammed macrophages. **a, b**, UMAP visualization of macrophage subsets in *Confetti* and *Red2Kras* lungs. **c**, Dotplot showing expression of key macrophage markers across subsets. **d**, Heatmap showing chemokines upregulated in reprogrammed macrophages relative to homeostatic alveolar macrophages, indicating enhanced potential for recruiting other immune cells.

Beyond alterations in antigen presentation programs, reprogrammed AMs exhibited elevated expression of chemokines critical for immune cell recruitment. In particular, *Cxcl2*, a ligand for *Cxcr2*, was markedly upregulated (**Revised Fig. 2d**; reproduced in **Rev4. Fig. 5c**). Notably, consistent with previous studies⁴, we confirmed that *Cxcr2* is predominantly expressed on neutrophils, a key population in lung tumour development (**Revised Ex Fig. 4d**; reproduced in **Rev4. Fig. 6f**)⁵⁻⁸. To investigate the functional role of these macrophages during early tumour development, we specifically depleted alveolar macrophages via intratracheal administration of clodronate liposomes following oncogenic activation (**Revised Fig. 2k**; reproduced in **Rev4_Fig. 6a**). Ablation of macrophages significantly inhibited tumour growth (**Revised Fig. 2l, m**; reproduced in **Rev4_Fig. 6b-d**). Importantly, this was accompanied by a marked reduction in neutrophil expansion and infiltration – key contributors to lung tumour development – suggesting that macrophages promote tumour growth likely through *Cxcl2*-*Cxcr2*-dependent neutrophil recruitment (**Revised Fig. 2n, p**; **Revised Ex Fig. 4e**; reproduced in **Rev4_Fig. 6e, g**)^{4,6,8}. In addition, reprogrammed macrophages expressed elevated levels of *Cxcl16* (**Revised Fig. 2d**; reproduced in **Rev4_Fig. 5c**), whose receptor, *Cxcr6*, is enriched in $\gamma\delta$ -T cells, a population critical for lung tumour development (**Rev4_Fig. 6f**)⁷. Notably, $\gamma\delta$ -T recruitment was markedly impaired in the absence of macrophages (**Revised Fig. 2o, q**; **Revised Ex Fig. 4f**; reproduced in **Rev4_Fig. 6e, h**). These results demonstrate that AM reprogramming is essential for early tumour development by orchestrating the recruitment of neutrophils and $\gamma\delta$ T cells. These data have been incorporated into the revised manuscript.

Together, our findings demonstrate that macrophage reprogramming involves both phenotypic and functional changes, promoting the recruitment of neutrophils and $\gamma\delta$ T cells to establish a tumour-promoting niche during early oncogenesis.

Reviewer4_Figure 6. Macrophages are essential for early tumour development. **a**, Experimental scheme to deplete macrophages by clodronate liposome treatment via intratracheal root. **b**, Representative confocal images showing alveolar macrophage depletion after clodronate liposome administration. **c**, Flow cytometry analysis and quantification of alveolar macrophages following treatment. Each dot represents an independent mouse. Statistical significance was determined using a two-tailed unpaired Student's t test. $*p < 0.05$. **d**, H&E staining of lungs treated with control or clodronate liposome (left) and quantification of average tumour size (right). Each dot represents an independent tumour mass from 4 independent mice. Statistical significance was determined using a two-tailed unpaired Student's t test. $****p < 0.0001$. **e**, Representative confocal images showing reduced expansion of neutrophils and $\gamma\delta$ T cells. RFP (red, tumour), Ly6G (green, upper panel), GL3 (green, lower panel), and DAPI (blue), scale bar, 100 μm . **f**, Dotplot showing the expression of key chemokine receptors in neutrophils and $\gamma\delta$ T cells. **g**, Flow cytometry analysis (left) and quantification (right) of neutrophils. Each dot represents an independent mouse. **h**, Flow cytometry analysis (left) and quantification (right) of $\gamma\delta$ T cells. Each dot represents an independent mouse. Statistical significance was determined using a two-tailed unpaired Student's t test. $*p < 0.05$.

-Regarding macrophage expansion, the study does not specify whether this results from the proliferation of resident alveolar macrophages, the recruitment and differentiation of monocytes, or both. Experimental clarification is needed.

Following the Reviewer's comments, we first investigated the characteristics of macrophages expanded in the inter-tumour region. IF staining revealed that macrophages expanding in the inter-tumour region were $F4/80^+$ macrophages co-expressed PDL-1 and CD11c, markers specific to alveolar macrophages and not interstitial/monocyte-derived macrophages (**Revised Fig. 2h**; **Revised Ex Fig. 2d**; reproduced in **Rev4_Fig. 7a, b**). To directly assess whether these reprogrammed AMs originate from monocytes, we orthotopically engrafted RFP^+ mutant organoids derived from *Red2Kras* lungs into the lungs of *CCR2-Cre^{ERT2}; Rosa-ZsGreen* mice, allowing lineage tracing of monocyte-derived cells (**Revised Ex Fig. 3a**; reproduced in **Rev4_Fig. 7c**). In these engrafted tumours, most macrophages surrounding RFP^+ tumour cells lacked *ZsGreen* labelling, indicating that reprogrammed AMs arise primarily from resident alveolar

macrophages rather than recruited monocytes (**Revised Ex Fig. 3b**; reproduced in **Rev4_Fig. 7d**). Only a small fraction of ZsGreen⁺ monocyte-derived macrophages was observed, and these were distributed both within and around tumours. Together, these results suggest that the macrophages expanding in the inter-tumour region during early tumour development are largely derived from resident alveolar macrophages and are actively reprogrammed by the emerging tumour niche. These data have been included in our revised manuscript.

Reviewer4_Figure 7. Reprogrammed macrophages expanding in the inter-tumour region originate primarily from resident alveolar macrophages during early tumour development. **a, b.** Representative confocal images showing alveolar macrophage markers in *Red2Kras* lungs at 4 weeks post-induction. F4/80 (green, a), RFP (red, tumours), PDL-1 (white, a), CD64 (green, b), CD11C (white, b), and DAPI (blue), scale bar, 100 μ m. **c.** Experiment scheme of orthotopic engraftment of RFP⁺ tumour organoids into the lungs of *CCR2-Cre^{ERT2}; ZsGreen* mice. **d.** Representative confocal images showing macrophages expanded around tumours. RFP (red, tumours), ZsGreen (lineage-traced monocytes, green), CD64 (white), and DAPI (blue), scale bar, 100 μ m. Insets (bottom) are enlargements of the dashed boxes (top). Scale bars, 50 μ m.

-While the study highlights a pro-inflammatory environment, it does not show how this state benefits tumor progression, especially given that inflammation can have a dual or opposing effects on cancer development.

We thank the Reviewer for this thoughtful comment. We agree that inflammation can exert both tumour-promoting and tumour-suppressive effects depending on the context, timing, and cellular composition. As addressed in this response above and our revised manuscript, depletion of macrophages during the early phase of tumour development markedly impaired tumour initiation, coinciding with reduced macrophage expansion and altered recruitment of neutrophils and $\gamma\delta$ -T cells (**Revised Fig. 2**; **Rev4-Fig. 6**). These results suggest that the establishment of a pro-inflammatory niche is essential for early tumour progression. However, as the Reviewer rightly notes, inflammatory responses may exert anti-tumour functions at later stages of tumour evolution,

such as during metastasis. We have now included these results and an expanded discussion of this dual role of inflammation in the revised manuscript.

-How generalizable are the findings also remains to be demonstrated in this study.

We thank the Reviewer for this valuable comment. To assess whether *EGFR*-mutant tumours engage similar epithelial-stromal reprogramming mechanisms, we re-analysed publicly available single-cell transcriptomics datasets from patients with *EGFR*-mutant adenocarcinoma (LUAD) and matched normal lung tissues (4 paired samples; **Rev4-Fig. 8a**). Sub-clustering of EpCAM⁺ epithelial cells identified an *AREG*^{high} population specifically enriched in *EGFR*-mutant LUAD, relative to AT2 cells from normal lungs (**Rev4-Fig. 8b-f**). Notably, this *AREG*^{high} population exhibited reduced expression of AT2 markers, such as *ETV5* and *LAMP3*, and increased expression of DATP-associated markers, such as *SOX9* and *ITGA2*, suggesting the emergence of a distinct, reprogrammed epithelial state in *EGFR*-mutant LUAD (**Rev4-Fig. 8f**). Furthermore, re-clustering of *COL1A1*⁺ mesenchymal cells revealed a fibrotic mesenchymal population characterised by *CTHRC1*, *RUNX1* and *RUNX2* expression, which was selectively enriched in *EGFR*-mutant LUAD specimens (**Rev4-Fig. 8b, c, g-i**). Importantly, both *AREG*-expressing tumour cells and these fibrotic mesenchymal states were present in stage-I *EGFR*-mutant LUAD, suggesting that fibrotic niche formation is initiated early and may similarly be driven by *EGFR*-mutant cells via the AREG – EGFR signalling axis (**Rev4-Fig. 8j, k**).

Reviewer4_Figure 8. Fibrotic niche signatures co-emerge with AREG-expressing tumour cells in early EGFR-mutant lung adenocarcinoma (LUAD). **a**, Schematic of the workflow for analysing scRNA-seq dataset of early-stage (I-III) human LUAD samples. Patients with confirmed *EGFR* mutations were selected, comprising 4 LUAD and 4 matched distant background lung samples. **b**, UMAP plot showing all integrated scRNA-seq from the *EGFR*-mutant LUAD dataset. **c**, Dotplot showing the expression of key marker genes distinguishing epithelial (*EpCAM*) and mesenchymal (*COL1A1*) cell clusters. **d**, UMAP plot of epithelial cell sub-clustering from (**b**), revealing 7 distinct populations, colour-coded by cellular states. **e**, UMAP plot showing epithelial cell distribution, colour-coded by dataset origin (*EGFR*-mutant LUAD vs normal). **f**, Heatmap showing marker gene expression in (**d**). **g**, UMAP plot of mesenchymal cell sub-clustering from (**b**), revealing 7 distinct populations, colour-coded by cellular states. **h**, UMAP plot showing mesenchymal cell distribution, colour-coded by dataset origin (*EGFR*-mutant LUAD vs normal). **i**, UMAP feature plots showing the expression of key marker genes for reprogrammed fibroblasts across mesenchymal clusters in (**g**). **j**, **k**, UMAP plots showing the distribution of epithelial cells (**j**) and mesenchymal cells (**k**), colour-coded by LUAD stage of dataset origin.

To further validate whether *EGFR* mutant cells can directly remodel their microenvironment, we developed a 3D inducible *EGFR*-mutant LUAD model by introducing EGFP reporter system to express *EGFR*^{L858R} specifically in primary mouse AT2 cells (**Revised Ex Fig. 18a**; reproduced in **Rev4-Fig. 9a**). Lineage-labelled AT2 cells were isolated, cultured as organoids, transduced with lentivirus at day 14 to induce *EGFR*^{L858R} expression alongside GFP, and subsequently orthotopically engrafted into NSG mouse lungs (**Revised Ex Fig. 18a, b**; reproduced in **Rev4-Fig. 9a, b**). After three weeks, *tdTom*⁺*EGFP*⁺ mutant cells formed expanding lesions enriched for Krt8 and Areg expression, hallmark features of DATP-like states also observed in *Kras*^{G12D}-driven models (**Revised Ex Fig. 18c, d**; reproduced in **Rev4-Fig. 9c, d**). Notably, engraftment of *EGFR*^{L858R}-mutant cells induced *Pdgfr*β⁺ fibrotic fibroblasts and macrophage expansion in adjacent niches (**Revised Ex Fig. 18e**; reproduced in **Rev4-Fig. 9c, d**).

Reviewer4_Figure 9. Establishment of pre-neoplastic niches driven by *EGFR*-mutant cells. a. Experiment scheme of the 3D inducible LUAD model established by introducing *EGFR*^{L858R} mutation into primary AT2 cells, followed by orthotopic engraftment into NSG mouse lungs. **b.** Flow cytometry analysis showing AT2 cells transduced with *EGFR*^{L858R} mutation. **c.** Representative confocal images showing that engrafted *tdTom*⁺*EGFP*⁺ mutant cells reprogram into *Krt8*⁺ cell states. **d.** Representative confocal images showing enriched *Areg* expression in *Krt8*⁺ mutant cell states (top) and the associated remodelling of fibrotic fibroblasts and macrophages in tumour regions (bottom). Tomato (red, tumours), *Areg* (green, d), *Pdgfr*β (green, d), *Krt8* (white, c,d), *CD64* (white, d), and DAPI (blue), scale bar, 100 μm.

Together, these findings demonstrate that oncogenic *EGFR* mutation reprograms AT2 cells into *Areg*⁺ states that actively remodel surrounding fibroblasts and macrophages, establishing a fibrotic, pro-tumorigenic niche. This identifies an epithelial-stromal reprogramming axis as a conserved early mechanism of niche formation across LUAD oncogenic drivers, extending beyond *KRAS* mutations. Supporting the broader significance of this pathway, a recent preprint (accompanying manuscript by Skrupskelyte *et al.*)¹¹ shows that survival of carcinogen-induced oesophageal squamous tumours similarly relies on EGF-mediated crosstalk between Sox9⁺ nascent tumour cells and fibrotic fibroblasts, illustrating that analogous epithelial-fibroblast circuits shape tumour initiation in diverse tissues. Together, these observations underscore the generalisability of our findings and highlight a shared *EGFR*-dependent pathway orchestrating early tumour development across cancer types.

References

- 1 Konkimalla, A. *et al.* Transitional cell states sculpt tissue topology during lung regeneration. *Cell Stem Cell* **30**, 1486–1502 e1489 (2023). <https://doi.org/10.1016/j.stem.2023.10.001>
- 2 Fang, Y. *et al.* RUNX2 promotes fibrosis via an alveolar-to-pathological fibroblast transition. *Nature* **640**, 221–230 (2025). <https://doi.org/10.1038/s41586-024-08542-2>
- 3 Halperin, C. *et al.* Global DNA Methylation Analysis of Cancer-Associated Fibroblasts Reveals Extensive Epigenetic Rewiring Linked with RUNX1 Upregulation in Breast Cancer Stroma. *Cancer Res* **82**, 4139–4152 (2022). <https://doi.org/10.1158/0008-5472.CAN-22-0209>
- 4 Zhang, X., Guo, R., Kambara, H., Ma, F. & Luo, H. R. The role of CXCR2 in acute inflammatory responses and its antagonists as anti-inflammatory therapeutics. *Curr Opin Hematol* **26**, 28–33 (2019). <https://doi.org/10.1097/MOH.0000000000000476>
- 5 Koyama, S. *et al.* STK11/LKB1 Deficiency Promotes Neutrophil Recruitment and Proinflammatory Cytokine Production to Suppress T-cell Activity in the Lung Tumor Microenvironment. *Cancer Res* **76**, 999–1008 (2016). <https://doi.org/10.1158/0008-5472.CAN-15-1439>
- 6 Quail, D. F. *et al.* Neutrophil phenotypes and functions in cancer: A consensus statement. *J Exp Med* **219** (2022). <https://doi.org/10.1084/jem.20220011>
- 7 Jin, C. *et al.* Commensal Microbiota Promote Lung Cancer Development via gammadelta T Cells. *Cell* **176**, 998–1013 e1016 (2019). <https://doi.org/10.1016/j.cell.2018.12.040>
- 8 Coffelt, S. B., Wellenstein, M. D. & de Visser, K. E. Neutrophils in cancer: neutral no more. *Nat Rev Cancer* **16**, 431–446 (2016). <https://doi.org/10.1038/nrc.2016.52>
- 9 Tsukui, T. *et al.* Collagen-producing lung cell atlas identifies multiple subsets with distinct localization and relevance to fibrosis. *Nat Commun* **11**, 1920 (2020). <https://doi.org/10.1038/s41467-020-15647-5>
- 10 Kim, N. *et al.* Single-cell RNA sequencing demonstrates the molecular and cellular reprogramming of metastatic lung adenocarcinoma. *Nat Commun* **11**, 2285 (2020). <https://doi.org/10.1038/s41467-020-16164-1>
- 11 Skrupskelyte, G. *et al.* Pre-cancerous Niche Remodelling Dictates Nascent Tumour Survival. *bioRxiv*, 2024.2007.2004.602022 (2024). <https://doi.org/10.1101/2024.07.04.602022>
- 12 England, F. J. *et al.* Sustained NF-kappaB activation allows mutant alveolar stem cells to co-opt a regeneration program for tumor initiation. *Cell Stem Cell* **32**, 375–390 e379 (2025). <https://doi.org/10.1016/j.stem.2025.01.011>
- 13 Palmieri, E. M. *et al.* Pharmacologic or Genetic Targeting of Glutamine Synthetase Skews Macrophages toward an M1-like Phenotype and Inhibits Tumor Metastasis. *Cell Rep* **20**, 1654–1666 (2017). <https://doi.org/10.1016/j.celrep.2017.07.054>
- 14 Tsukui, T., Wolters, P. J. & Sheppard, D. Alveolar fibroblast lineage orchestrates lung inflammation and fibrosis. *Nature* **631**, 627–634 (2024). <https://doi.org/10.1038/s41586-024-07660-1>
- 15 Li, Z. *et al.* Alveolar Differentiation Drives Resistance to KRAS Inhibition in Lung Adenocarcinoma. *Cancer Discov* **14**, 308–325 (2024). <https://doi.org/10.1158/2159-8290.CD-23-0289>
- 16 Kadur Lakshminarasimha Murthy, P. *et al.* Human distal lung maps and lineage hierarchies reveal a bipotent progenitor. *Nature* **604**, 111–119 (2022). <https://doi.org/10.1038/s41586-022-04541-3>
- 17 Basil, M. C. *et al.* Human distal airways contain a multipotent secretory cell that can regenerate alveoli. *Nature* **604**, 120–126 (2022). <https://doi.org/10.1038/s41586-022-04552-0>
- 18 Choi, J. *et al.* Inflammatory Signals Induce AT2 Cell-Derived Damage-Associated Transient Progenitors that Mediate Alveolar Regeneration. *Cell Stem Cell* **27**, 366–382 e367 (2020). <https://doi.org/10.1016/j.stem.2020.06.020>
- 19 Sun, R. *et al.* Amphiregulin couples IL1RL1(+) regulatory T cells and cancer-associated fibroblasts to impede antitumor immunity. *Sci Adv* **9**, eadd7399 (2023). <https://doi.org/10.1126/sciadv.add7399>
- 20 Zaiss, D. M. *et al.* Amphiregulin enhances regulatory T cell-suppressive function via the epidermal growth factor receptor. *Immunity* **38**, 275–284 (2013). <https://doi.org/10.1016/j.immuni.2012.09.023>
- 21 Savage, T. M. *et al.* Amphiregulin from regulatory T cells promotes liver fibrosis and insulin resistance in non-alcoholic steatohepatitis. *Immunity* **57**, 303–318 e306 (2024). <https://doi.org/10.1016/j.immuni.2024.01.009>

- 22 Panetti, C. *et al.* The co-inhibitory receptor TIGIT promotes tissue-protective functions in T cells. *Nat Immunol* (2025). <https://doi.org/10.1038/s41590-025-02300-w>
- 23 Kaiser, K. A., Loffredo, L. F., Santos-Alexis, K. L., Ringham, O. R. & Arpaia, N. Regulation of the alveolar regenerative niche by amphiregulin-producing regulatory T cells. *J Exp Med* **220** (2023). <https://doi.org/10.1084/jem.20221462>
- 24 Zhou, J. *et al.* Cancer-associated fibroblasts derived amphiregulin promotes HNSCC progression and drug resistance of EGFR inhibitor. *Cancer Lett* **622**, 217710 (2025). <https://doi.org/10.1016/j.canlet.2025.217710>
- 25 Mucciolo, G. *et al.* EGFR-activated myofibroblasts promote metastasis of pancreatic cancer. *Cancer Cell* **42**, 101–118 e111 (2024). <https://doi.org/10.1016/j.ccell.2023.12.002>
- 26 Hongu, T. *et al.* Perivascular tenascin C triggers sequential activation of macrophages and endothelial cells to generate a pro-metastatic vascular niche in the lungs. *Nat Cancer* **3**, 486–504 (2022). <https://doi.org/10.1038/s43018-022-00353-6>
- 27 Murdamoothoo, D. *et al.* Tenascin-C immobilizes infiltrating T lymphocytes through CXCL12 promoting breast cancer progression. *EMBO Mol Med* **13**, e13270 (2021). <https://doi.org/10.15252/emmm.202013270>

Referees' comments:

Referee #1 (Remarks to the Author):

The authors have nicely addressed most of my comments, adding substantial mechanistic insights to their work.

Several remaining issues should be addressed before this manuscript is accepted for publication.

1. In response to my comment on Figure 1, the authors added new staining and claim that “IF co-staining for *Pdgfra* and *Pdgfrβ* in both Confetti and *Red2Kras* lungs showed reduced *Pdgfra* expression in reprogrammed fibroblasts expressing *Pdgfrβ* located within RFP+ tumours (Revised Fig. 1i; reproduced in Rev1-Fig. 2b).” I still see a lot of PDGFRalpha staining in the representative image. Can the authors explain this? This should be quantified.

Response: We appreciate the Reviewer’s valuable comment, which allowed us to clarify our conclusions. As proposed in the original manuscript, we believe that at 2 weeks post-induction, fibroblasts undergo a transition toward fibrotic states, characterized by a reduction in *Pdgfra* expression and the acquisition of fibrotic markers, including *Pdgfrβ*. Consistent with the Reviewer’s observation, we detected reduced *Pdgfra* expression in fibroblasts in close proximity to mutant cells, as well as fibroblasts co-expressing both *Pdgfra* and *Pdgfrβ* within individual clones, indicative of a mosaic transitional state (**Revised Fig. 1i**). Importantly, we observed that the majority of fibroblasts in small clones co-express both markers, whereas fibroblasts in large clones have completely lost *Pdgfra* expression and exclusively express *Pdgfrβ* (**Rev1-Fig. 1**). Together, these data strongly support a model in which *Pdgfra*⁺ alveolar fibroblasts undergo progressive reprogramming toward a fibrotic phenotype. We have clarified this transitional mosaic phenotype in the revised manuscript (*highlighted in yellow on page 6*).

Reviewer1_Figure 1. *Pdgfra*⁺ alveolar fibroblasts transition into *Pdgfrβ*⁺ fibrotic states during early oncogenesis. Representative confocal images showing fibroblasts and lineage-labelled mutant AT2 cells in *Red2Kras* lungs at 2 weeks following *Kras* activation. DAPI (blue), *Pdgfra* (grey), *Pdgfrβ* (green) and RFP (red, tumour). Right panels are magnifications of left panels. Scale bar, 50μm. Images representative of 3 independent biological replicates.

2. The authors nicely show the spatiotemporal dynamics of reprogrammed fibrotic fibroblasts in new Figure 3. However there is no pValue in Reviewer Figure 3B – if these results are not statistically significant, the authors should clearly state it and not present this as a change.

Response: As requested, we have included the statistical analysis in our revised manuscript (**Revised Fig. 3c**).

3. In many of the revised Figures (Reviewer Figure 3, 4, 5, 7) the authors show only 3 mice/Ns per group and do not mention in the Fig. legends whether these experiments have been repeated and how many times. Showing data from only 3 mice in only one experiment is not sufficient. This is also true for some of the new IF imaging experiments (for example Reviewer Fig 10) where the author show a representative image but do not specify how many mice or images were taken. This must be addressed.

Response: We thank the Reviewer for raising this important point. We have now explicitly clarified experimental reproducibility and sample sizes throughout the revised manuscript, with this information provided in the figure legends as well as in the “Confocal imaging, processing and quantification” and “Statistical analysis and reproducibility” sections of the Methods (*highlighted in yellow on pages 31 and 34-35*). Briefly, all *in vivo* experiments were performed in at least two independent experiments, with individual mice considered as biological replicates, and all *in vitro* assays were performed in at least three independent experiments, with summary statistics calculated from experiment-level mean values. Representative imaging data, including Reviewer Fig. 10 (Extended Data Fig. 18), and the corresponding quantifications were obtained by analyzing 10-20 tumours per mouse or a minimum of 7 fields of view per mouse or experimental condition, across the indicated numbers of animals and independent experiments. Furthermore, as requested by another Reviewer, we performed nested t test statistical analyses that accounted for within-mouse or within-experiment variability where appropriate, and confirmed that these results were consistent with analyses based on mouse-level (*in vivo*) or experiment-level (*in vitro*) means. All of this information has been incorporated into the revised manuscript.

4. The authors should cite and discuss previous studies that have not been mentioned - PMID 34624218 and PMCID: PMC9889778.

Response: As suggested, we have cited these studies in our revised manuscript (*yellow highlighted in page 19*).

Referee #3 (Remarks to the Author):

The revised manuscript incorporates substantial new analyses and experiments that fully address my previous concerns, clarifying the results and supporting the conclusions.

This study defines the temporal and spatial dynamics of fibroblasts and macrophages relative to mutant AT2 cells during early oncogenesis. The authors demonstrate that mutant AT2 cells initiate fibrotic reprogramming of adjacent alveolar fibroblasts through an Areg–EGFR signaling axis, a key mechanism driving tumour–niche co-evolution. These fibrotic fibroblasts subsequently remodel the immune compartment by driving alveolar macrophage (AM) expansion and phenotypic rewiring through the Tnc–TLR4 axis, establishing a permissive immunosuppressive niche that promotes immune-cell recruitment and the emergence of inflammatory fibroblasts, ultimately contributing to tumor growth. Importantly, disrupting these tumor-niche interactions halts mutant-cell reprogramming as well as stromal and immune remodelling. The development of inducible human LUAD organoids further supports that similar microenvironmental reprogramming occurs in human disease, highlighting a potentially targetable window for therapeutic intervention in early-stage lung cancer.

Overall, the findings are solid and rigorous, well supported by multiple mouse and human models, and provide compelling evidence for a spatiotemporal signalling axis coordinating early tumor-niche evolution.

Thus, I believe the manuscript is suitable for publication in Nature.

However, I would like to suggest some minor changes to further improve the readiness and comprehensiveness of the study:

1. It would help to indicate in Figure legend in Fig.2, Ext Data Fig. 2, 4, what the markers indicate, i.e: CD3, T cells. What is GL3? Or what is CD11c or CD11b labeling?

Response: Following the Reviewer's suggestion, we have now included explicit descriptions of the markers and their corresponding cell types to the figure legends for Fig. 2, Extended Data Fig. 2, and Extended Data Fig. 4 in the revised manuscript.

2. Extended Data Fig. 3 is not referenced in the manuscript.

Response: We have described the Extended Data Fig. 3 in our manuscript (page: 7, line: 160-166).

3. Pag. 321, Fig 4 I-o, the increase in AT2 cells is not significant so in the text, it can only be indicated a tendency.

Response: We thank the Reviewer for this important point. As requested, we have amended this statement in our revised manuscript (page: 12, line: 328).

4. A more clear description of how cell counts were conducted. It seems that at least 7 fields per mouse lung were counted, in n=3 in some cases. If so, nested t test should have been performed to keep into account the variability between the different fields in 1 mouse. Additionally, the statistics paragraph should include how group size (n) was calculated.

Response: We thank the Reviewer for raising this important point. We have now explicitly clarified experimental reproducibility and sample sizes throughout the revised manuscript, with this information provided in the figure legends as well as in the "Confocal imaging, processing and quantification" and "Statistical analysis and reproducibility" sections of the Methods (*highlighted in yellow on pages 31 and 34-35*). Briefly, all *in vivo* experiments were performed in at least two independent experiments, with individual mice considered as biological replicates, and all *in vitro* assays were performed in at least three independent experiments, with summary statistics calculated from experiment-level mean values. Representative imaging data and the corresponding quantifications were obtained by analyzing 10-20 tumours per mouse or a minimum of 7 fields of view per mouse or experimental condition, across the indicated numbers of animals and independent experiments.

As suggested by the Reviewer, to account for variability among multiple fields quantified within individual mice/ experiments, we performed nested t-tests for the relevant datasets. This reanalysis led to the same conclusions as those reported in the original manuscript, in which the mean value per mouse/experiment was used for statistical comparisons (**Rev3-Fig. 1-3**). Therefore, we have retained the original statistical analysis, while including the nested analysis in this Response and noting in the Methods that nested analyses yield equivalent results. The Methods have been updated accordingly (*highlighted in yellow on page 34-35*).

Reviewer3_Figure 1.
Statistical analysis using two-tailed nested t test for the datasets presented in Figure 3 and Extended Data Figure 5. *P* values for each comparison are indicated in the panels. Each dot represents one analysed image. For panels **a–c**, each column represents one *in vivo* biological replicate (grouped by time point). For panels **d–f**, each column represents one *in vitro* biological replicate (grouped by condition). *, *P*<0.05; **, *P*<0.01.

Reviewer3_Figure 2. Statistical analysis using two-tailed nested t test for the datasets presented in Figure 4. *P* values for each comparison are presented in the panels. Each dot represents one analysed image. Each column represents one *in vivo* biological replicate (grouped by experimental condition). *, *P*<0.05; **, *P*<0.01.

Reviewer3_Figure 3. Statistical analysis using two-tailed nested t test for the datasets presented in Figure 5 and Extended Data Figure 13. *P* values for each comparison are presented in the panels. Each dot represents one analysed image. Each column represents one *in vivo* biological replicate (grouped by experimental condition or genotype). *, *P*<0.05; **, *P*<0.01.

Referee #4 (Remarks to the Author):

The authors have adequately addressed my comments in the revised manuscript, which is improved.

Response: We thank the Reviewer for their positive evaluation and are pleased that the revised manuscript addresses their comments.